# Benchmarking monolayer MoS$_2$ and WS$_2$ field-effect transistors

Amritanand Sebastian [1], Rahul Pendurthi[1], Tanushree H. Choudhury[2], Joan M. Redwing[2,3,4] & Saptarshi Das [1,3,4 ✉]

Here we benchmark device-to-device variation in field-effect transistors (FETs) based on monolayer MoS$_2$ and WS$_2$ films grown using metal-organic chemical vapor deposition process. Our study involves 230 MoS$_2$ FETs and 160 WS$_2$ FETs with channel lengths ranging from 5 μm down to 100 nm. We use statistical measures to evaluate key FET performance indicators for benchmarking these two-dimensional (2D) transition metal dichalcogenide (TMD) monolayers against existing literature as well as ultra-thin body Si FETs. Our results show consistent performance of 2D FETs across 1 × 1 cm$^2$ chips owing to high quality and uniform growth of these TMDs followed by clean transfer onto device substrates. We are able to demonstrate record high carrier mobility of 33 cm$^2$ V$^{-1}$ s$^{-1}$ in WS$_2$ FETs, which is a 1.5X improvement compared to the best reported in the literature. Our experimental demonstrations confirm the technological viability of 2D FETs in future integrated circuits.

[1] Department of Engineering Science and Mechanics, Penn State University, University Park, PA 16802, USA. [2] 2D Crystal Consortium-Materials Innovation Platform (2DCC-MIP), Penn State University, University Park, PA 16802, USA. [3] Department of Materials Science and Engineering, Penn State University, University Park, PA 16802, USA. [4] Materials Research Institute, Penn State University, University Park, PA 16802, USA. ✉email: sud70@psu.edu

Two-dimensional (2D) semiconducting materials beyond graphene[1,2] are receiving increasing attention owing to their ultra-thin body nature that can mitigate detrimental short-channel effects in aggressively scaled devices through improved electrostatics, enabling them to replace or complement the aging Si technology[3–5]. Molybdenum disulfide ($MoS_2$) and tungsten disulfide ($WS_2$), belonging to the family of transition metal dichalcogenides (TMDs), have been studied extensively in this context. In fact, high performance $MoS_2$ field-effect transistors (FETs) with a contact pitch of 70 nm and 42 nm have already been experimentally demonstrated[6,7]. Circuit level implementations of 2D FETs such as inverters, logic operators, ring oscillators, and radio-frequency devices have also been achieved[8–12]. Recently, a microprocessor based on $MoS_2$ FETs was reported[13]. Additionally, 2D FETs have found applications in various emerging technologies such as sensors for internet of things, neuromorphic computing, biomimetic devices, valleytronics, straintronics, and optoelectronic devices[14–21]. While initial demonstrations of prototype devices relied on exfoliated flakes, the 2D community has rapidly transitioned towards the growth of large-area films to address manufacturing needs for any commercial applications. In this context, chemical vapor deposition (CVD)[22,23] and metal-organic CVD (MOCVD)[7,24] are the most promising techniques, enabling growth of high quality 2D materials with different thermal budgets on various substrates. In fact, there are several reports demonstrating high-performance FETs based on CVD and MOCVD grown monolayer $MoS_2$ and $WS_2$. However, most of these studies are based on one or only a few devices.

To assess the potential of 2D materials for future very large scale integrated (VLSI) circuits, it is important to study the variation in key device parameters that determine the ON-state and OFF-state performance across a large number of devices. Unfortunately, there are only a few studies that report device-to-device variation in 2D FETs[7,25,26]. Smithe et al. measured multiple parameters across 200 $MoS_2$ FETs and demonstrated low threshold voltage variation and low contact resistance on the order of $1 \, k\Omega - \mu m$[25]. Similarly. Xu et al. analyzed 380 top-gated $MoS_2$ FETs and reported variation in threshold voltage and electron mobility[26]. However, both works concentrate on longer channel devices where the effects of contact resistance are not pronounced. In a separate study, Smithe et al.[22] measured scaled $MoS_2$ FETs based on synthetic monolayers; however, they did not provide any statistics. Smets et al.[7] demonstrated the most significant study on scaling of CVD grown monolayer $MoS_2$, wherein multiple devices with channel lengths ranging from 5 μm down to 29 nm were measured. However, their study was focused on the OFF-state performance. Finally, all of the aforementioned studies are based on $MoS_2$ FETs, and none exist for $WS_2$ FETs.

This work focuses on a comprehensive study of variation in key parameters related to both OFF-state and ON-state performance, such as threshold voltage, subthreshold slope, ratio of maximum to minimum current, field-effect carrier mobility, contact resistance, drive-current, and carrier saturation velocity, for different channel lengths ranging from 5 μm down to 100 nm using 230 $MoS_2$ FETs and 160 $WS_2$ FETs. In addition, we offer extensive benchmarking of our devices with respect to the abovementioned demonstrations as well as ultra-thin body (UTB) silicon (Si) on insulator (SOI) FETs with similar gate lengths to assess the technological viability and maturity of 2D FETs. Using statistical measures such as mean, median, standard deviation, and minimum/maximum values, we show low device-to-device variation. We are also able to demonstrate record high carrier mobility of $33 \, cm^2 V^{-1} s^{-1}$ in $WS_2$ FETs, which is a 1.5X improvement compared to the best reported in the literature. We

attribute our accomplishments to the epitaxial growth of highly crystalline 2D monolayers on sapphire substrate via MOCVD technique at 1000 °C using chalcogen and sulfur precursors that minimize carbon contamination in the film, as well as to the clean transfer of the film from the growth substrate to the device fabrication substrate.

## Results

### Synthesis and characterization of monolayer $MoS_2$ and $WS_2$.
$MoS_2$ and $WS_2$ were deposited by MOCVD on epi-ready 2″ diameter c-plane sapphire wafers. Figure 1 summarizes the growth, structural, and optical characterization of the MOCVD grown $MoS_2$ and $WS_2$. Figure 1a shows the schematic of the MOCVD system, comprising of a cold-wall horizontal reactor with an inductively heated graphite susceptor equipped with wafer rotation as previously described[27]. Molybdenum hexacarbonyl ($Mo(CO)_6$) and tungsten hexacarbonyl ($W(CO)_6$) were used as metal precursors, while hydrogen sulfide ($H_2S$) was the chalcogen source with $H_2$ as the carrier gas. $MoS_2$ was deposited in a single step process at 1000 °C, where coalesced monolayer growth across the 2″ wafer was achieved in 18 min. $WS_2$ was deposited using a multi-step process with nucleation at 850 °C and lateral growth at 1000 °C, resulting in coalesced monolayer growth across the 2″ wafer in 10 min[28]. In both cases, after growth the substrate was cooled in $H_2S$ to 300 °C to inhibit decomposition of the $MoS_2$ and $WS_2$ films. Figure 1b shows uniformly grown $MoS_2$ and $WS_2$ films over 2″ sapphire wafers. Further growth details can be found in the "Methods" section. The morphology of the monolayer films at the center and edge of the 2″ wafer is shown in Fig. 1c, d for $MoS_2$ and $WS_2$, respectively, using atomic force microscopy (AFM). Height profiles obtained from scratch testing confirm monolayer film formation (see Supplementary Fig. 1a, b). The monolayers are fully coalesced, with undulations arising from steps on the sapphire surface. The overall bilayer density is low but a higher density of bilayers is present at the center of the $MoS_2$ film compared to the $WS_2$ film. The in-plane X-ray diffraction (XRD) patterns in Fig. 1e, f highlight the epitaxial relation between the sulfide monolayers and the underlying sapphire substrates. The full-width at half maxima of the φ-scan peaks are 0.3° and 0.17° for $MoS_2$ and $WS_2$, respectively, indicating a low rotational misorientation of domains within the monolayers. The films were transferred to $Al_2O_3$/Pt/TiN/$p^{++}$-Si substrates for device fabrication, as discussed later. The transferred film quality was assessed using Raman maps as shown in Fig. 1g, h, and photoluminescence (PL) maps as shown in Fig. 1i, j, for $MoS_2$ and $WS_2$, respectively. Raman maps show less than 5% variation in the representative $A_{1g}$ peak position. The uniform PL peak positions observed at 1.84 eV for $MoS_2$ and 1.97 eV for $WS_2$ correspond to their monolayer response. Representative Raman and PL spectra are included in the Supplementary Fig. 1c–f.

### Monolayer $MoS_2$ and $WS_2$ device fabrication and characterization.
To investigate the electrical properties of the MOCVD grown TMD films, back-gated FETs were fabricated on $Al_2O_3$/Pt/TiN/$p^{++}$-Si substrates. 50 nm $Al_2O_3$ gate dielectric was deposited using atomic layer deposition (ALD). The choice of a thin, high-k gate dielectric with an effective oxide thickness (EOT) of 22 nm, compared to conventionally used 300 nm $SiO_2$, was to allow for better gate electrostatics. The Pt/TiN/$p^{++}$-Si stack acts as the gate electrode (see "Methods" section for more details on gate dielectric fabrication) for each substrate. The TMD films were transferred from sapphire (growth substrates) onto the $Al_2O_3$ substrates via the poly(methyl methacrylate) (PMMA)-assisted wet-transfer process[29], as shown in Fig. 2a (see "Methods" section

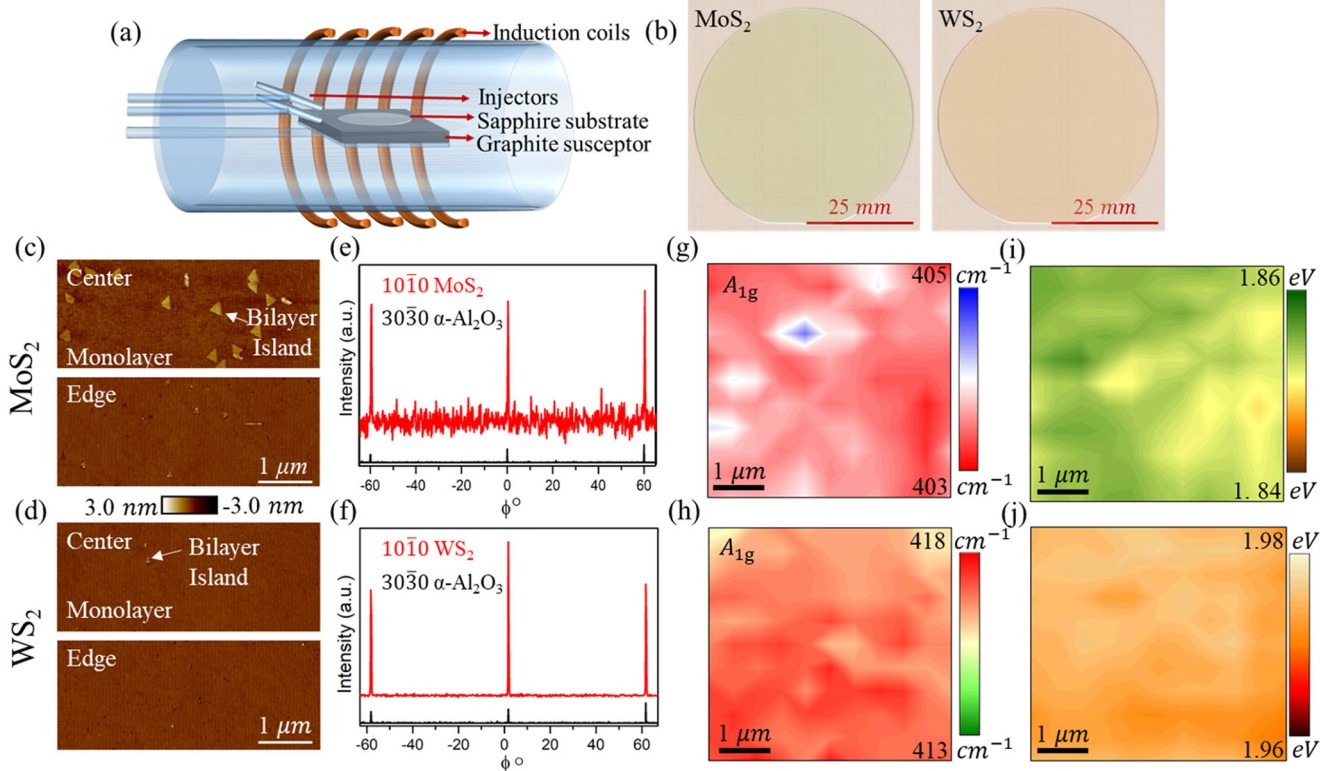

**Fig. 1 Monolayer film growth and characterization. a** Schematic of the MOCVD system with a cold-wall horizontal reactor. **b** 2″ sapphire wafer with MOCVD grown $MoS_2$ and $WS_2$. AFM images of **c** $MoS_2$ and **d** $WS_2$ at the center and edge of the respective wafers. Towards the center of the $MoS_2$ film, few bilayers are seen. In-plane XRD ϕ-scan of **e** $MoS_2$ and **f** $WS_2$ on sapphire (α-$Al_2O_3$), showing the epitaxial relationship between the monolayers and the sapphire substrate. Raman map of the $A_{1g}$ peak position for **g** $MoS_2$ and **h** $WS_2$ films transferred from the growth substrate onto the device fabrication substrate with 50 nm ALD $Al_2O_3$. Low variation in the peak position is observed for both $MoS_2$ and $WS_2$ with an average of ≈403.5 $cm^{-1}$ and ≈417 $cm^{-1}$, respectively. PL peak map of **i** $MoS_2$ with an average of ≈1.85 eV and **j** $WS_2$ with an average of ≈1.97 eV, confirm monolayer films. PL is a characteristic of monolayer film owing to indirect to direct bandgap transition.

for more details on transfer of monolayer films). Following transfer, electron beam (e-beam) lithography and dry etching using $SF_6$ plasma were used to isolate the channel area of each device. Next, transmission line measurement (TLM) structures were defined using another set of e-beam exposures. Finally, e-beam evaporation was performed to sequentially deposit 40 nm Ni and 30 nm Au to serve as the contacts for the FETs (see "Methods" section for more details on device fabrication). The TLM structures were designed to have channel lengths ($L_{CH}$) of 100 nm, 200 nm, 300 nm, 400 nm, 500 nm, 1 μm, 2 μm, 3 μm, 4 μm, and 5 μm, while the channel width ($W$) was kept constant at 5 μm. Figure 2b, c, respectively, show the schematic and scanning electron microscope (SEM) image of the fabricated TLM structures. Figure 2d–g show the transfer characteristics, i.e., drain current ($I_{DS}$) versus gate voltage ($V_{GS}$), for different drain voltages ($V_{DS}$) in linear and logarithmic scales for representative longest-channel length ($L_{CH} = 5$ μm) and shortest-channel length ($L_{CH} = 100$ nm) FETs, for both $MoS_2$ and $WS_2$. Strong n-type conduction is observed due to Fermi-level pinning of the contact metal close to the conduction band of both $MoS_2$ and $WS_2$[30]. Figure 2h–k show the corresponding output characteristics, i.e., $I_{DS}$ versus $V_{DS}$, for different $V_{GS}$. Measurement protocols are described in the "Methods" section.

**Device-to-device variation in monolayer $MoS_2$ and $WS_2$ FETs.** To understand the variation in the FET performance across the entire $1 \times 1$ $cm^2$ substrates, as well as to study of the impact of channel length scaling on FET performance, 230 $MoS_2$ FETs (23 TLM structures) and 160 $WS_2$ FETs (16 TLM structures) were

measured. Figure 3a, b display the transfer characteristics of all measured $MoS_2$ and $WS_2$ FETs, respectively, for different $L_{CH}$, which were used to extract key device parameters. For each parameter, the mean, median, standard deviation, minimum, and maximum values are reported. Finally, median values are used for benchmarking since they reflect the central tendency, even in the presence of outliers in the data, and offer higher accuracy in case of skewed distributions. Devices with the best number for a given parameter are termed as "champion" devices.

*Threshold voltage.* Threshold voltage is extracted using three different methods: linear extrapolation ($V_{t_{lin}}$), Y-function ($V_{t_Y}$), and constant-current method ($V_{t_{cc}}$). Supplementary Fig. 2a–c describes the extraction of $V_{t_{lin}}$, $V_{t_Y}$, and $V_{t_{cc}}$, Supplementary Fig. 2d, e show their corresponding median values as a function of $L_{CH}$, and Supplementary Fig. 2f, g show their distributions across all devices for $MoS_2$ and $WS_2$, respectively. Supplementary Note 1 and Supplementary Table 1 summarize the device-to-device variations. It was found that the threshold voltage is independent of the channel length for both $MoS_2$ and $WS_2$ FETs. Figure 4a, b show the distributions of $V_{t_{lin}}$ for all measured $MoS_2$ and $WS_2$ FETs, respectively. Median $V_{t_{lin}}$ of 2.9 V with a standard deviation of $\sigma_{V_t} = 0.8$ V is obtained for $MoS_2$, and median $V_{t_{lin}}$ of 6.4 V with a $\sigma_{V_t} = 0.8$ V is obtained for $WS_2$. Threshold voltage was found to be more positive for $WS_2$ FETs compared to $MoS_2$ FETs, which can be attributed to higher intrinsic n-type doping of $MoS_2$ either due to the specific nature of the impurity present in the MOCVD grown $MoS_2$ film or due to surface charge transfer induced

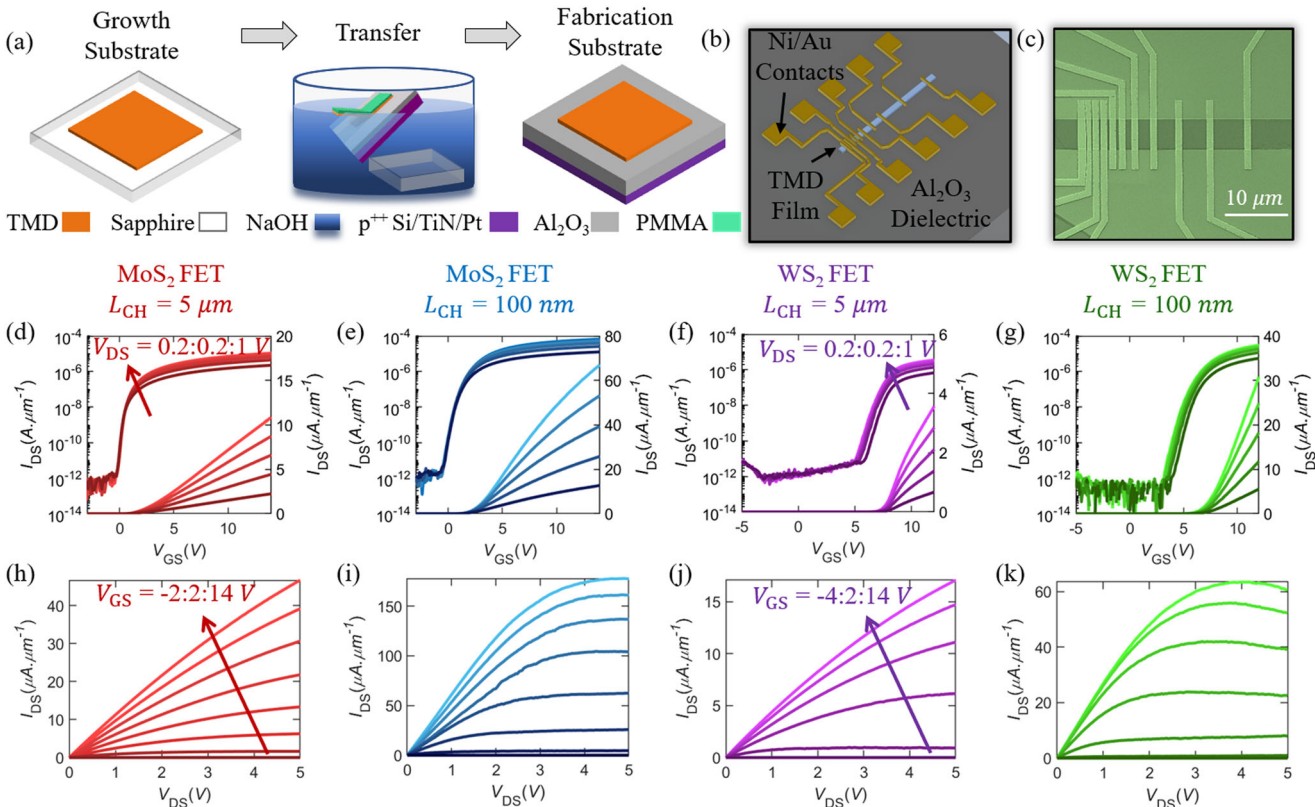

**Fig. 2 Device fabrication and electrical measurements. a** Schematic representation of PMMA-assisted wet transfer of monolayer TMD films from sapphire (growth substrate) to an $Al_2O_3$ substrate. **b** Schematic representation and **c** SEM of a TLM structure defined using e-beam lithography. TLM structures with channel length ($L_{CH}$) from 100 nm to 5 µm are fabricated and the channel width is defined to be 5 µm. Transfer characteristics, i.e., drain current ($I_{DS}$) versus gate voltage ($V_{GS}$), for different drain voltages ($V_{DS}$) in logarithmic and linear scale for **d** longest-channel ($L_{CH} = 5$ µm) and **e** shortest-channel ($L_{CH} = 100$ nm) $MoS_2$ field-effect transistors (FETs), and **f** longest-channel ($L_{CH} = 5$ µm) and **g** shortest-channel ($L_{CH} = 100$ nm) $WS_2$ FETs. Corresponding output characteristics, i.e., $I_{DS}$ versus $V_{DS}$, for different $V_{GS}$ for **h** longest-channel and **i** shortest-channel $MoS_2$ FETs, and **j** longest-channel and **k** shortest-channel $WS_2$ FETs.

doping due to the underlying ALD grown $Al_2O_3$. This charge transfer is accredited to the higher conduction band offset between $MoS_2$ and $Al_2O_3$ compared to $WS_2$ and $Al_2O_3$[31].

Variation in threshold voltage is widely used for benchmarking emerging devices based on novel materials[25]. Note that median $V_{t_{lin}}$ depends on the work function of the gate metal and unintentional/intrinsic doping of the 2D material and that both $V_{t_{lin}}$ and $\sigma_{V_t}$ depend on the thickness of the gate oxide. Hence for a fair comparison we use $S\sigma_{V_t}$, which is defined as the projected threshold voltage variation at a scaled effective oxide thickness (SEOT) obtained using Eq. (1). We use SEOT = 0.9 nm for comparison with other literature results.

$$S\sigma_{V_t} = \sigma_{V_t} \frac{SEOT}{EOT} \tag{1}$$

This equation assumes linear scaling of variation in threshold voltage with respect to the EOT. However, for ultra-scaled devices, deviation from the linear scaling can be expected due to increased effect of metal-gate granularity[32]. For our $MoS_2$ and $WS_2$ FETs, we project $S\sigma_{V_t} = 33$ mV, which is similar to the value projected for CVD grown monolayer $MoS_2$ FETs reported by Smithe et al.[25]. We also employed this method to other reports on top-gated and wafer-scale monolayer $MoS_2$ FETs and extracted $S\sigma_{V_t} = 45$ mV for[26] and $S\sigma_{V_t} = 11$ mV for[12], respectively. Recently, Smets et al.[7] have demonstrated $\sigma_{V_t} = 44$ mV for an EOT of 1.9 nm that would correspond to $S\sigma_{V_t} = 20$ mV for monolayer $MoS_2$ FETs with channel lengths scaled down to

30 nm. These results are compared with the state-of-the-art UTB SOI and Si FinFET (Table 1). Channel dimensions are included in Table 1 since $\sigma_{V_t}$ has been found to be inversely proportional to the channel area in ultra-scaled devices which is shown using Pelgrom plots[32,33]. However, we did not observe such a trend due to relatively large channel areas in our $MoS_2$ and $WS_2$ FETs. It is encouraging that monolayer 2D FETs show $S\sigma_{V_t}$ comparable to the state-of-the art Si FETs in spite of an order of magnitude smaller body thickness. Note that UTB Si FETs are expected to encounter challenges associated with the precise thickness control, random dopant fluctuations, and detrimental quantum confinement effects beyond 5 nm body thickness[34,35], which are unlikely for 2D monolayers. At the same time further improvement in threshold voltage variation can be achieved for 2D FETs through optimization of the monolayer growth and improvement in the fabrication process flow (see Supplementary Note 2 for further discussion). Hence, 2D materials offer an alternative for the realization of UTB MOSFETs. The exhibition of low device-to-device variation in this work, which can be attributed to uniform and contaminant-free MOCVD growth of monolayer TMDs and clean device fabrication process can accelerate the incorporation of 2D FETs in future VLSI technologies.

*Subthreshold slope.* Subthreshold slope (SS) is extracted over 1 ($SS_1$), 2 ($SS_2$), 3 ($SS_3$), and 4 ($SS_4$) orders of magnitude change in $I_{DS}$ for $MoS_2$ and $WS_2$ FETs, respectively. Supplementary Fig. 3a, b show the median values for $SS_1$, $SS_2$, $SS_3$, and $SS_4$ as a function of $L_{CH}$, and Supplementary Fig. 3c, d show the distributions for

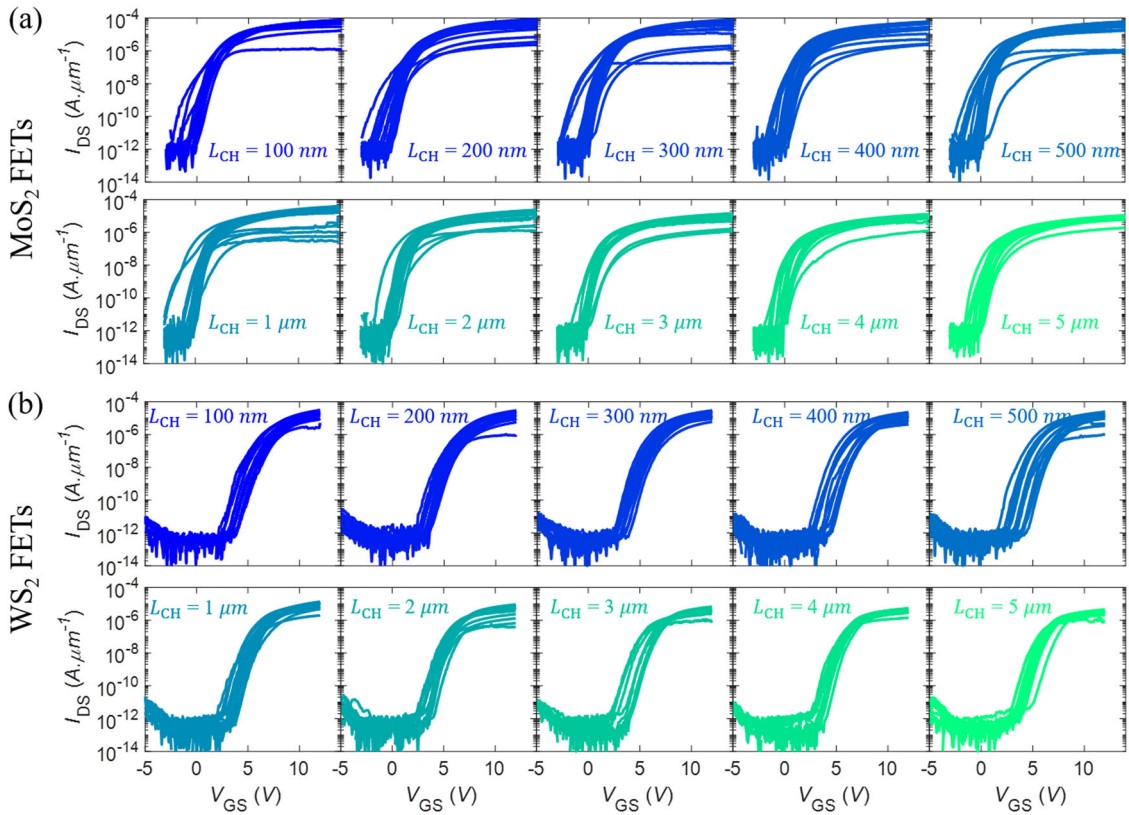

**Fig. 3 Statistics of scaled devices.** A total of 230 MoS$_2$ FETs and 160 WS$_2$ FETs were fabricated using 23 and 16 TLM structures with channel lengths ranging from $L_{CH} = 100$ nm to $L_{CH} = 5$ μm for **a** MoS$_2$ and **b** WS$_2$, respectively, to analyze the device-to-device variation and impact of scaling on the device performance. The corresponding transfer characteristics, i.e., $I_{DS}$ versus $V_{GS}$, for $V_{DS}$ of 1 V are shown in the logarithmic scale.

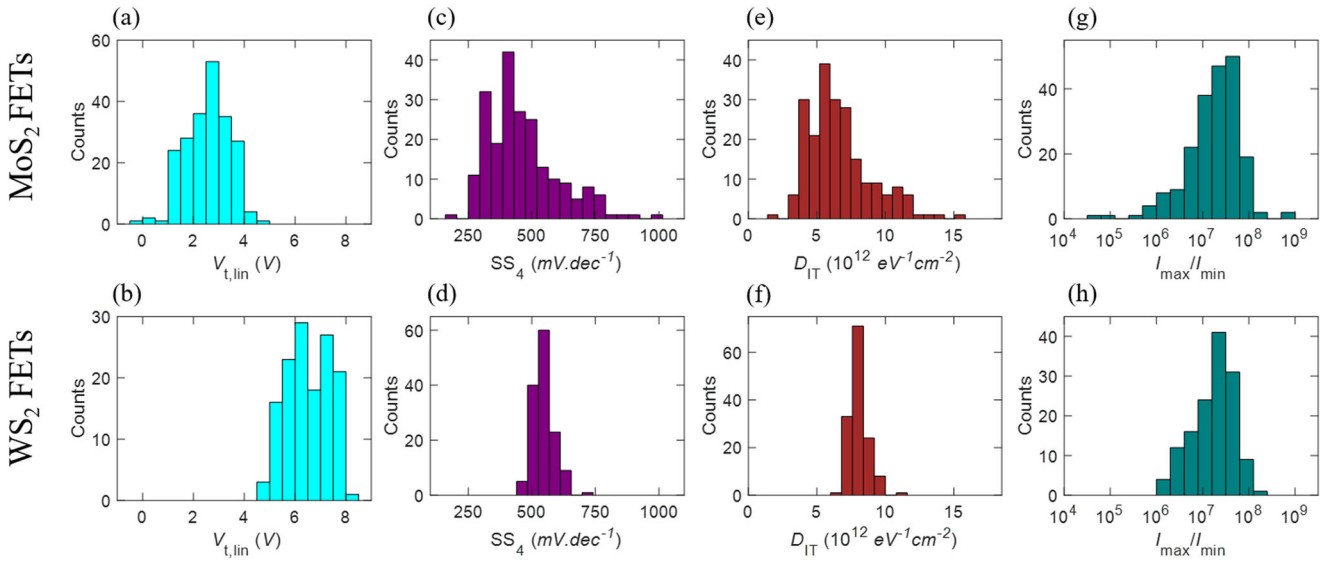

**Fig. 4 Variation in OFF-state performance.** Histograms showing the variation in threshold voltage extracted using linear extrapolation ($V_{t,lin}$) for **a** MoS$_2$ and **b** WS$_2$ FETs. The median values for these extracted threshold voltages were found to be more positive for WS$_2$ FETs compared to MoS$_2$ FETs due to higher intrinsic n-type doping of MoS$_2$. Histograms of SS extracted across 4 orders of magnitude change in the drain current (SS$_4$) for **c** MoS$_2$ and **d** WS$_2$ FETs. The deviation of SS from the ideal 60 mV.dec$^{-1}$ corresponds to the presence of interface traps. Histograms of interface trap density ($D_{IT}$) for **e** MoS$_2$ and **f** WS$_2$ FETs calculated from SS$_4$. Histograms of maximum to minimum current ratio ($I_{max}/I_{min}$) for **g** MoS$_2$ and **h** WS$_2$ FETs.

SS$_1$ and SS$_4$ for all MoS$_2$ and WS$_2$ devices, respectively. Supplementary Table 2 summarizes the device-to-device variation in SS. For a FET with ohmic contacts, it is expected that SS$_1$ = SS$_2$ = SS$_3$ = SS$_4$. However, for a Schottky barrier (SB) FET, the SS may increase when extracted for higher orders of magnitude change in

$I_{DS}$. A greater increase can be attributed to higher SB height at the metal/semiconductor interface, which not only limits the ON-current but also impacts the OFF-state performance. In the existing 2D FET literature there is a tendency to report SS value without mentioning the orders of magnitude change in $I_{DS}$ over

**Table 1 Benchmarking device-to-device variation in threshold voltage.**

| | $\sigma_{V_t}$ (V) | Gate dielectric | $S\sigma_{V_t}$ (V) at SEOT = 0.9 nm | Channel dimensions (µm) |
|---|---|---|---|---|
| [25]—MoS$_2$ | 1.05 | 30 nm SiO$_2$ | $33 \times 10^{-3}$ | $W = 11.6$, $L_{CH} = 4$-8.6 |
| [26]—MoS$_2$ 1 continuous layer | 0.25 | 30 nm HfO$_2$ | $45 \times 10^{-3}$ | $W = $ -, $L_{CH} = 30$ |
| [26]—MoS$_2$ 1 layer + ML | 0.1 | 30 nm HfO$_2$ | $19 \times 10^{-3}$ | $W = $ -, $L_{CH} = 30$ |
| [12]—MoS$_2$ | 0.17 | 30 nm Al$_2$O$_3$ | $11 \times 10^{-3}$ | $W = 30$, $L_{CH} = 4$ |
| [7]—MoS$_2$ | $44 \times 10^{-3}$ | 4 nm HfO$_2$ | $20 \times 10^{-3}$ | $W = 1$, $L_{CH} = 0.1$ |
| Our work-MoS$_2$, WS$_2$ | 0.8 | 50 nm Al$_2$O$_3$ | $33 \times 10^{-3}$ | $W = 5$, $L_{CH} = 0.1, 0.2, 0.3, 0.4, 0.5, 1, 2, 3, 4, 5$ |
| [33]—UTB SOI | $24.5 \times 10^{-3}$ | EOT = 1.65 nm | $13 \times 10^{-3}$ | $W = 0.060$, $L_{CH} = 0.025$ |
| [32]—FinFET | $10 \times 10^{-3}$ | EOT = 0.8 nm | $11 \times 10^{-3}$ | $W = 0.0075$, $L_{CH} = 0.034$ |

which it is evaluated. This leads to considerable discrepancy and unfair comparisons. In fact, most SS values are reported for only one or two orders of magnitude of the drain current, whereas circuit operations require at least four orders of magnitude ON/ OFF ratio to be technologically relevant.

We found that the median SS values are independent of $L_{CH}$ for both MoS$_2$ and WS$_2$ FETs (Supplementary Fig. 3a, b). Fig. 4c, d show the distributions of SS$_4$ for all measured MoS$_2$ and WS$_2$ FETs, respectively. A median SS$_4$ of 431.9 mV.dec$^{-1}$ with a standard deviation of $\sigma_{SS} = 138.1$ mV.dec$^{-1}$ is obtained for MoS$_2$, and a median SS$_4$ of 541.4 mV.dec$^{-1}$ with a $\sigma_{SS} = 41.8$ mV.dec$^{-1}$ is obtained for WS$_2$. The median SS$_4$ values show slight increase from the corresponding median SS$_1$ values of 327.1 mV.dec$^{-1}$ and 438.2 mV.dec$^{-1}$ for MoS$_2$ and WS$_2$, respectively (Supplementary Table 2). However, no significant difference is found in the standard deviation values for SS$_1$ and SS$_4$. Note that the "champion" MoS$_2$ FET demonstrates SS$_1 = 93.3$ mV.dec$^{-1}$ and SS$_4 = 166$ mV.dec$^{-1}$, and the "champion" WS$_2$ FET demonstrates SS$_1 = 295.6$ mV.dec$^{-1}$ and SS$_4 = 452.8$ mV.dec$^{-1}$. The deviation of SS from its ideal value of 60 mV.dec$^{-1}$even for "champion" devices can be explained using Eq. (2).

$$SS = \frac{mk_B T}{q}\ln(10); m = \left(1 + \frac{C_S}{C_{OX}} + \frac{C_{IT}}{C_{OX}}\right); C_{IT} = qD_{IT} \quad (2)$$

Here, $k_B$ is the Boltzmann constant, $T$ is the temperature, $q$ is the electronic charge, $m$ is the body factor, $C_S$ is the semiconductor capacitance, $C_{IT}$ is the interface trap capacitance, $C_{OX}$ is the oxide capacitance, and $D_{IT}$ is the interface trap density. For fully depleted UTB FETs such as monolayer MoS$_2$ and WS$_2$ FETs, $C_S = 0$. In case of a clean oxide-semiconductor interface, $C_{IT} \ll C_{OX}$, ensuring that $m = 1$ and SS = 60 mV.dec$^{-1}$. Clearly, in our MoS$_2$ and WS$_2$ FETs, $m > 1$ indicates the presence of interface traps at the 2D/ dielectric interface (finite value of $C_{IT}$).

*Interface traps.* To evaluate the quality of the interface, we have extracted $D_{IT}$ using Eq. (2) and the corresponding distributions are shown in Fig. 4e, f for MoS$_2$ and WS$_2$ FETs, respectively. Median $D_{IT}$ of $6.2 \times 10^{12}$ eV$^{-1}$ cm$^{-2}$ and $8 \times 10^{12}$ eV$^{-1}$ cm$^{-2}$ were obtained for MoS$_2$ and WS$_2$, respectively. The device-to-device variation in $D_{IT}$ is shown in Supplementary Table 2. For fully depleted UTB Si MOSFETs with 35 nm thick Si and 110 nm gate length, SS = 80 mV.dec$^{-1}$ for an EOT = 4 nm, which corresponds to a $D_{IT} = 1.5 \times 10^{12}$ eV$^{-1}$ cm$^{-2}$[36]. Note that, while the $D_{IT}$ values for our monolayer 2D FETs are comparable with state- of-the-art Si FETs, thicker EOT = 22 nm results in smaller $C_{OX}$ and hence higher median values for the SS for MoS$_2$ and WS$_2$ FETs. For a fair comparison, we project the scaled-SS (SSS) for an EOT of 0.9 nm using the $D_{IT}$. We found SSS to be 76 mV.dec$^{-1}$ and 80 mV.dec$^{-1}$ for MoS$_2$ and WS$_2$, respectively, and 64 mV.dec$^{-1}$ for the UTB Si MOSFET in ref. [36]. A similar exercise was performed for other reports on MoS$_2$ FETs from the literature and the results are summarized in Table 2. The impact of higher $D_{IT}$ at the

TMD/Al$_2$O$_3$ interface can be mitigated either by scaling the EOT (i.e., increasing $C_{OX}$)[37] or by improving the interface (i.e., reducing $D_{IT}$). The presence of structural defects such as sulfur vacancies are known to introduce trap sites which contribute to $D_{IT}$. It has been found that $D_{IT}$ can be reduced by various surface passivation techniques[38,39]. In addition, photoresist residue from the litho- graphy and/or the wet transfer process can cause an increase in $D_{IT}$. Therefore it is possible to reduce $D_{IT}$ through further optimization of growth, post-growth processing, and improvement in fabrication process flow.

*Current ON/OFF ratio.* Fig. 4g, h show the distribution of the ratio of maximum to minimum current ($I_{max}/I_{min}$) across all MoS$_2$ and WS$_2$ FETs, respectively. Here, $I_{max}$ is the maximum current obtained from the transfer characteristics for $V_{DS} = 1$ V and $I_{min}$ is the average noise floor. Note that the true device current in the OFF-state is beyond the measurement range of the instrument. See Supplementary Fig. 4a, b for the distribution of $I_{max}$ and $I_{min}$, Supplementary Fig. 4c, d for the distribution of $I_{max}/I_{min}$ for different $L_{CH}$ for MoS$_2$ and WS$_2$ FETs, and Sup- plementary Table 3 for a summary of device-to-device variation in $I_{max}/I_{min}$. The median and standard deviation for $I_{max}/I_{min}$ were found to be $2.1 \times 10^7$ and $5.5 \times 10^7$ for MoS$_2$ FETs and $2.1 \times 10^7$ and $2.6 \times 10^7$ for WS$_2$ FETs. These values are over an order of magnitude higher than the $I_{max}/I_{min}$ of $1.3 \times 10^6$ for UTB Si MOSFETs[36]. $I_{max}/I_{min}$ is benchmarked against literature reports for $L_{CH} = 100$ nm as shown in Supplementary Table 4. Note that the key OFF-state performance indicators, i.e., threshold voltage, SS, $D_{IT}$, and $I_{max}/I_{min}$, are mostly found to be independent of $L_{CH}$. Even for $L_{CH} = 100$ nm, no detrimental short-channel effects are observed, which is expected and can be ascribed to the atomically thin body nature of monolayer TMDs, as well as the use of thin and high-k Al$_2$O$_3$ as the gate dielectric with EOT = 22 nm.

*Field-effect mobility and contact resistance.* Field-effect mobility ($\mu_{FE}$) is an important device parameter that strongly influences the ON-state performance of a FET. While intrinsic mobility is a material related parameter, $\mu_{FE}$ is determined by extrinsic effects, such as contact resistance ($R_c$), and often depends on how it is extracted from the device characteristics. Three popular methods for extracting $\mu_{FE}$ are peak transconductance ($\mu_{g_m}$), Y-function ($\mu_Y$)[40], and TLM ($\mu_{TLM}$) as described in Supplementary Note 3. Figure 5a, b, show the distribution and the corresponding median values for $\mu_{g_m}$ as a function of $L_{CH}$ for MoS$_2$ and WS$_2$ FETs, respectively. Additionally, 25th and 75th percentile values of the distribution are also marked. Clearly, $\mu_{g_m}$ shows a strong $L_{CH}$ dependence, with the median value varying from 23.9 cm$^2$ V$^{-1}$ s$^{-1}$ to 3.6 cm$^2$ V$^{-1}$ s$^{-1}$ for MoS$_2$ and 29 cm$^2$ V$^{-1}$ s$^{-1}$ to 2.7 cm$^2$ V$^{-1}$ s$^{-1}$ for WS$_2$ as the devices are scaled from $L_{CH} = 5$ µm down to $L_{CH} = 100$ nm. Supplementary Fig. 5a, b shows a similar analysis of $\mu_Y$ for MoS$_2$ and WS$_2$ FETs, respectively and

**Table 2 Benchmarking median subthreshold slope for $L_{CH}$ = 100 nm.**

|  | SS (mV.dec$^{-1}$) | EOT (nm) | Gate dielectric | $D_{IT}$ ($10^{12}$ eV$^{-1}$cm$^{-2}$) | SSS (mV.dec$^{-1}$) at SEOT = 0.9 nm |
|---|---|---|---|---|---|
| [7]—MoS$_2$ | 80 | 1.9 | 4 nm HfO$_2$ | $3.7 \times 10^{12}$ | 70 |
| [7]—MoS$_2$ | 160 | 2.7 | 8 nm HfO$_2$ | $1.3 \times 10^{13}$ | 93 |
| [7]—MoS$_2$ | 200 | 3.8 | 12 nm HfO$_2$ | $1.3 \times 10^{13}$ | 93 |
| [7]—MoS$_2$ | 1350 | 50 | 50 nm SiO$_2$ | $9.2 \times 10^{12}$ | 83 |
| Our work-MoS$_2$ | 450 | 22 | 50 nm Al$_2$O$_3$ | $6.3 \times 10^{12}$ | 76 |
| Our Work-WS$_2$ | 550 | 22 | 50 nm Al$_2$O$_3$ | $8 \times 10^{12}$ | 80 |
| [36]—UTB SOI | 80 | 4 | 4 nm SiO$_2$ | $1.8 \times 10^{12}$ | 64 |

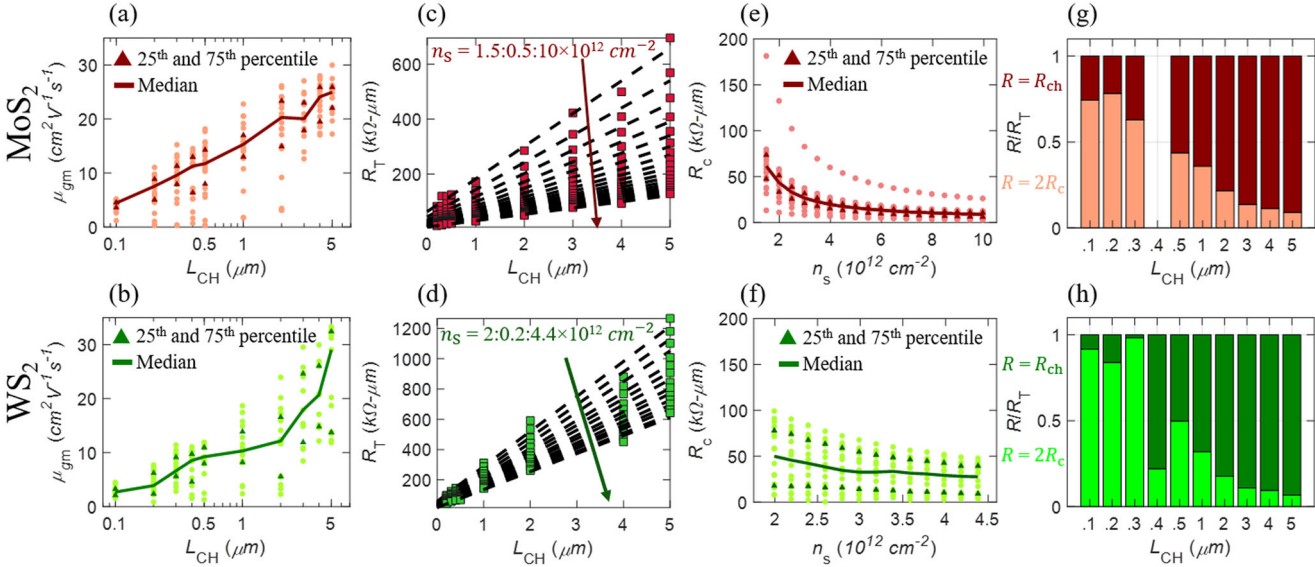

**Fig. 5 Device-to-device variation in field-effect mobility and contact resistance.** Distribution of mobility extracted using peak transconductance ($\mu_{g_m}$) for different channel lengths for **a** MoS$_2$ and **b** WS$_2$ FETs. Median, 25th percentile, and 75th percentile is also denoted. Total resistance ($R_T$) versus $L_{CH}$ for **c** MoS$_2$ and **d** WS$_2$ for different carrier concentrations ($n_S$) extracted using a representative TLM structure. The distribution of contact resistance ($R_c$) across multiple TLM structures, extracted from the y-intercepts in **c** and **d**, as a function of $n_S$ for **e** MoS$_2$ and **f** WS$_2$, respectively. The relative contribution of $R_c$ and channel resistance ($R_{ch}$) to the total resistance for **g** MoS$_2$ and **h** WS$_2$ for different $L_{CH}$. In scaled devices, as $R_{ch}$ scales with the channel length, the contribution of $R_c$ (note that $R_c$ is independent of $L_{CH}$), i.e., $2R_c/R_T$, is more significant compared to $R_{ch}$, i.e., $R_{ch}/R_T$.

Supplementary Table 5 summarizes the device-to-device variation in $\mu_{g_m}$ and $\mu_Y$. Both $\mu_{g_m}$ and $\mu_Y$ extracted from shorter-channel devices show significant reduction in their median values, indicating the dominant role of $R_c$ in scaled 2D FETs[6]. The contact resistance is seen as a result of Fermi-level pinning at the metal/TMD contact interface, resulting in a finite SB height[30]. To investigate further, we used the TLM structure shown in Fig. 2c to extract $R_c$ and evaluate its impact on $L_{CH}$ scaling as shown in Fig. 5c–f. We used Eq. (3) to extract $R_c$.

$$R_T = 2R_c + R_{ch}; R_{ch} = \frac{L_{CH}}{\mu_{TLM}C_{OX}(V_{GS} - V_{t_{lin}})} = \frac{L_{CH}}{qn_S\mu_{TLM}};$$
$$n_S = \frac{C_{OX}(V_{GS} - V_{t_{lin}})}{q} \quad (3)$$

Here, $R_T$ is the total measured resistance of the FET, and $R_{ch}$ is the channel resistance, which is directly proportional to $L_{CH}$ and inversely proportional to the carrier density ($n_S$) when the FET is measured in the linear operation regime. However, $R_c$ is independent of $L_{CH}$ and hence can be extracted from the y-intercept of $R_T$ versus $L_{CH}$ plots, as shown in Fig. 5c, d for MoS$_2$ and WS$_2$, respectively, for different $n_S$[41] (see Supplementary Note 4 for further discussion on the extraction of $n_S$). Figure 5e, f show the distribution of corresponding extracted $R_c$ as a function of $n_S$. A steady decrease in $R_c$ with increasing $n_S$ is attributed to the

phenomenon of contact-gating in global back-gated FET geometry, since the SB width at the metal/2D interface is modulated by the back-gate voltage[30]. Lower SB width allows for easier carrier tunneling, reducing $R_c$. For the MoS$_2$ FET, the median $R_c$ value was found to be 9.2 k$\Omega$−$\mu$m, corresponding to $n_S = 1 \times 10^{13}$ cm$^{-2}$. However, for WS$_2$, $n_S$ was limited to $4.4 \times 10^{12}$ cm$^{-2}$, owing to the more positive $V_{t_{lin}}$, resulting in a higher median $R_c = 29.2$ k$\Omega$−$\mu$m. For a case of identical carrier concentration, $n_S = 2.7 \times 10^{12}$ cm$^{-2}$, similar median $R_c$ values of 33 k$\Omega$−$\mu$m and 39.4 k$\Omega$−$\mu$m are obtained for MoS$_2$ and WS$_2$, respectively. The difference in $R_c$ between MoS$_2$ and WS$_2$ can be explained from the fact that the charge neutrality level is closer to the conduction band for MoS$_2$ than it is for WS$_2$, resulting in a higher SB height at the Ni/WS$_2$ contact interface compared to the Ni/MoS$_2$ contact interface[42].

The relative effect of $R_c$ is assessed for different $L_{CH}$. Figure 5g, h show the contribution of $R_c$ and $R_{ch}$ to the total resistance $R_T$ using stacked bar plots as a function of $L_{CH}$ for MoS$_2$ and WS$_2$, respectively. It is clear that for $L_{CH} \leq 1$ $\mu$m, the contact effects are significant since $2R_c > R_{ch}$. This explains why the extracted $\mu_{g_m}$ is $L_{CH}$ dependent and is severely underestimated by more than 80% for both MoS$_2$ and WS$_2$ when extracted from scaled devices with $L_{CH} = 100$ nm. Since $\mu_{g_m}$ extraction is limited by $R_c$, extracting $\mu_{TLM}$ following Eq. (3) is more appropriate for short channel devices. Supplementary Fig. 5c, d show the distribution of $\mu_{TLM}$

**Table 3 Benchmarking ON-state performance at $V_{DS} = 1\,V$ (best values are compared with median/mean values shown within parentheses).**

| | $\mu$(cm$^2$V$^{-1}$s$^{-1}$) | $R_c$(kΩ−μm) | $I_{ON}$(μA.μm$^{-1}$) | $n_S$(cm$^{-2}$) |
|---|---|---|---|---|
| [25]—MoS$_2$ | $\mu_{g_m} = 42$ (34.2) | 0.73 (1) | 22, $L_{CH} = 5.4$ μm | $1.3 \times 10^{13}$ |
| [22]—MoS$_2$ | $\mu_{TLM} = 20$ | 6.5 | 270, $L_{CH} = 80$ nm | $1 \times 10^{13}$ |
| [12]—MoS$_2$ | $\mu_{g_m} = 80$ ($\approx$40) | 2.4 | 13, $L_{CH} = 4$ μm | $6.6 \times 10^{12}$ |
| [65]—MoS$_2$ | $\mu_{TLM} = 30$ | 1.7 | 260, $L_{CH} = 10$ nm | $4.7 \times 10^{13}$ |
| [7]—MoS$_2$ | $\mu_{TLM} = 15$ | 1 | 250, $L_{CH} = 29$ nm | $1.5 \times 10^{13}$ |
| [26]—MoS$_2$ | $\mu_{4\text{-}point} \approx 75$ (70) | 14 | – | – |
| Our work-MoS$_2$ | $\mu_{TLM} = 47$ (27) | 3(9.2) | 73 (54), $L_{CH} = 100$ nm | $1 \times 10^{13}$ |
| [64]—WS$_2$ | $\mu_{g_m} = 11$ | – | 25, $L_{CH} = 4$ μm | $2.1 \times 10^{13}$ |
| [47]—WS$_2$ | $\mu_{g_m} = 20.4$ | – | 0.6, $L_{CH} = 1$ μm | $2.5 \times 10^{12}$ |
| [50]—WS$_2$ | $\mu_{g_m} = 5$ | – | $\approx$0.05, $L_{CH} = 10$ μm | $\approx 7.2 \times 10^{12}$ |
| [50]—WS$_2$ (Graphene contact) | $\mu_{g_m} = 50$ (27) | – | $\approx$1.1, $L_{CH} = 10$ μm | $\approx 7.2 \times 10^{12}$ |
| Our work-WS$_2$ | $\mu_{TLM} = 33$ (16) | 2.1 (29) | 26 (17), $L_{CH} = 100$ nm | $4.4 \times 10^{12}$ |
| [51]—UTB SOI | $\mu_{4\text{-}point} = 6$ | – | $\approx 35 \ast 10^{-3}$, $L_{CH} = 100$ μm | $\approx 9 \times 10^{12}$ |

across MoS$_2$ and WS$_2$ TLM structures, respectively, and Supplementary Table 6 summarizes the variation across the different TLM structures. The extracted median value for $\mu_{TLM}$ was found to be 27 cm$^2$ V$^{-1}$ s$^{-1}$ and 16 cm$^2$ V$^{-1}$ s$^{-1}$ for MoS$_2$ and WS$_2$ FETs, respectively. Long channel devices are less vulnerable to $R_c$ and corresponding $\mu_{g_m}$ values are more accurate representations of intrinsic channel mobility, albeit with some challenges as described by Nasr et al.[43] Nevertheless, our "champion" long-channel MoS$_2$ and WS$_2$ FETs with $L_{CH} = 5$ μm demonstrated $\mu_{g_m} = 30$ cm$^2$ V$^{-1}$ s$^{-1}$ and 33 cm$^2$ V$^{-1}$ s$^{-1}$, respectively. Similarly, "champion" MoS$_2$ and WS$_2$ TLM structures demonstrated $\mu_{TLM} = 46$ cm$^2$ V$^{-1}$ s$^{-1}$ and 33 cm$^2$ V$^{-1}$ s$^{-1}$, respectively.

Table 3 shows the benchmarking of our "champion" devices with the best reports from the literature using $\mu_{FE}$ ($\mu_{g_m}$ for longer channel devices and $\mu_{TLM}$ for shorter channel devices) and $R_c$ for both MoS$_2$ and WS$_2$. We have also included median/mean values wherever applicable. Note that while higher $\mu_{FE}$ values have been reported based on "champion" exfoliated and CVD grown MoS$_2$ FETs[7,12,24,25,44–46], our report is statistically more significant as it demonstrates variation across multiple TLM structures. For WS$_2$, $\mu_{FE} = 33$ cm$^2$ V$^{-1}$ s$^{-1}$ is the highest reported, 1.5X better than the previous report on synthetic WS$_2$[47]. Higher $\mu_{FE}$ values reported for WS$_2$ are either for exfoliated materials at room temperature[48] and low temperatures[49], or for CVD grown materials with contact engineering via the use of multilayer graphene as interlayers[50]. More interestingly, UTB Si MOSFETs with 0.9 nm thick Si show $\mu_{FE} \approx 6$ cm$^2$ V$^{-1}$ s$^{-1}$[51], which is more than 2 orders of magnitude smaller compared to bulk Si mobility and is primarily attributed to thickness fluctuation in UTB Si.

Metal/2D contact resistances are comparatively high even for the "champion" devices with $R_c = 3$ kΩ−μm and $R_c = 2.1$ kΩ−μm for MoS$_2$ and WS$_2$, respectively, when compared to the $R_c = 0.1$ kΩ−μm typically reported for state-of-the-art Si FETs. However, various methods have been developed to reduce the effect of SB-limited carrier transport in 2D TMDs[52], such as work function engineering to reduce the SB height[30], introduction of interlayers such as graphene to decouple the metal/2D interface to alleviate Fermi-level pinning[53,54], and achieving higher carrier concentration underneath or near the metal/2D contacts through substitutional or surface charge transfer doping to reduce the SB width[42,55]. Nevertheless, our MOCVD grown monolayer MoS$_2$ FETs demonstrate $R_c$ similar to values reported in the literature[7,22,25,56]. The "champion" devices are

benchmarked in Table 3. To the best of our knowledge, this is the first report of $R_c$ for synthetic WS$_2$. Additionally, our work marks the first study on the extraction of contact resistance from multiple TLM structures for both MoS$_2$ and WS$_2$. Smithe et al.[25] have demonstrated a pseudo-TLM analysis where independent devices with different channel lengths and widths were used to extract the distribution of $R_T$. TLM analysis is done on the devices between 10th and 90th percentile[25]. Our demonstration involves the extraction of contact resistance from separate TLM structures and finding the variation across these TLM structures, and the analysis is not limited to a percentile limit.

*Drive-current and saturation velocity.* Finally, high performance FETs are benchmarked using the drive current ($I_{ON}$) that is achievable for a given supply voltage ($V_{DS} = V_{DD}$). Higher values of $I_{ON}$ ensure faster circuit operation as the intrinsic delay of a FET is proportional to $CV_{DD}/I_{ON}$, where $C$ is the load capacitance. In digital electronics, higher $I_{ON}$ allows larger fan-out. Figure 6a, b display the output characteristics of MoS$_2$ and WS$_2$ FETs, respectively, for different channel lengths, which were used to assess the ON-state performance of the devices. At high biases, high current density leads to self-heating, resulting in negative differential resistance (NDR) behavior. This is a common phenomenon seen in ultra-thin body FETs, including SOI FETs[57], nanowire FETs[58], graphene FETs[59], and, more recently, exfoliated multilayer MoS$_2$ FETs[60] and CVD grown monolayer MoS$_2$ FETs[61]. It is possible to reduce or eliminate the self-heating effect through pulsed measurements with pulse widths less than 100 μs[60].

Figure 7a–d show the median for $I_{ON}$ as a function of $L_{CH}$ for $V_{DS} = 1$ V and $V_{DS} = 5$ V for MoS$_2$ and WS$_2$ FETs, respectively, extracted from their respective output characteristics. For both TMDs, at low $V_{DS} = 1$ V, i.e., in the linear region, $I_{ON}$ is expected to demonstrate an inverse channel length dependence following Eq. (4).

$$\frac{I_{ON}}{W} = \frac{I_{DS,LIN}}{W} = q n_S \mu_{g_m} \frac{V_{DS}}{L_{CH}} \qquad (4)$$

This trend is observed for both MoS$_2$ and WS$_2$ FETs in Fig. 7a, b, respectively, for channel lengths $L_{CH} \geq 1$ μm. However, for devices with channel length $L_{CH} < 1$ μm, the inverse channel length dependence is obscured by $R_c$. Similar linear dependence is observed for $I_{ON}$ in longer-channel devices ($L_{CH} \geq 1$ μm) at

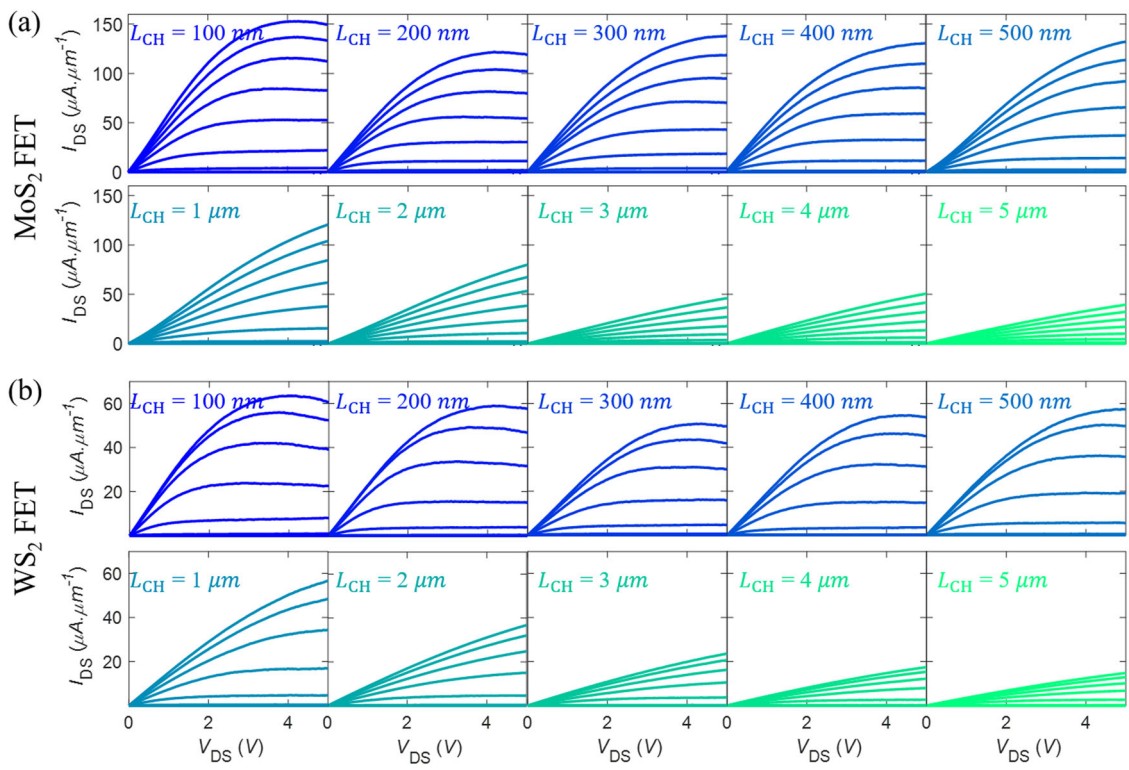

**Fig. 6 ON-state performance of monolayer MoS$_2$ and WS$_2$ FETs.** Output characteristics for channel lengths ranging from $L_{CH} = 100$ nm to $L_{CH} = 5$ μm, obtained from a representative transmission-line measurement structure for **a** MoS$_2$ and **b** WS$_2$ FETs. Current saturation is achieved in shorter-channel devices for both MoS$_2$ and WS$_2$ FETs.

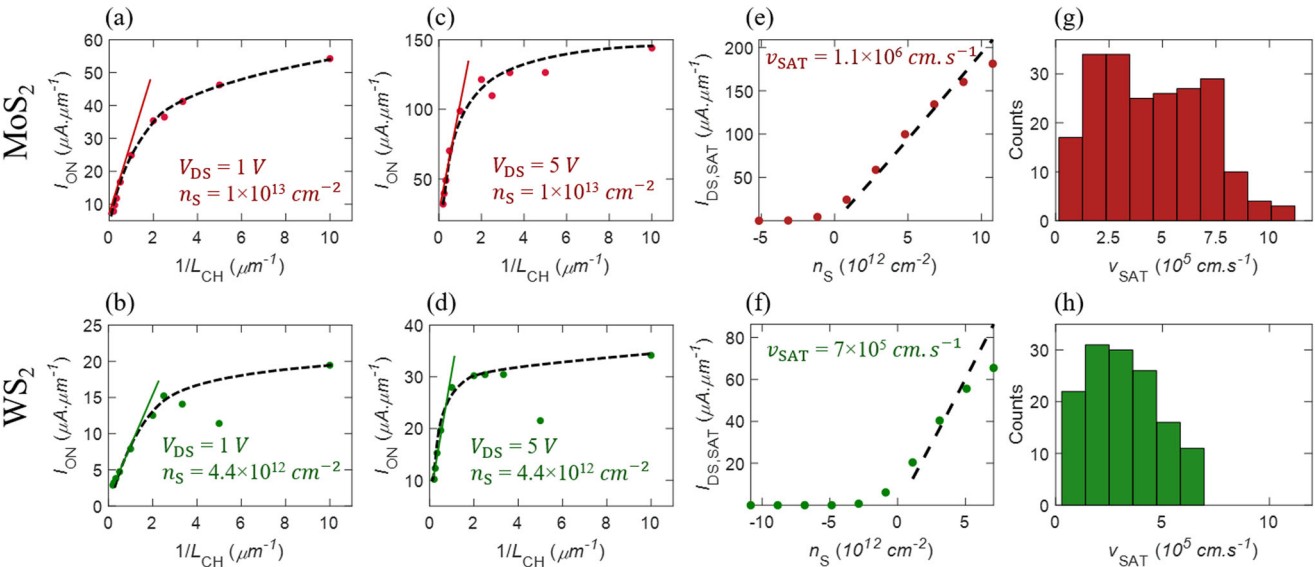

**Fig. 7 Drive current and saturation velocity.** Median of drive current ($I_{ON}$) as a function of $1/L_{CH}$ for **a** MoS$_2$ and **b** WS$_2$ FETs at $V_{DS}$ of 1 V corresponding to the linear region of the FETs, and **c** MoS$_2$ and **d** WS$_2$ FETs at $V_{DS}$ of 5 V corresponding to the saturation region of the FETs. These are extracted at carrier concentrations ($n_S$) of $1 \times 10^{13}$ cm$^{-2}$ and $4.4 \times 10^{12}$ cm$^{-2}$ for MoS$_2$ and WS$_2$, respectively. The saturation current ($I_{DS,SAT}$) for **e** MoS$_2$ and **f** WS$_2$, extracted from their corresponding shorter-channel devices ($L_{CH} < 1$ μm) as a function of $n_S$. The slope indicates the saturation velocity ($v_{SAT}$). The distribution of $v_{SAT}$ for **g** MoS$_2$ and **h** WS$_2$ shorter-channel FETs.

$V_{DS} = 5$ V for both MoS$_2$ and WS$_2$ FETs in Fig. 7c, d, respectively, following Eq. (5).

$$\frac{I_{ON}}{W} = \frac{I_{DS,SAT}}{W} = \frac{C_{OX}\mu_{g_m}(V_{GS} - V_{t_{lin}})^2}{2L_{CH}} = \frac{q^2\mu_{g_m}}{C_{OX}}\frac{n_S^2}{2L_{CH}} \quad (5)$$

These results are in accordance with classic long-channel FET characteristics (i.e., at low drain bias, the device operates in the

linear regime (Eq. (4)), whereas for $V_{DS} \geq V_{GS} - V_{t_{lin}}$, the channel is pinched-off, resulting in current saturation). The saturation current follows a square-law dependence on the overdrive voltage and, therefore, on $n_S$ (Eq. (6))[62]. In shorter-channel devices ($L_{CH} < 1$ μm), as the lateral electric field ($\xi \approx \frac{V_{DS}}{L_{CH}}$) becomes more than the critical electric field ($E_C$), the carrier velocity reaches

saturation velocity ($v_{\text{SAT}}$). This leads to current saturation, with the saturation current being independent of $L_{\text{CH}}$ as described by Eq. (6)[62].

$$\frac{I_{\text{ON}}}{W} = \frac{I_{\text{DS,SAT}}}{W} = C_{\text{OX}} v_{\text{SAT}} \left( V_{\text{GS}} - V_{t_{\text{lin}}} \right) = q n_{\text{S}} v_{\text{SAT}} \qquad (6)$$

However, in order to observe current saturation due to velocity saturation, the drain bias must meet the criterion given by Eq. (7).

$$L_{\text{CH}} \frac{v_{\text{SAT}}}{\mu_{g_{\text{m}}}} < V_{\text{DS}} < \left( V_{\text{GS}} - V_{t_{\text{lin}}} = \frac{q n_{\text{S}}}{C_{\text{OX}}} \right) \qquad (7)$$

For example, as seen in Fig. 6, current saturation is achieved at $V_{\text{DS}} = 4$ V for 100 nm $MoS_2$ FET and $WS_2$ FET, which is much lower than the corresponding $V_{\text{GS}} - V_{t_{\text{lin}}}$ of 11.6 V and 6.7 V, respectively. This explains why the drive current in shorter-channel $MoS_2$ and $WS_2$ FETs display little-to-no channel length dependence for high drain bias ($V_{\text{DS}} = 5$ V), as seen in Fig. 7c, d. Nevertheless, scaled $MoS_2$ and $WS_2$ FETs with channel lengths of 100 nm demonstrate high median drive currents of $I_{\text{ON}} = 54$ μA. $μm^{-1}$ and $I_{\text{ON}} = 17$ μA.$μm^{-1}$, respectively, for $V_{\text{DS}} = 1$ V and $I_{\text{ON}} = 146$ μA.$μm^{-1}$ and $I_{\text{ON}} = 30$ μA.$μm^{-1}$, respectively, for $V_{\text{DS}} = 5$ V. Furthermore, $I_{\text{ON}}$ at $V_{\text{DS}} = 5$ V can reach as high as 161 μA.$μm^{-1}$ and 53 μA.$μm^{-1}$ in "champion" $MoS_2$ and $WS_2$ FETs, respectively. The distribution of $I_{\text{ON}}$ for $V_{\text{DS}} = 1$ V and $V_{\text{DS}} = 5$ V as a function of $L_{\text{CH}}$ is shown in Supplementary Fig. 6 and the corresponding device-to-device variation is summarized in Supplementary Table 7 for $MoS_2$ and $WS_2$ FETs. The higher drive current seen for $MoS_2$ FETs compared to that of $WS_2$ FETs is a direct consequence of lower $V_{t_{\text{lin}}}$, which allows for higher $n_{\text{S}}$ in $MoS_2$ channels. Further improvement in the drive current of scaled 2D FETs can be achieved by reducing $R_c$. Note that, while there are reports of higher $I_{\text{ON}}$ in large-area grown $MoS_2$ films, none of the earlier studies provide extensive device statistics[22,63–65]. $I_{\text{ON}}$ for UTB Si MOSFET is 35 nA.$μm^{-1}$ for 0.9 nm thick Si[51]. The "champion" devices are benchmarked in Table 3. Supplementary Table 8 shows benchmarking of our statistical study on $MoS_2$ FETs using field-effect mobility and drive current (at $V_{\text{DS}} = 2$ V) with similar channel length dependent statistical studies from the literature. The mean and standard deviation is compared with the $L_{\text{CH}}$ dependence and plotted in Supplementary Fig. 7. Better performance is seen for our channel length dependence study compared to ref. [66] for both the drive current and mobility.

Finally, saturation velocity ($v_{\text{SAT}}$) is another key material parameter that determines $I_{\text{ON}}$ in scaled FETs. This is because at low lateral electric field ($\xi$) the average electron drift velocity increases linearly through the mobility ($v_d = \mu_{\text{FE}} \xi$), but at large electric fields, which are easily achievable in sub-micron FETs, the carrier velocity saturates. Thus, $I_{\text{ON}}$ becomes less dependent on $\mu_{\text{FE}}$ and is instead proportional to $v_{\text{SAT}}$, following Eq. (6). Additionally, high $v_{\text{SAT}}$ is needed for faster switching[11]. Figure 7e–h show the extraction of $v_{\text{SAT}}$ and the distribution of $v_{\text{SAT}}$ for $MoS_2$ and $WS_2$, respectively. The linear dependence of the saturation current ($I_{\text{DS,SAT}}$) on $n_{\text{S}}$ following Eq. (6), is used to extract $v_{\text{SAT}}$. Median $v_{\text{SAT}}$ values of $6.4 \times 10^5$ cm.$s^{-1}$ and $4 \times 10^5$ cm.$s^{-1}$ and "champion" $v_{\text{SAT}}$ values of $1.1 \times 10^6$ cm.$s^{-1}$ and $6.9 \times 10^5$ cm.$s^{-1}$ are obtained for $MoS_2$ and $WS_2$, respectively. The corresponding device-to-device variations are summarized in Supplementary Table 9. The $v_{\text{SAT}}$ values are significantly lower compared to bulk Si with $v_{\text{SAT}} \approx 10^7$ cm.$s^{-1}$[67,68]. Nathawat et al. have reported higher $v_{\text{SAT}} \approx 6 \times 10^6$ cm.$s^{-1}$ in CVD grown monolayer $MoS_2$[69]. However, their measurements were done using nanosecond range pulses to reduce the impact of self-heating and hot carrier capture by deep oxide traps. For $WS_2$, this is the first report of $v_{\text{SAT}}$.

## Discussion

In conclusion, we have performed a detailed study of device-to-device variation and impact of channel length scaling on the electrical parameters, such as threshold voltage, subthreshold slope, density of interface trap states, ratio of minimum to maximum current, field-effect electron mobility, drive current, contact resistance, and saturation velocity, of MOCVD grown $MoS_2$ and $WS_2$ monolayer based FETs using statistical measures such as median, mean, standard deviation, and minimum/maximum values and have benchmarked our findings against other similar reports from 2D literature as well as UTB Si FETs. While in absolute terms the spatial variations in the respective benchmarking parameters appear to be large for $MoS_2$ and $WS_2$ FETs, when compared at scaled oxide thickness, our results are not significantly different from the projected variations for UTB Si FETs. Our "champion" long-channel $MoS_2$ and $WS_2$ FETs with $L_{\text{CH}} = 5$ μm demonstrated electron mobilities of 30 $cm^2$ $V^{-1}$ $s^{-1}$ and 33 $cm^2$ $V^{-1}$ $s^{-1}$, respectively, when extracted using peak transconductance and 46 $cm^2$ $V^{-1}$ $s^{-1}$ and 33 $cm^2$ $V^{-1}$ $s^{-1}$, respectively, when extracted using TLM method. For synthetic monolayer $WS_2$ films, these are the highest reported room temperature electron mobilities, 1.5X better than the best report from the literature. Similarly, our "champion" shortest channel length $MoS_2$ and $WS_2$ FETs, with $L_{\text{CH}} = 100$ nm, demonstrated drive currents as high as 161 μA.$μm^{-1}$ and 53 μA.$μm^{-1}$ for $V_{\text{DS}} = 5$ V at carrier densities of $n_{\text{S}} = 1 \times 10^{13}$ $cm^{-2}$ and $4.4 \times 10^{12}$ $cm^{-2}$, respectively, in spite of the presence of high contact resistances. We attribute our accomplishments to the epitaxial growth of highly crystalline 2D monolayers on sapphire substrate via MOCVD at 1000 °C using chalcogen and sulfur precursors that minimize carbon contamination in the film, as well as to the clean transfer of the film from the growth substrate to the device fabrication substrate. Our findings suggest that 2D FETs are promising alternatives for future VLSI circuits.

## Methods

**MOCVD growth**. Uniform monolayer deposition was achieved in a cold-wall horizontal reactor with an inductively heated graphite susceptor equipped with wafer rotation as previously described[27]. Molybdenum hexacarbonyl (Mo(CO)$_6$) and tungsten hexacarbonyl (W(CO)$_6$) were used as metal precursors while hydrogen sulfide (H$_2$S) was the chalcogen source with H$_2$ as the carrier gas. Mo(CO)$_6$ maintained at 10 °C and 950 Torr in a stainless-steel bubbler was used to deliver 0.036 *sccm*. W(CO)$_6$ maintained in a bubbler at 10 °C and 760 Torr delivered $6.4 \times 10^{-4}$ sccm. The flow rate of H$_2$S was 400 sccm and the reactor pressure was 50 Torr for both sulfides. $MoS_2$ was deposited in a single step process at 1000 °C where coalesced monolayer growth across the 2″ wafer was achieved in 18 min. $WS_2$ was deposited using a multi-step process with nucleation at 850 °C and lateral growth at 1000 °C, which resulted in coalesced monolayer growth across the 2″ wafer in 10 min[28]. In both cases, after growth, the substrate was cooled in H$_2$S to 300 °C to inhibit decomposition of the $MoS_2$ and $WS_2$ films.

**Material characterization**. A Bruker Icon atomic force microscope was used to measure surface morphology and film thickness. Scanasyst AFM tips with a nominal tip radius of ≈2 nm and spring constant of 0.4 N$m^{-1}$ were used in the peak-force tapping mode for the measurements. Photoluminescence (PL) maps were acquired over a $5 \times 5$ μ$m^2$ area with a laser wavelength of 532 nm and 300 grooves per mm grating in a WITec apyron Confocal Raman Microscope. A PANalytical MRD diffractometer with a 5-axis cradle was used for in-plane X-ray diffraction characterization of the sulfide films[70]. A Cu anode X-ray tube operated at 40 kV accelerating voltage and 45 mA filament current was used as the X-ray source. On the primary beam side, a mirror with ¼° slit and Ni filter were used to filter the Cu Kα line. On the diffracted beam side, an 0.27° parallel plate collimator with 0.04 rad Soller slits with PIXcell detector in open detector mode were employed. To determine the in-plane epitaxial relation of the film with respect to a substrate, sample surface was ≈2–4° away from the X-ray incidence plane.

**Transfer of monolayer films**. Both the $MoS_2$ and $WS_2$ films were grown on 2″ sapphire wafers. The 2″ sapphire wafers were then cut into $1 \times 1$ $cm^2$ pieces. For each material, two (2) $1 \times 1$ $cm^2$ sapphire substrates were chosen, one corresponding to the center and another one corresponding to the edge of the 2-inch wafer. To fabricate the FETs, monolayer $MoS_2$ and $WS_2$ films grown on sapphire

substrates were transferred onto $1 \times 1$ cm$^2$ device fabrication substrates, i.e., 50 nm Al$_2$O$_3$ on Pt/TiN/p$^{++}$-Si, using a PMMA (polymethyl-methacrylate) -assisted wet transfer process. First, the sapphire substrate with the monolayer film was spin coated with PMMA and then baked at 180 °C for 90 s. The corners of the spin coated films were scratched using a razor blade and immersed inside a 1 M NaOH solution kept at 90 °C. Capillary action drew NaOH into the substrate/film interface, separating the PMMA/monolayer film stack from the sapphire substrate. The separated film was then rinsed multiple times inside a water bath and finally transferred onto the 50 nm alumina substrate and baked at 50 °C and 70 °C for 10 min each to remove moisture and residual PMMA, ensuring a pristine interface.

**Gate dielectric fabrication.** Direct replacement of thermally oxidized SiO$_2$ with a high-κ dielectric such as Al$_2$O$_3$ grown *via* atomic layer deposition (ALD) is a logical choice to scale the EOT. However, we found that a Al$_2$O$_3$/p$^{++}$-Si interface is not ideal for back gated FET fabrication owing to higher gate leakage current, more interface trap states, and large hysteresis, all of which negatively impact the performance of the device. Replacing Si with Pt, a large work function metal (5.6 eV) allows for minimal hysteresis and trap state effects[71]. Since Pt readily forms a Pt silicide at temperatures as low as 300 °C, a 20 nm TiN diffusion barrier deposited by reactive sputtering was placed between the p$^{++}$ Si and the Pt, permitting subsequent high temperature processing[72]. This conductive TiN diffusion barrier allows the back-gate voltage to be applied to the substrate, thus simplifying the fabrication and measurement procedures. The polycrystalline Pt introduces very little surface roughness to the final Al$_2$O$_3$ surface, with a rms roughness of 0.7 nm.

**Device fabrication.** Back gated field-effect transistors (FET) are fabricated using e-beam lithography. To define the channel region the substrate is spin coated with PMMA and baked at 180 °C for 90 s. The photoresist is then exposed to e-beam and developed using 1:1 mixture of 4-methyl-2-pentanone (MIBK) and 2 propanol (IPA). The monolayer MoS$_2$ film is subsequently etched using sulfur hexafluoride (SF$_6$) at 5 °C for 30 s. Next the sample is rinsed in acetone and IPA to remove the photoresist. In order to fabricate the source/drain contacts the substrate is again spin coated with MMA and PMMA followed by the e-beam lithography, developed using MIBK and IPA, and e-beam evaporation of 40 nm Ni/30 nm Au stack. Finally, the photoresist is rinsed away by lift off process using acetone and IPA.

**Electrical characterization.** Lake Shore CRX-VF probe station and Keysight B1500A parameter analyzer were used to perform the electrical characterization at room temperature in high vacuum (≈10$^{-6}$ Torr). Standard DC sweeps are used for the measurements of transfer and output characteristics of all devices. To ensure that the FETs are stabilized, they are conditioned by multiple repetitions of the same measurement. The transfer characteristics are measured three times to condition each FET and the fourth measurement is used for the analysis. The output characteristics are measured twice following the transfer characteristics and the second measurement is used for the analysis. We have found that no burn-in procedure is needed to ensure proper contact formation. Both MoS$_2$ and WS$_2$ FETs were measured as-fabricated.

**Reporting summary.** Further information on research design is available in the Nature Research Reporting Summary linked to this article.

## Data availability
The datasets generated during and/or analyzed during the current study are available from the corresponding authors on reasonable request.

## Code availability
The codes used for plotting the data are available from the corresponding authors on reasonable request.

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

## Acknowledgements

The work was partially supported by Army Research Office (ARO) through Contract Number W911NF1920338. Authors also acknowledge the support from the National Science Foundation (NSF) through the Pennsylvania State University 2D Crystal Consortium–Materials Innovation Platform (2DCCMIP) under NSF cooperative agreement DMR-1539916.

## Author contributions

S.D. conceived the idea. S.D. and A.S. designed the experiments and wrote the manuscript. A.S. and R.P. performed the measurements, S.D., A.S., and R.P. analyzed the data, discussed the results, and agreed on their implications. T.H.C, and J.M.R synthesized and characterized MoS₂ and WS₂ monolayers. All authors contributed to the preparation of the manuscript.

## Competing interests

The authors declare no competing interests.
