## [Peer Review File · Nature Communications]

REVIEWER COMMENTS

Reviewer #1 (Remarks to the Author):

Very good review with very useful data for the community.

Reviewer #2 (Remarks to the Author):

The authors measure back-gated transistor characteristics of synthesized and transferred MoS₂ and WS₂ on Al₂O₃ on Si. The paper is significant in that it measures the properties of 230 and 130 FETs respectively and reports the distributions of the characteristics vs. gate length. There is little data like this and the results show the current state-of-the-art of synthesized channel devices in a university process. The analysis of the data however (on one wafer of MoS₂ and one wafer of WS₂) is uncritical and does not compare with Si MOSFETs even as it purports to assess technology readiness level. Despite what appears to be enormous variation relative to Si MOSFETs, the authors conclude in the end that "Finally, low device-to-device variation, and high performance seen for scaled MoS₂ and WS₂ FETs exhibit high technology readiness level." This seems at odds with the consistent and high variation they observe vs. Si CMOS, not to mention that they only obtain n-channel FETs in this study as there is no channel conductivity control. The data in this study is interesting however the analysis lacks a serious consideration and critical comparison with Si. It appears that the TRL is still at level 1 or 2 and not "high." Below are comments and questions.

1. Fig 1b suggests 2 inch wafer process: Fig. 2a suggests smaller substrates What area is patterned with devices
2. Fig 1 caption. "ow variation" should be "Low variation"
3. Fig 1 caption, the statement "Intense PL is a characteristic of monolayer film owing to indirect to direct bandgap transition." There is no measurement of intensity given so this statement seems strange. One would expect direct transitions to be stronger than indirect transitions so what is the measure of "intense PL" and why does this indicate a single monolayer film?
4. Fig. 2. Does the current scale with W/L? If not, please explain.
5. Fig. 2. Why are these devices stated to be contact limited? The 100 nm channel length transistors do not show "contact dominated transport" in the output characteristics in that the drain current in saturation does not increase linearly with V_{DS}.
6. What testing protocol in the I-V measurements is utilized. Often 2D materials drift and change under repeated testing and measurement conditions need to be stabilized to get repeatability. What slew rate and slew direction are used for the I-V measurements to insure repeatability and absence of spurious charging? What burn-in procedure is utilized to ensure contacts are formed and stable? What is the magnitude of hysteresis measured in double sweeps? When negative differential resistance (NDR) and conductance fluctuations are observed, are these repeatable.
7. Fig. 2(i) and (j) What is the origin of the NDR? Is it stable on repetitive sweeps and under different sweep rates?
8. The terms long and short channel are applied to 5 and 0.1 μm FETs. It is not clear that 0.1 μm is short enough to be considered a "short channel" FET in this ultrathin channel geometry. The manifestation of short channel effect is that the drain current depends on $(1 + \lambda V_{DS})$. It appears both of the "short channel" MoS₂ and WS₂ transistors are long channel devices in that a linear dependence of I_D on V_{DS} is not observed. The discussion of the transistors as short and long channel appears inappropriate relative to the usual meaning of these terms.

9. Eq (1) Since mobility often depends on VDS in 2D materials, more discussion is needed to clarify how mobility is reported. Is this the peak mobility with VDS? If so, does it satisfy the condition that VDS is sufficiently low to justify use of Eq. 1? In this part of the paper, the authors appear to be just giving one number instead of the median value and standard deviation. This is inconsistent.

10. Define "champion" devices? I assume this means the best transistors by some measure, but what criteria were applied in selecting these devices? Was it the best device for a single measure or was it best assuming a trade-off of measures?

11. page 7. Please clarify how these carrier densities are computed. It seems it is from the oxide capacitance, but it is not clear whether fringe and interface trap capacitance is included. The interface trap capacitance is later shown to be large with respect to Cox.

12. There does not appear to be data to substantiate that these transistors have monolayer vs. multilayer channels. AFM data of Fig. 1c,d, does not establish a ML film as no step heights are shown and the triangle pointed to says bilayer. There seems to be no way to tell the thickness of the film under the triangle. The PL measurements in 1i,j can be calibrated by other measurements to indicate the thickness, but needs verification by another physical measurement or substantiation by a model to be used to get a thickness. Please clarify how these films were confirmed to be single monolayer.

13. It appears that the results of this paper are based on two growths which are shown on 50 mm wafers in Fig. 1. Then some fraction of these films (1x1 cm²?) were transferred to a Si substrate. Please clarify how many growths are included in this study so there is no ambiguity. Or are the growths on cm² pieces?

14. All figures could benefit from a doubling of font size. They are not easy to read even on a large screen. On Fig. 3 which shows 20 plots, the x and y-axes are repeated 20 times. The figures could be enlarged if the redundant labeling was removed. This approach could be used to advantage in almost every figure.

15. Benchmarking typically refers to checking something by comparison to some standard. This paper appears to be characterizing the variation in device properties across two wafers, one MoS₂ and one WS₂. There is no standard or basis for comparison applied except to find median values and distributions on these two films. It seems relevant to compare to Si nMOSFETs if the aim is to assess TRL.

16. There does not appear to be any significant statistical analysis applied to this data. I am not sure what the special meaning of "statistical benchmarking"; it seems you are just plotting multiple device characteristics or extracted parameters on the same scale. What total area is mapped in this study?

17. Understanding the spatial variation could be important to shed light on the origin of the variation. The origin of the variation is not dealt with relative to what it needs to be. The factors appear to be variation in the film crystallinity, carrier density variations, or thickness uniformity. I would expect the film coalesces into a polycrystalline layer. What is known about this? This should be clarified and some attempt should be made to assess the physical causes of the variability.

18. A more accurate title might be "Variation in synthetic, layer-transferred MoS₂ and WS₂ field-effect transistors" unless some benchmarking to Si MOSFETs can be added. There seems no more statistical analysis than are typically applied to measurements.

19. Terms like ALD, FETs, MOCVD, SEM, TLM, TMD, TRL etc. and parameter symbols are defined multiple times in the paper. These need only be defined once.

20. The authors state "A fair assessment of the technology readiness level (TRL) necessitates large device statistics to understand the variation in the FET performance across the entire chip, as well as thorough study of the impact of channel length scaling on the FET performance of the

technology. This was accomplished ..." This idea that TRL needs to be assessed is raised by the authors and then it is said to be accomplished, however the authors do not seriously address this question. What is the TRL of MoS₂ and WS₂ FETs? What are the measures? Based on the authors findings, it is not clear there is enough data to support TRL assessment. How can a study of 2 wafers using a transfer process provide data to assess TRL?

21. Fig. 4abcd It is not clear why the authors choose to plot the threshold voltage and SS using this variety of definitions. Why is it relevant that SS improves when compared over decades. The 3D plots do not allow the methods to be closely compared and the distribution widths seem to contain the most relevant information. What conclusions can be drawn from this comparison across these different methodologies? It seems relevant to make comparison against published measures for CMOS at some gate length.

22. The figure captions are so long that it is not easy to scan them for the relevant information about what is plotted. The captions often discuss the data and repeat information already given in the text. The captions would be much improved if they were more compact and the discussion in the captions consolidated into the text where the repeats are eliminated.

23. As an example Fig. 4 caption states "None of the key performance indicators related to the OFF-state, i.e. threshold voltage, SS...show any noticeable and detrimental short-channel effects, which can be ascribed to the ultra-thin body nature of monolayer TMDs, as well as the use of high-k Al₂O₃ as the gate dielectric." This discussion belongs in the text. The smallest gate lengths tested here are not short channel devices so it is not surprising that they do not show short-channel effects. They would not be short channel device even if the TMD thickness were thicker and this second argument about Al₂O₃ is similarly not well explained. Quantifying some of these assessments would be more valuable than providing these general assertions.

24. Fig. 4 caption "Furthermore, low device-to-device variation in these parameters (Table 1) imply that high quality and uniform growth was achieved by our MOCVD technique." Move this to the discussion of Table 1 and quantify the statements. What is "low?" What is the measure of "high quality?" What is the measure of "uniform." How do these compare with Si?

25. page 10. The authors state the median values for V_t across different techniques and then say "These results establish low device-to-device variation across the substrate, which can be attributed to uniform and contaminant-free MOCVD growth of monolayer TMDs and clean device fabrication process." The distributions extend over volts which shows a technology with variation too large for VLSI use, yet the authors develop a conclusion the films are uniform and contaminant-free. The results should be quantified and a serious comparison made.

26. The discussion of SS is similarly cursory. Considering SS₄, which is a minimum standard for MOSFETs, the median values are 431.9 and 541.4 mV (I don't know why the authors think they have 4 significant digits of accuracy in these measurements) indicating a large interface trap density, yet this is ignored and they say this can be "mitigated by scaling the oxide thickness." which does not solve the inherent trap problem. I see no particular insights drawn by computing SS₁ SS₂ or SS₃. This is compared in the supplementary material and that is probably where it belongs. Obviously the SS will be monotonically lower if you focus on smaller voltage intervals. The variation in SS is another strong indicator of nonuniformity and this is not analyzed. Why is the interface trap density not computed or assessed?

27. Fig. 4ef, Certainly DIBL can be computed given the definition, but what measurements indicate that DIBL is due to the drain field reaching through to the source? One place to look for this is in the common source characteristics and these show more dI_D/dV_{DS} in the 5 μm devices than the 0.1 μm devices. This seems to argue against the simple explanation of the authors.

28. Why is eq (1) used to analyze the mobility vs. eq. (2)? Why should the square law model be favored over the Y-function model for this analysis. Are the two models in agreement? The U_y is given in the supplemental data. What is the reason for the gate length dependence? It is said to be due to "contact effects", but what are these contact effects?

29. The analysis of resistance ignores the interface trap density which appears to be large with respect to the channel sheet density. It seems the plots in Fig 5ij of RC vs. ns have little meaning. The most important aspect is the contact resistance variation which as with all the parameters is large with respect to use in applications.

30. I find the remaining discussion of the data in Figs 6 and 7 and the tables to be simple and uncritical. Typically a few statements giving some values from the figures and then sweeping uncritical statements related to long and short channel effects which do not appear to be manifest from the measurements.

31. The final statement "Finally, low device-to-device variation, and high performance seen for scaled MoS2 and WS2 FETs exhibit high technology readiness level." appears to be at odds with the measured data.

Reviewer #3 (Remarks to the Author):

The paper is high quality, well written and in clear English. It provides a complete characterization of MoS2 and WS2 FETs. The considered metrics are well chosen (drive current, threshold voltage, field-effect carrier mobility, carrier saturation velocity, contact resistance, subthreshold slope, current ON/OFF ratio, and drain-induced barrier lowering). The MX2 literature needs additional variability studies like this one. Here are my comments:

1. (crucial) a record mobility for WS2 of 33 cm²/Vs is quoted in abstract and text, but several higher values are found in literature. Can you specify if you mean only for monolayer and/or only synthetic material?

-this paper shows 83 <https://onlinelibrary.wiley.com/doi/abs/10.1002/adma.201502222>

-this paper shows 50 <https://onlinelibrary.wiley.com/doi/abs/10.1002/adfm.201703448>

-this paper shows 140 <https://pubs.acs.org/doi/10.1021/nn502362b>

2. (crucial) You mention "However, one must admit that a genuine assessment of the technology readiness level (TRL) for 2D materials should include a comprehensive benchmarking and scaling study of all relevant transport parameters using statistically significant number of devices."

I think the state of the art and goal of the paper is not sufficiently well introduced. I recommend mentioning the following papers on variability

- [41] (your paper overlaps strongest with this one)

- [7]

- "H. Xu et al., Small, 14, 48, p.1803465 (2018)"

- [5]

Can you mention these papers in the introduction, what they lack, and what is innovative in your paper?

3. I think "substrate agnostic growth" is incorrect. The substrate is crucial, as you later state, "Fig. 1e and 1f highlight the epitaxial relation between the sulfide monolayers and the underlying sapphire substrates"

4. In our team, when we measure an uncapped TMD channel device twice, the second curve has a strong positive V_t shift (1V shift for 4V sweep) because traps have been pre-charged. In a TLM set, all devices are connected together and share the same back gate, so measuring the first device pre-charges the traps of all devices. This is why we avoid connected TLM sets and have only isolated devices.

Did you also observe this effect? Because this affects your extracted DIBL (re-measuring the same device at different V_d) and V_t roll off (several channel lengths in a TLM set). If the pre-charging effect is present, measuring a TLM set in both directions will yield different results.

5. Impact of sapphire edge to center (bilayer island density) is shown by AFM. Did you make electrical devices from both regions? Or if only from one region, can you mention which in the

article?

6. In fig2a it might help to draw an additional layer of PMMA. The current image shows that after delamination, the TMD is free-floating in the liquid without any support layer.

7. From fig2f, the SS of WS₂ is much worse than MoS₂. I think this is not discussed in the text. What could be the cause of the worse electrostatic control?

8. "...high performance with ON-state saturation current reaching 161 $\mu\text{A}/\mu\text{m}$ and 53 $\mu\text{A}/\mu\text{m}$, respectively, for $V_d = 5\text{ V}$ in "champion" devices". Could you additionally report champion I_{on} values at $V_d=1\text{V}$, which is more relevant?

9. (crucial) Also in other instances, instead of embracing the use of your large datasets and reporting median values, there is a tendency to focus on top performers (e.g. contact resistance of 2.6k $\Omega\text{-}\mu\text{m}$, DIBL of 1mV/V, mobility of 33, benchmarking tables). I suggest mentioning top performers but to focus on median values since that is the strength of this paper.

10. Fig4a-d with 3D histograms makes it difficult to read the axes and compare the values. Have you considered overlapping 2D histograms like in this example (<https://i.stack.imgur.com/3eQWs.png>), or cumulative probability plots?

11. (crucial) After the results of fig.4a-d, you claim "These results establish low device-to-device variation across the substrate, which can be attributed to uniform and...". Later you also claim "Furthermore, low device-to-device variation in these parameters imply that high quality and uniform growth of monolayer MoS₂ and WS₂ was achieved by our MOCVD technique". Can you first compare your σ_{mVt} to the four references on variability?

12. "SS was also found to be independent of the channel length for both MoS₂ and WS₂, suggesting that detrimental short-channel effects are absent from the subthreshold device characteristics". SS degradation is expected at shorter channel lengths than your minimum 100nm. I propose you mention you're outside the range and shorter channels are needed to check this.

13. "The deviation of the SS from the ideal value of 60 mV/dec is attributed to the presence of traps states" Can you explicitly evaluate D_{it} statistics?

14. You first make the statement "... neither MoS₂ nor WS₂ FETs show any notable increase in DIBL as the channel length is scaled from 5 μm down to 300nm. This highlights the superior electrostatic integrity of 2D FETs ..." but then you mention DIBL is present at 100nm and 200nm. I recommend removing the first part, since only the shortest channel values are relevant. If you want to claim competitive DIBL of MX₂ to Silicon, and can quote the DIBL value for Silicon at $L_{ch}=100\text{nm}$?

15. "Higher DIBL values for WS₂ FETs may indicate stronger capacitive coupling between the channel and the drain electrode, but requires more in-depth investigation, which is beyond the scope of this paper." I think this can be explained with WS₂ having worse SS, so more D_{it} , and hence less electrostatic control by the gate, so more DIBL. The results are therefore in agreement.

16. (crucial) "Remarkably, for "champion" devices DIBL values as low as 1.1 mV/V and 2.6 mV/V were recorded for MoS₂ and WS₂, respectively". Given the large standard deviation, and the large amount of devices with negative DIBL in the histogram, this DIBL value close to zero doesn't make sense, so I recommend removing this sentence.

17. When extracting I_{on}/I_{off} , can you mention how the value is obtained? If it's actually I_{max}/I_{min} , this is strongly dependent on the chosen measurement voltage range. Can you mention if I_{off} is taken at a fixed offset from V_t and therefore mainly determined by SS, or by the gate leakage floor or something else? Is it possible to identify this for each device and e.g. color code the plots? Since the definition of I_{off} is lacking, I don't understand the statement "... that I_{on}/I_{off} is expected to be independent of the channel length, when extracted at constant electric

field". I think the entire paragraph is difficult to understand and should be re-arranged so the goal and key message is stated clearly upfront.

18. "Ion/Ioff recorded for both TMDs are at par with the current CMOS technology". In Si CMOS, Ion/Ioff is mainly limited by fixed $V_{dd}=0.7V$. If for your devices you consider the entire sweep range, you need $V_{dd}=20V$. So i think you need to soften the statement.

19. "In summary, none of the key performance indicators related to the OFF-state, i.e. threshold voltage, SS, DIBL, Ion/Ioff show any noticeable and detrimental short-channel effects due to aggressive length scaling". You minimum channel length is 100nm, at which you observe an increased DIBL. But Silicon also has negligible short channel effects at 100nm. So i suggest strongly softening the statement.

20. 3D bar charts in fig.5a, b, i, j are hard to read. I think the format of e.g (c) is much easier to read, and with additional quantile boxplots in the same plot, all required data would be shown.

21. In general, I find the captions too long and hard to read, and they overlap strongly with the main text repeating the arguments instead of focusing on the key message. E.g. the caption of figure 4 contains this very long explanation which is better suited for the main text "The median values for these extracted threshold voltages (Table 1) were found to be more positive for WS₂ FETs compared to MoS₂ FETs, which can be ascribed to higher intrinsic n-type doping of MoS₂ either due to specific nature of impurity present in the MoS₂ film grown using metal-organic chemical vapor deposition (MOCVD) or surface charge transfer induced doping due to the underlying Al₂O₃ grown using atomic layer deposition (ALD). This charge transfer is accredited to the higher conduction band offset between MoS₂ and Al₂O₃ compared to WS₂ and Al₂O₃"

22. The discussion of contact-limited on-state current and 2-point probe field effect mobility, and the need for TLM structures, has been discussed in many other papers. I suggest moving this discussion and the formulas to the appendix.

23. Fig.5(g, h) only show top performers. Can you consider putting either every TLM line on the plot, and the median e.g. in red, or maybe a band to show the outer limits? Additionally it would be useful to calculate the error bar on expectation value of the contact resistance.

24. Fig.5(k,l) did you consider stacked bar charts (for each Lch: one bar for R_{ch}, one for R_c on top of it) to show the relative contribution of both resistances? It might be easier to understand intuitively.

25. The median long-channel extrinsic field effect mobility extracted from gm is 24cm²/Vs for MoS₂ and 33cm²/Vs for WS₂. Since these are valid for the longest channels where the impact of the contact is negligible, the same values should be obtained from the TLM mobility. You mention "From the slopes, following Eq 6, the effective mobility was extracted as 18 cm²/Vs and 14 cm²/Vs for the long-channel devices, 2.6cm²/Vs and 1 cm²/Vs, for the short-channel devices for MoS₂ and WS₂, respectively" but it is not clear if this is the TLM mobility, which should yield only a single value valid for all channel lengths, extracted from the slope of fig5(e) and (f).

26. This is more about personal preference; Many paragraphs are starting with e.g. "Fig. 5a-d and Table 2 show". I recommend starting with the key message/finding.

27. (crucial) The overall structure of the article should be improved and should be more concise. The last section "Benchmarking Monolayer MoS₂ and WS₂ FETs" contains 11 paragraphs but only one is about benchmarking, so i recommend splitting this across different sections. The paragraph starting with "Finally, high performance FETs are benchmarked..." spans several pages and is about a mix of several topics (1. the need for higher Ion, 2. Ion-Lch dependence (already treated in previous paragraph), 3. contact resistance limited (already treated in previous paragraph), 4. pinchoff and velocity saturation, 5. Ion of champion devices). We recommend rearranging this part, deleting repetitions or moving parts to the supplementary, to keep only the key novel findings.

28. [47] is reliable extraction of saturation velocity for MoS₂. Can you comment how your

extraction method differs from theirs, and why you obtain different v_{sat} values?

29. Table 4 contains values for I_{on} , but the V_d (5V?) is not mentioned. Including the data at $V_d=1V$ would be more relevant.

30. Tables 5.1 and 5.2 show champion values only. Can you please add the median values (e.g. in brackets) for your work and for the references if they are reported.

31. The caption of fig6. has partially cropped text (FET?) on the last line.

32. [48] has $</inf>$

Quentin Smets

Response to Reviewers' Comments

Reviewer's Comment

Our Response

Changes Made in the Manuscript

Reviewer #1 (Remarks to the Author):

Very good review with very useful data for the community.

We are happy to know that the reviewer finds our data useful for the community.

Reviewer #2 (Remarks to the Author):

The authors measure back-gated transistor characteristics of synthesized and transferred MoS₂ and WS₂ on Al₂O₃ on Si. The paper is significant in that it measures the properties of 230 and 130 FETs respectively and reports the distributions of the characteristics vs. gate length. There is little data like this and the results show the current state-of-the-art of synthesized channel devices in a university process. The analysis of the data however (on one wafer of MoS₂ and one wafer of WS₂) is uncritical and does not compare with Si MOSFETs even as it purports to assess technology readiness level. Despite what appears to be enormous variation relative to Si MOSFETs, the authors conclude in the end that ““Finally, low device-to-device variation, and high performance seen for scaled MoS₂ and WS₂ FETs exhibit high technology readiness level.” This seems at odds with the consistent and high variation they observe vs. Si CMOS, not to mention that they only obtain n-channel FETs in this study as there is no channel conductivity control. The data in this study is interesting however the analysis lacks a serious consideration and critical comparison with Si. It appears that the TRL is still at level 1 or 2 and not “high.” Below are comments and questions.

We are glad that the reviewer finds this work significant and acknowledges the fact that there is little data like this on state-of-the-art synthetic 2D material-based FETs. We agree with the reviewer's comment that comparison with Si MOSFET is important. We have included several benchmarking tables in the revised manuscript comparing our MOCVD grown monolayer MoS₂ and WS₂ FETs against other similar studies on 2D FETs found in the literature as well as ultra-

thin body (UTB) Si n-MOSFETs with similar gate lengths [1-6]. In the light of our analysis and response to reviewer's specific questions below, the reviewer will find that while in absolute terms the spatial variations in the respective benchmarking parameters appear to be large for MoS₂ and WS₂ FETs, when compared at scaled effective oxide thickness (*EOT*), our results are not significantly different compared to the projected variations for UTB Si n-MOSFETs. Nevertheless, we agree with the reviewer that the data in this study are interesting enough that the assessment for technology readiness level (TRL) is not necessary. A point-by-point response to the comments and concerns raised by the reviewer can be found below.

1. Fig 1b suggests 2-inch wafer process: Fig. 2a suggests smaller substrates What area is patterned with devices

We would like to thank the reviewer for raising this question. Both MoS₂ and WS₂ were grown on 2-inch sapphire wafer using metal organic chemical vapor deposition (MOCVD) technique. The 2-inch sapphire wafers were then cut into $1 \times 1 \text{ cm}^2$ pieces. For each material, two (2) $1 \times 1 \text{ cm}^2$ sapphire substrates were chosen, one corresponding to the center, and another one corresponding to the edge of the 2-inch wafer. Next the films were transferred to $1 \times 1 \text{ cm}^2$ device fabrication substrates i.e. 50 nm Al₂O₃ on Pt/TiN/p⁺⁺-Si following which MoS₂ and WS₂ field-effect transistors (FETs) were fabricated across the substrates with a total count of 230 and 130 FETs, respectively.

We have included the above information in the *Method* section in the revised manuscript.

2. Fig 1 caption. “ow variation” should be “Low variation”

We have corrected this typographical error in the revised manuscript.

3. Fig 1 caption, the statement “Intense PL is a characteristic of monolayer film owing to indirect to direct bandgap transition.” There is no measurement of intensity given so this statement seems strange. One would expect direct transitions to be stronger than indirect

transitions so what is the measure of “intense PL” and why does this indicate a single monolayer film?

We agree with the reviewer’s comment.

We have removed the phrase ‘Intense’ and revised the statement to “PL is a characteristic of monolayer film owing to indirect to direct bandgap transition.”

4. Fig. 2. Does the current scale with W/L? If not, please explain.

We would like to point out that all FETs were made with a constant channel width of $W = 5 \mu\text{m}$. Hence the scaling trend can be captured through the dependence of the drive-current (I_{ON}) on the channel length (L_{CH}). Fig. R1a and R1b show the median value for I_{ON} as a function of L_{CH} extracted at $V_{DS} = 1 \text{ V}$ for $n_s = 1 \times 10^{13} \text{ cm}^{-2}$ and $n_s = 4.4 \times 10^{13} \text{ cm}^{-2}$ for MoS₂ and WS₂ FETs, respectively. I_{ON} is expected to scale linearly with $1/L_{CH}$ following Eq R1.

$$\frac{I_{ON}}{W} = \frac{I_{DS,LIN}}{W} = qn_s\mu_{gm} \frac{V_{DS}}{L_{CH}} \quad [R1]$$

Figure R1. Extracted median values for I_{ON} as a function of $1/L_{CH}$ for a) MoS₂ and b) WS₂ FETs at V_{DS} of 1 V. The relative contribution of R_c and R_{ch} to the total resistance, R_T for c) MoS₂ and d) WS₂ FETs for different L_{CH} .

Here, $I_{DS,LIN}$ is the drain current extracted from the linear region, q is the charge of an electron, n_s is the carrier concentration, and μ_{gm} is the field-effect mobility. This trend is observed for both MoS₂ and WS₂ FETs in Fig. R1a and R1b, respectively, for longer devices with $L_{CH} \geq 1 \mu m$. However, for shorter devices with $L_{CH} < 1 \mu m$, the inverse channel length dependence is obscured by the contact resistance (R_c). Note that the total resistance (R_T) of an FET device is given by, $R_T = 2R_c + R_{ch}$. Here, R_{ch} is the channel resistance. While R_{ch} scales with L_{CH} , R_c remains constant, i.e. independent of L_{CH} , resulting in higher contribution of R_c in R_T for shorter devices. The relative contributions of R_{ch} and R_c to R_T are shown using stacked bar plots in Fig. R1c and R1d, for MoS₂ and WS₂ FETs, respectively. The increase in the contribution of R_c to R_T for shorter devices leads to the deviation from the linear current scaling trend as a function of $1/L_{CH}$.

We have added this discussion in the revised manuscript.

5. Fig. 2. Why are these devices stated to be contact limited? The 100 nm channel length transistors do not show “contact dominated transport” in the output characteristics in that the drain current in saturation does not increase linearly with VDS.

We thank the reviewer for raising this point. Note that the contact resistance ($2R_c$) in our MoS₂ and WS₂ FETs is a direct consequence of finite Schottky barrier (SB) heights at the metal/MoS₂ and metal/WS₂ contact interfaces, respectively. Unlike, the contact resistance in Si MOSFET, which is independent of the applied biases, the SB dominated $2R_c$ in MoS₂ and WS₂ FETs depends on V_{DS} and gate voltage (V_{GS}) as shown in Fig. R2a-c. In fact, one can write, $2R_c = R_s + R_d$, where, R_s and R_d are the contact resistances due to the SBs at the source and the drain contacts,

Figure R2 Dependence of R_s and R_d on V_{DS} and V_{GS} shown through energy band diagrams.

respectively. In the ON-state of device operation, i.e. for V_{GS} greater than the threshold voltage (V_T), $R_s \approx 0$ due to the electrostatic thinning of the SB (Fig. R2b) However, R_d depends on V_{DS} . For large V_{DS} , $R_d \approx 0$ as the SB is eliminated at the drain contact (Fig. R2c). Therefore, in the saturation region, $2R_c \approx 0$, and hence the drain current does not increase linearly with V_{DS} . However, in the linear region, R_d is finite and can dominate current transport in shorter channel length devices.

6. What testing protocol in the I-V measurements is utilized. Often 2D materials drift and change under repeated testing and measurement conditions need to be stabilized to get repeatability. What slew rate and slew direction are used for the I-V measurements to insure repeatability and absence of spurious charging? What burn-in procedure is utilized to ensure contacts are formed and stable? What is the magnitude of hysteresis measured in double sweeps? When negative differential resistance (NDR) and conductance fluctuations are observed, are these repeatable.

We would like to thank the reviewer for raising these questions.

Our measurement protocol is as follows: To ensure that the FETs are stabilized, they are conditioned by multiple repetitions of the same measurements. The transfer characteristics i.e. drain current (I_{DS}) versus V_{GS} are measured three times to condition each FET and the fourth measurement is used for the analysis. Fig. R3a shows 30 dual sweep measurements of a representative MoS₂ FET after conditioning. None to minimal variation between measurements confirms that the conditioning process is robust. The output characteristics i.e. I_{DS} versus V_{DS} are measured twice following the transfer characteristics and the second measurement is used for the analysis.

Standard DC sweeps using a Keysight B1500 semiconductor parameter analyzer was used for the measurements of transfer and output characteristics of all devices. However, we have now performed pulsed measurements and found that the FET characteristics are robust under various slew rates. Fig R3b shows the comparison between the DC measurement and pulsed measurements, using pulse widths of 1, 10, and 100 *ms*, keeping the step size constant to achieve different slew rates. We found no observable difference between these measurements. Similarly,

the measurements were robust when the slew direction was changed as shown in Fig R3c. Fig R3c also shows that the device exhibits minimal hysteresis ($<100\text{ mV}$).

We have found that no burn-in procedure is needed to ensure proper contact formation. Both MoS_2 and WS_2 FETs were measured as-fabricated.

The negative differential resistance (NDR) behavior was also found to be repeatable. Fig R3d shows the output characteristics measured at different slew rates using pulse widths of 1, 10, and 100 ms.

Figure R3. a) Transfer characteristics is measured 30 times for an MoS_2 FET in both directions using dual sweep measurements, following device conditioning. Minimal variation across the measurements indicate robust conditioning. b) Transfer characteristics measured using pulsed measurements with different slew rates and with DC measurements demonstrating that the measurements are not affected by the slew rate. c) The effect of slew direction is shown with a dual sweep measurement. The FET exhibits minimal hysteresis ($\approx 100\text{ mV}$). d) Output characteristics under different slew rates. Negative differential resistance behavior is seen consistently across all slew rates.

We have included the measurement protocols in the **Method** section in the revised manuscript.

7. Fig. 2(i) and (j) What is the origin of the NDR? Is it stable on repetitive sweeps and under different sweep rates?

As discussed in response to your comment # 7, the NDR behavior was found to be repeatable at different slew rates obtained using pulse widths of 1, 10, and 100 *ms*. The negative differential resistance behavior is seen due to self-heating effect. At high biases, high current density leads to self-heating. This is a common phenomenon seen in ultra-thin body FETs including silicon on insulator (SOI) FETs [7], nanowire FETs [8], graphene FETs [9], and more recently exfoliated multilayer MoS₂ FETs [10] as well as chemical vapor deposition (CVD) grown monolayer MoS₂ FETs [11]. These experimental findings indicate that the thermal boundary conductance at MoS₂/dielectric interface can be significantly low, which can limit the energy dissipation and degrade the performance of an FET resulting in the reduction of the drain current. It is possible to reduce or eliminate self-heating effect through pulsed measurements with pulse widths less than 100 μ s [10].

We have included the above discussion and references in the revised manuscript.

8. The terms long and short channel are applied to 5 and 0.1 μ m FETs. It is not clear that 0.1 μ m is short enough to be considered a “short channel” FET in this ultrathin channel geometry. The manifestation of short channel effect is that the drain current depends on $(1+\Lambda \cdot V_{DS})$. It appears both of the “short channel” MoS₂ and WS₂ transistors are long channel devices in that a linear dependence of I_D on V_{DS} is not observed. The discussion of the transistors as short and long channel appears inappropriate relative to the usual meaning of these terms.

We agree with the reviewer. We have changed the naming convention. We refer to the long channel devices as devices with $L_{CH} \geq 1 \mu\text{m}$ and the short channel devices as devices with $L_{CH} < 1 \mu\text{m}$.

9. Eq (1) Since mobility often depends on V_{DS} in 2D materials, more discussion is needed to clarify how mobility is reported. Is this the peak mobility with V_{DS} ? If so, does it satisfy the condition that V_{DS} is sufficiently low to justify use of Eq. 1? In this part of the paper, the

authors appear to be just giving one number instead of the median value and standard deviation. This is inconsistent.

The reviewer has raised an important point in regard to the specific question related to Eq (1) which is noted as Eq. R2 in this document. We have indeed reported the μ_{g_m} corresponding to the peak transconductance obtained from the slope of the transfer characteristics using Eq. R2.

$$\mu_{g_m} = \frac{dI_{DS}}{dV_{GS}} \left(\frac{L_{CH}}{WC_{OX}V_{DS}} \right) \quad [R2]$$

Here, C_{OX} is the oxide capacitance. We have found that μ_{g_m} is independent of V_{DS} in the chosen measurement range as shown in Fig. R4a and Fig. R4b for representative MoS₂ and WS₂ FETs, respectively. This is expected since $V_{DS} = 1\text{ V}$ corresponds to the linear operating regime, for both MoS₂ and WS₂ FETs justifying the use of Eq. R2 for the mobility extraction.

Reviewer's observation that in this part of the paper we have mentioned just one number for the mobility instead of the median value and standard deviation is correct. To avoid inconsistency, we have removed the discussion on field effect mobility from this section since the statistical analysis on the same is provided later.

Figure R4. Mobility extracted from the transfer characteristics for representative a) MoS₂ and b) WS₂ FETs for V_{DS} ranging from 0.2 V to 1 V.

10. Define “champion” devices? I assume this means the best transistors by some measure, but what criteria were applied in selecting these devices? Was it the best device for a single measure or was it best assuming a trade-off of measures?

We agree with the reviewer that “champion” device needs definition. “Champion” devices are defined for a single device parameter. For example, when “champion” devices are defined with respect to mobility, the FET with the best mobility is referred and the other parameters are not taken into consideration.

We have made this clarification in the revised manuscript.

11. page 7. Please clarify how these carrier densities are computed. It seems it is from the oxide capacitance, but it is not clear whether fringe and interface trap capacitance is included. The interface trap capacitance is later shown to be large with respect to Cox.

The reviewer brings up an important point. The carrier densities were indeed computed using C_{OX} . The contributions from fringe and interface trap capacitance (C_{IT}) were ignored due to the reasons mentioned below. C_{IT} was obtained using Eq. R3.

$$SS = \frac{k_B T \ln(10)}{q} \left(1 + \frac{C_{IT}}{C_{OX}} \right) \quad [R3]$$

Here, SS is the subthreshold slope, k_B is the Boltzmann’s constant, and T is the temperature. After evaluating C_{IT} from the subthreshold slope, Eq. R4 was used to evaluate n_s in the ON-state.

$$n_s = \frac{C_G}{q} (V_{GS} - V_{tlin}); \quad C_G = \frac{C_{OX} (C_{IT} + C_s)}{C_{OX} + C_{IT} + C_s}; \quad C_s = \frac{q^2 n_s}{k_B T} \quad [R4]$$

Here, C_G is the total gate capacitance, V_{tlin} is the threshold voltage, and C_s is the semiconductor capacitance. Note that, $C_s = 6.13 \times 10^{-1} F/m^2$ and $C_s = 2.7 \times 10^{-1} F/m^2$ corresponding to $n_s = 1 \times 10^{13} cm^{-2}$ and $n_s = 4.4 \times 10^{12} cm^{-2}$ for MoS₂ and WS₂, respectively. Maximum C_{IT} is found to be $2.5 \times 10^{-2} F/m^2$ and $1.7 \times 10^{-2} F/m^2$ for MoS₂ and WS₂, respectively and $C_{OX} = 1.6 \times 10^{-3} F/m^2$. Hence, in the ON-state $C_s \gg C_{IT}$ as well as $C_s \gg C_{OX}$, resulting in $C_G \approx C_{OX}$, simplifying Eq. R4 into Eq. R5.

$$n_s = \frac{C_{OX}}{q} (V_{GS} - V_{tlin}) \quad [R5]$$

To further validate our assumption, n_s is extracted using Eq. R4 and Eq. R5. Maximum n_s is obtained using the maximum overdrive voltage ($V_{GS} - V_{tlin}$), where $V_{GS} = 14 V$ and $V_{GS} = 12 V$ for MoS₂ and WS₂ FETs, respectively. Fig. R5a and Fig. R5b show the distribution of n_s for MoS₂

and WS_2 , respectively. The distributions are almost identical, justifying the use of Eq. R5 throughout the manuscript. Further, Table R1 summarizes the contribution of different capacitances to the distribution of n_s for MoS_2 and WS_2 FETs. Fringe capacitances were not considered since they are not expected to play a significant role in the extraction of n_s .

Figure R5 Distribution of maximum value of n_s obtained by using Eq. R4 and Eq. R5 for a) MoS_2 and b) WS_2 FETs.

Table R1: Variation in Carrier Densities n_s (10^{12} cm^{-2})				
	MoS_2		WS_2	
	Median, Mean \pm SD	Min, Max	Median, Mean \pm SD	Min, Max
$C_G = \frac{C_{OX} (C_{IT} + C_s)}{C_{OX} + C_{IT} + C_s}$	11.32, 11.34 \pm 0.83	9.37, 14.03	5.48, 5.43 \pm 0.84	3.92, 7.41
$C_G = \frac{C_{OX} C_s}{C_{OX} + C_s}$	11.32, 11.34 \pm 0.83	9.37, 14.03	5.48, 5.43 \pm 0.84	3.92, 7.41
$C_G = C_{OX}$	11.35, 11.36 \pm 0.83	9.39, 14.05	5.51, 5.45 \pm 0.84	3.95, 7.43

We have added the above justification in the **Supplementary Note 4**.

12. There does not appear to be data to substantiate that these transistors have monolayer vs. multilayer channels. AFM data of Fig. 1c,d, does not establish a ML film as no step heights are shown and the triangle pointed to says bilayer. There seems to be no way to tell the thickness of the film under the triangle. The PL measurements in 1i,j can be calibrated by other measurements to indicate the thickness but needs verification by another physical measurement or substantiation by a model to be used to get a thickness. Please clarify how these films were confirmed to be single monolayer.

The reviewer has a valid concern. We have now included the atomic force microscopy (AFM) data showing the step heights for MoS₂ and WS₂ confirming monolayers as shown in Fig. R6a and R6b for MoS₂ and WS₂, respectively [12]. Fig. R6c and Fig. R6d show Raman peak separation of $\approx 18 \text{ cm}^{-1}$ for MoS₂ and $\approx 60 \text{ cm}^{-1}$ for WS₂, respectively, which further confirm the presence of monolayers. Finally, photoluminescence (PL) peaks shown in Fig. R6e and Fig. R6f for MoS₂ and WS₂, respectively, are unique to monolayers and severely suppressed in thicker films.

Figure R6 AFM image and height profile across the red line for (a) MoS₂ and (b) WS₂. Raman spectrum with the characteristic peaks for (c) MoS₂ with E_{2g} at 386.6 cm⁻¹ and A_{1g} at 403.5 cm⁻¹ and (d) WS₂ with E_{2g} at 357 cm⁻¹, A_{1g} at 417.1 cm⁻¹, and 2LA(M) at 352.5 cm⁻¹. PL spectrum for (e) MoS₂ and (f) WS₂ show the characteristic monolayer response with peaks at 1.85 eV and 1.97 eV, respectively.

We have included the above information **Supplementary Figure 1**.

13. It appears that the results of this paper are based on two growths which are shown on 50 mm wafers in Fig. 1. Then some fraction of these films (1x1 cm²?) were transferred to a Si substrate. Please clarify how many growths are included in this study so there is no ambiguity. Or are the growths on cm² pieces?

Reviewer's observation is correct. Both MoS₂ and WS₂ were grown on 2-inch sapphire wafer using MOCVD technique. The 2-inch sapphire wafers were then cut into 1 × 1 cm² pieces. For

each material, two (2) $1 \times 1 \text{ cm}^2$ sapphire substrates were chosen, one corresponding to the center, and another one corresponding to the edge of the 2-inch wafer. Next the films were transferred to $1 \times 1 \text{ cm}^2$ device fabrication substrates i.e. $50 \text{ nm Al}_2\text{O}_3$ on Pt/TiN/p⁺⁺-Si following which MoS₂ and WS₂ FETs were fabricated across the substrates with a total count of 230 and 130 FETs, respectively. We have characterized the films obtained from one growth each for MoS₂ and WS₂.

We have made this clarification in the *Methods* section revised manuscript.

14. All figures could benefit from a doubling of font size. They are not easy to read even on a large screen. On Fig. 3 which shows 20 plots, the x and y-axes are repeated 20 times. The figures could be enlarged if the redundant labeling was removed. This approach could be used to advantage in almost every figure.

We thank the reviewer for their suggestion. We have increased the font sizes for the figures and removed redundant labeling.

15. Benchmarking typically refers to checking something by comparison to some standard. This paper appears to be characterizing the variation in device properties across two wafers, one MoS₂ and one WS₂. There is no standard or basis for comparison applied except to find median values and distributions on these two films. It seems relevant to compare to Si n-MOSFETs if the aim is to assess TRL.

We agree with the reviewer's comment that benchmarking must be performed against some standards. We have included several benchmarking tables in the revised manuscript comparing our MOCVD grown monolayer MoS₂ and WS₂ FETs against other similar studies on 2D FETs found in the literature and also ultra-thin body (UTB) Si n-MOSFETs with similar gate lengths [1-6] as discussed below.

Variation in threshold voltage and intrinsic carrier concentration are routinely used for benchmarking emerging devices based on novel materials [1]. Note that the actual magnitude of the threshold voltage is dependent on the thickness of the gate oxide, work function of the gate metal, and unintentional/intrinsic doping of the 2D material. Therefore, the standard deviation of

the threshold voltage (σ_{V_t}) is a better benchmarking metric, which also manifests in the variation in the intrinsic carrier concentration (σ_n) following Eq. R6.

$$\sigma_n = \frac{\sigma_{V_t} C_{OX}}{q} \quad [R6]$$

However, unlike σ_n , σ_{V_t} depends on the oxide thickness and hence for a fair comparison we use $S\sigma_{V_t}$, which is defined as the projected threshold voltage variation for a scaled effective oxide thickness (EOT) of 0.9 nm obtained using Eq. R7.

$$S\sigma_{V_t} = \sigma_{V_t} \frac{0.9}{EOT \text{ (in nm)}}; \quad EOT = \frac{t_{OX} \epsilon_{SiO_2}}{\epsilon_{OX}} \quad [R7]$$

Here ϵ_{SiO_2} is the dielectric constant of SiO₂. Our MoS₂ and WS₂ FETs with 50 nm Al₂O₃ gate dielectric show $\sigma_{V_t} = 0.8 V$ that corresponds to $\sigma_n = 8 \times 10^{11} \text{ cm}^{-2}$ and $S\sigma_{V_t} = 33 \text{ mV}$. These values are in agreement with the variability projected for CVD grown monolayer MoS₂ FETs reported by Smithe *et al.* [1]. We also employed this method to other reports on top-gated and wafer scale monolayer MoS₂ FETs and extracted $S\sigma_{V_t} = 45 \text{ mV}$ for [2] and $S\sigma_{V_t} = 11 \text{ mV}$ for [3], respectively. Recently, Smets *et al.*, [4] have demonstrated $\sigma_{V_t} = 44 \text{ mV}$ for an EOT of 1.9 nm that would correspond to $S\sigma_{V_t} = 20 \text{ mV}$ for monolayer MoS₂ FETs with channel lengths scaled down to 30 nm. These results are summarized in Table R2.

Table R2. Benchmarking Variation in Threshold voltage				
	σ_{V_t} (V)	σ_n (cm ⁻²)	Gate Dielectric	$S\sigma_{V_t}$ (V) at $EOT = 0.9 \text{ nm}$
[1]-MoS ₂	1.05	8×10^{11}	30 nm SiO ₂	33×10^{-3}
[2]-MoS ₂ 1 continuous layer	0.25	1.1×10^{12}	30 nm HfO ₂	45×10^{-3}
[2]-MoS ₂ 1 layer + ML	0.1	4.6×10^{11}	30 nm HfO ₂	19×10^{-3}
[3]-MoS ₂	0.17	2.8×10^{11}	30 nm Al ₂ O ₃	11×10^{-3}
[4]-MoS ₂	44×10^{-3}	5×10^{11}	4 nm HfO ₂	20×10^{-3}
Our Work- MoS₂, WS₂	0.8	8×10^{11}	50 nm Al ₂ O ₃	33×10^{-3}
[5] UTB SOI	78.5×10^{-3}	5.1×10^{12}	$EOT = 0.33 \text{ nm}$	214×10^{-3}

Remarkably, these variations are much smaller than $\sigma_{V_t} = 78.5 \text{ mV}$ for an EOT of 0.33 nm , i.e., $S\sigma_{V_t} = 214 \text{ mV}$ projected for UTB Si MOSFETs with a thickness of 2 nm [5]. Hence, 2D materials offer an alternative for the realization of UTB MOSFETs. Moreover, while it is possible to reduce the threshold voltage variation for 2D materials through optimization of growth and improvement in fabrication process flow, it is fundamentally limited for UTB Si owing to the significant thickness variation and random dopant fluctuations. In addition, detrimental quantum confinement effects lead to increase in the bandgap of ultra-thin Si. Finally, there is no known manufacturable solution for Si beyond the 5 nm technology node opening up opportunities for 2D materials for advanced scaling nodes.

Next, we use SS and density of interface trap states (D_{IT}) for benchmarking the OFF-state performance as shown in Table R3. For fully depleted UTB Si MOSFETs with 35 nm thick Si and 110 nm gate length, $SS = 80 \text{ mV/dec}$ for an $EOT = 4 \text{ nm}$ [6]. The median values of SS for our MoS₂ and WS₂ FETs are much higher, $SS = 450 \text{ mV/dec}$ and $SS = 550 \text{ mV/dec}$, respectively, for an $EOT = 22 \text{ nm}$ at similar channel length of 100 nm . However, SS depends on EOT following Eq. R8.

$$SS \text{ (mV/decade)} = \frac{k_B T \ln(10)}{q} \left(1 + \frac{C_{IT}}{C_{OX}}\right) = 60 \left(1 + q D_{IT} \frac{\epsilon_{SiO_2}}{EOT}\right); C_{IT} = q D_{IT} \quad [R8]$$

In fact, D_{IT} is a better metric for benchmarking since it is independent of EOT . Nevertheless, for a fair comparison of SS , we extracted D_{IT} corresponding to the SS using Eq. R8, and recalculated scaled- SS (SSS) for an EOT of 0.9 nm . We found that the projected values for SSS are 76 mV/dec and 80 mV/dec for MoS₂ and WS₂, respectively, and 64 mV/dec for UTB Si MOSFETs. A similar exercise was performed for other reports on MoS₂ FETs from the literature and included in Table R3.

Table R3. Benchmarking Median Subthreshold slope for $L_{CH} = 100 \text{ nm}$					
	$SS \text{ (mV/dec)}$	$EOT \text{ (nm)}$	Gate Dielectric	D_{IT} ($10^{12} \text{ eV}^{-1} \text{ cm}^{-2}$)	$SSS \text{ (mV/dec)}$ at $EOT = 0.9 \text{ nm}$
[4]-MoS ₂	80	1.9	4 nm HfO ₂	3.7×10^{12}	70
[4]-MoS ₂	160	2.7	8 nm HfO ₂	1.3×10^{13}	93
[4]-MoS ₂	200	3.8	12 nm HfO ₂	1.3×10^{13}	93
[4]-MoS ₂	1350	50	50 nm SiO ₂	9.2×10^{12}	83

Our Work - MoS₂	450	22	50 nm Al ₂ O ₃	6.3×10^{12}	76
Our Work - WS₂	550	22	50 nm Al ₂ O ₃	8×10^{12}	80
[6] UTB SOI	80	4	4 nm SiO ₂	1.8×10^{12}	64

The median value for I_{max}/I_{min} is 3.5×10^7 and 3.9×10^7 for both MoS₂ and WS₂ FETs, respectively for channel length of 100 nm. These values are over an order of magnitude higher than the I_{max}/I_{min} of 1.3×10^6 for UTB Si MOSFETs at similar channel lengths [6]. The comparison of I_{max}/I_{min} is tabulated in Table R4. Please note that in all demonstrations in Table R4, I_{max}/I_{min} is extracted across the entire sweep range of V_{GS} ($V_{GS,Range}$). Since the sweep range is strongly dependent on the EOT , it is useful to project the voltage ranges ($SV_{GS,Range}$) to a similar EOT of 0.9 nm using Eq. R9.

$$SV_{GS,Range} = \frac{V_{GS,Range} \times 0.9}{EOT} \quad [R9]$$

We demonstrate a higher I_{max}/I_{min} compared to UTB Si MOSFETs, however the measurement range for our devices is higher. Simulation results have predicted similar I_{max}/I_{min} for both Si and MoS₂ FETs for aggressively scaled devices [13].

Table R4. Benchmarking Median I_{max}/I_{min} for $L_{CH} = 100$ nm				
	I_{max}/I_{min}	EOT (nm)	$V_{GS,Range}$ (V)	$SV_{GS,Range}$ (V) at $EOT = 0.9$ nm
[4]-MoS ₂	$\approx 7 \times 10^6$	1.9	1.5 to -0.5 (2)	0.94
[4]-MoS ₂	$\approx 4 \times 10^6$	50	-	-
Our Work - MoS₂	3.5×10^7	22	14 to -3 (17)	0.65
Our Work - WS₂	3.9×10^7	22	12 to -5 (17)	0.65
[6] UTB SOI	1.3×10^6	4	1.8 to -0.2 (2)	0.45

The above discussions and tables have been added in the revised manuscript.

16. There does not appear to be any significant statistical analysis applied to this data. I am not sure what the special meaning of “statistical benchmarking”; is seems you are just

plotting multiple device characteristics or extracted parameters on the same scale. What total area is mapped in this study?

We have replaced “statistical benchmarking” with “variation” wherever applicable. Both MoS₂ and WS₂ FETs were fabricated across two substrates, each with an area of $1 \times 1 \text{ cm}^2$. One substrate has film transferred from the center of the 2-inch sapphire wafer (growth substrate) and another one from the edge of the same sapphire wafer.

17. Understanding the spatial variation could be important to shed light on the origin of the variation. The origin of the variation is not dealt with relative to what it needs to be. The factors appear to be variation in the film crystallinity, carrier density variations, or thickness uniformity. I would expect the film coalesces into a polycrystalline layer. What is known about this? This should be clarified and some attempt should be made to assess the physical causes of the variability.

We agree with the reviewer that it is important to discuss the possible origin of the variation seen in the electrical characteristics of monolayer MoS₂ and WS₂ FETs. However, as discussed elaborately in response to your comment #15, while in absolute terms the spatial variations in the respective benchmarking parameters appear to be large for MoS₂ and WS₂ FETs, when compared at scaled $EOT = 0.9 \text{ nm}$, our results are not significantly different from the projected variations for UTB Si n-MOSFETs. This is not entirely surprising since our MoS₂ and WS₂ monolayers synthesized by the MOCVD process grow epitaxially on the sapphire substrates and are single-crystalline in nature. This is confirmed by in-plane XRD phi-scans in Fig. R7a and R7b, for MoS₂ and WS₂, respectively, which show six-fold rotational symmetry and epitaxial alignment of the monolayer with the underlying sapphire. If the films were polycrystalline with a high degree of misorientation within the plane of the film, then we would expect to see multiple peaks at different angles in the in-plane XRD scan. These films were also transferred to a Quantifoil Cu grid to investigate the microstructure in TEM by using selected area diffraction pattern (SAED) and dark field imaging. As shown in detailed materials characterization of these films (<https://www.mri.psu.edu/2d-crystal-consortium/user-facilities/thin-films/list-thin-film-samples-available>) the respective SAED patterns show a single crystalline pattern, while composite dark

field maps illuminate two contrasting regions in the monolayer films. For MoS₂ these regions correspond to anti-phase domains. For WS₂ however, the regions are unidirectional and are separated by translational boundaries. Additional information about the MOCVD growth and structural properties of the films can be found in related manuscripts on arXiv <https://arxiv.org/ftp/arxiv/papers/2006/2006.11668.pdf> <https://arxiv.org/ftp/arxiv/papers/2006/2006.10952.pdf>

We believe that it is possible to reduce the spatial variations in 2D FETs through further optimization of growth and improvement in fabrication process flow, which is unlikely for UTB Si owing to significant thickness variations at length scales similar to monolayer MoS₂ and WS₂. In addition, random dopant fluctuations, detrimental quantum confinement effects leading to increase in the bandgap of ultra-thin Si, and lack of manufacturable solution for Si beyond the 5 nm technology node open up opportunities for 2D materials for advanced scaling nodes.

Figure R7 XRD of monolayer WS₂ and MoS₂. In-plane XRD ϕ scan of a) MoS₂ and b) WS₂ on sapphire (α -Al₂O₃), showing epitaxial relationship between the monolayers and sapphire substrate.

We have added discussion in the revised manuscript and **Supplementary Notes 2**

18. A more accurate title might be “Variation in synthetic, layer-transferred MoS₂ and WS₂ field-effect transistors” unless some benchmarking to Si MOSFETs can be added. There seems no more statistical analysis than are typically applied to measurements.

We thank the reviewer for their suggestion. With the revised version, where we have benchmarked MoS₂ and WS₂ FETs with UTB Si MOSFETs and other reports of 2D FETs, we believe that our title is appropriate now.

19. Terms like ALD, FETs, MOCVD, SEM, TLM, TMD, TRL etc. and parameter symbols are defined multiple times in the paper. These need only be defined once.

We have made sure that they are only defined once.

20. The authors state “A fair assessment of the technology readiness level (TRL) necessitates large device statistics to understand the variation in the FET performance across the entire chip, as well as thorough study of the impact of channel length scaling on the FET performance of the technology. This was accomplished ...” This idea that TRL needs to be assessed is raised by the authors and then it is said to be accomplished, however the authors do not seriously address this question. What is the TRL of MoS₂ and WS₂ FETs? What are the measures? Based on the authors findings, it is not clear there is enough data to support TRL assessment. How can a study of 2 wafers using a transfer process provide data to assess TRL?

The reviewer raises an important point. We agree with the reviewer that it is neither necessary nor appropriate to invoke the idea of TRL assessment for this manuscript. Therefore, we have replaced TRL level assessment with advancement in state-of-the-art 2D FETs. Nevertheless, in the revised version, we have included more extensive benchmarking of various parameters such as threshold voltage, subthreshold slope, drain-induced barrier lowering, ratio of maximum to minimum current, mobility, and drive current with other demonstrations of 2D FETs in the literature. Furthermore, we have compared the results of our MoS₂ and WS₂ FETs with UTB Si MOSFETs at a similar gate length. These comparisons naturally show the potential for 2D FETs without invoking the need for TRL assessment.

21. Fig. 4abcd It is not clear why the authors choose to plot the threshold voltage and SS using this variety of definitions. Why is it relevant that SS improves when compared over

decades. The 3D plots do not allow the methods to be closely compared and the distribution widths seem to contain the most relevant information. What conclusions can be drawn from this comparison across these different methodologies? It seems relevant to make comparison against published measures for CMOS at some gate length.

We are happy to provide some explanation for our approach.

In 2D FET literature, threshold voltage is extracted using various methods such as linear extrapolation, Y-function method, and constant-current method. Linear extrapolation is the most common among these techniques. However, poor ON-state performance, presence of SB at the metal/semiconductor interface, and contact-gating effect in a back-gated geometry can limit the use of linear extrapolation. Y-function method is more appropriate for contact dominated FETs. It is found that for the MoS₂ and WS₂ FETs presented in this study, the distributions of threshold voltage extracted from linear extrapolation and Y-function methods are very similar, confirming that our FETs are well behaved. Finally, constant current method is simply another threshold voltage extraction technique, which is discussed for completeness.

The SS is extracted across one (SS_1), two (SS_2), three (SS_3), and four (SS_4) orders of magnitude of the drain current. While for an FET with Ohmic contact it is expected that $SS_1 = SS_2 = SS_3 = SS_4$, for an SB FET it is expected to increase when extracted across higher orders of magnitude of the drain current. A greater increase can be related to higher SB height at the metal/semiconductor interface and hence poorer carrier injection. In the existing 2D FET literature there is a tendency to report SS values without mentioning the orders of magnitude of the drain current over which they are evaluated. This leads to considerable discrepancy and unfair comparisons. In fact, most SS values are reported for only one or two orders of magnitude of the drain current, whereas circuit operations require at least four orders of magnitude ON/OFF ratio to be technologically relevant. By accessing SS_1 , SS_2 , SS_3 , and SS_4 we have tried to bridge the gap between the comparison with the existing 2D literature and what is technologically required.

Additionally, we have compared our devices with UTB Si at similar gate lengths as response to reviewer's comment #15.

We have moved the different definitions to *Supplementary Information file* and also converted the 3D plots to overlapping 2D plots.

22. The figure captions are so long that it is not easy to scan them for the relevant information about what is plotted. The captions often discuss the data and repeat information already given in the text. The captions would be much improved if they were more compact and the discussion in the captions consolidated into the text where the repeats are eliminated.

We thank the reviewer for their comment. The figure captions have been revised accordingly.

23. As an example Fig. 4 caption states “None of the key performance indicators related to the OFF-state, i.e. threshold voltage, SS ...show any noticeable and detrimental short-channel effects, which can be ascribed to the ultra-thin body nature of monolayer TMDs, as well as the use of high-k Al_2O_3 as the gate dielectric.” This discussion belongs in the text. The smallest gate lengths tested here are not short channel devices so it is not surprising that they do not show short-channel effects. They would not be short channel device even if the TMD thickness were thicker and this second argument about Al_2O_3 is similarly not well explained. Quantifying some of these assessments would be more valuable than providing these general assertions.

We agree with the reviewer suggestions. We have moved the highlighted discussion from Fig. 4 caption to the main text. We also agree with the reviewer that none of our devices including the ones with $L_{CH} = 100\text{ nm}$ would qualify as short channel devices given that we are using ultra-thin body channel, i.e. monolayer MoS_2 and WS_2 , and relatively thin and high-k gate dielectric, i.e. 50 nm thick Al_2O_3 with an EOT of 22 nm. Therefore, following reviewer’s recommendation, we have removed the general assertion and revised the statements accordingly.

24. Fig. 4 caption “Furthermore, low device-to-device variation in these parameters (Table 1) imply that high quality and uniform growth was achieved by our MOCVD technique.” Move this to the discussion of Table 1 and quantify the statements. What is “low?” What is the measure of “high quality?” What is the measure of “uniform.” How do these compare with Si?

Reviewer's comments are appreciated. The statement has been moved to the discussion. In response to reviewer's comment #15, we have quantified the device-to-device variation and benchmarked the relevant device parameters with state-of-the-art 2D FETs and UTB Si MOSFETs. Similarly, in response to reviewer's comment #17, we have provided additional details on film uniformity and crystallinity.

25. page 10. The authors state the median values for V_t across different techniques and then say "These results establish low device-to-device variation across the substrate, which can be attributed to uniform and contaminant-free MOCVD growth of monolayer TMDs and clean device fabrication process." The distributions extend over volts which shows a technology with variation too large for VLSI use, yet the authors develop a conclusion the films are uniform and contaminant-free. The results should be quantified and a serious comparison made.

We agree with reviewer that our results on device-to-device variation must be quantified. In response to reviewer's comment #15, we have quantified the device-to-device variation and benchmarked the relevant device parameters with state-of-the-art 2D FETs and UTB Si MOSFETs. Similarly, in response to reviewer's comment #17, we have provided additional details on film uniformity and crystallinity. Note that while in absolute terms the spatial variations in the respective benchmarking parameters appear to be large for MoS₂ and WS₂ FETs, when compared at scaled $EOT = 0.9 \text{ nm}$, our results are not significantly different from the projected variations for UTB Si n-MOSFETs.

26. The discussion of SS is similarly cursory. Considering SS4, which is a minimum standard for MOSFETs, the median values are 431.9 and 541.4 mV (I don't know why the authors think they have 4 significant digits of accuracy in these measurements) indicating a large interface trap density, yet this is ignored and they say this can be "mitigated by scaling the oxide thickness." which does not solve the inherent trap problem. I see no particular insights drawn by computing SS1 SS2 or SS3. This is compared in the supplementary material and that is probably where it belongs. Obviously, the SS will be monotonically lower if you focus

on smaller voltage intervals. The variation in SS is another strong indicator of nonuniformity and this is not analyzed. Why is the interface trap density not computed or assessed?

We thank the reviewer for their comments. We have revised the SS values with 3 significant digits of accuracy instead of 4. We are happy to extract D_{IT} , and we agree with the reviewer that it will be useful to include the statistics of D_{IT} . D_{IT} was evaluated using Eq. R8. Here, subthreshold slope across four orders of magnitude is used. The distribution of D_{IT} is shown using histograms in Fig. R8a and R8b for MoS_2 and WS_2 , respectively. Median values of $6.2 \times 10^{12} \text{ eV}^{-1}\text{cm}^{-2}$ and $8 \times 10^{12} \text{ eV}^{-1}\text{cm}^{-2}$ were obtained for MoS_2 and WS_2 , respectively. While we agree with the reviewer that the high density of interface traps cannot be mitigated by scaling the oxide thickness, its impact on SS can be considerably reduced. To demonstrate this, the subthreshold slope is projected for an EOT of 0.9 nm . The distribution of subthreshold slope for EOT of 0.9 nm is shown in Fig. R8c and R8d. The variation is summarized in Table R5, with median SS of 76 mV/dec and 80 mV/dec , for MoS_2 and WS_2 , respectively. These median values for scaled oxides have been benchmarked against state-of-the-art 2D FETs and UTB Si FETs as response to reviewer's comment #15. To further improve the SS , the density of interface trap states must be reduced. The

Figure R8. Variation in D_{IT} for a) MoS_2 and b) WS_2 FETs. The projected subthreshold slope for an EOT of 0.9 nm for a) MoS_2 and b) WS_2 FETs.

presence of structural defects such as sulfur vacancies are known to introduce trap sites which contribute to D_{IT} . It has been found that D_{IT} can be reduced by various surface passivation techniques [14, 15]. In addition, photoresist residue from the lithography and/or the wet transfer process can cause an increase in D_{IT} . Therefore, it is possible to reduce D_{IT} through further optimization of growth, post-growth processing, and improvement in fabrication process flow. The rationale for extracting SS across 1, 2, 3, and 4 orders of magnitudes is discussed in response to reviewer's comment #21.

Table R5: Variation in D_{IT} and SS				
	MoS_2		WS_2	
	Median, Mean \pm SD	Min, Max	Median, Mean \pm SD	Min, Max
$D_{IT}(10^{12} \text{ eV}^{-1}\text{cm}^{-2})$	6.2, 6.6 \pm 2.3	1.75, 15.7	8, 8.1 \pm 0.7	6.5, 11.2
$SS_{4,EOT=0.9}(\text{mV/dec})$	76, 76 \pm 6	64, 99	80, 80 \pm 2	76, 88

The discussion of D_{IT} and figures are added in the revised manuscript.

27. Fig. 4ef, Certainly DIBL can be computed given the definition, but what measurements indicate that DIBL is due to the drain field reaching through to the source? One place to look for this is in the common source characteristics and these show more dI_D/dV_{DS} in the 5 μm devices than the 0.1 μm devices. This seems to argue against the simple explanation of the authors.

We agree with the reviewer that a more extensive study using aggressively scaled devices is required to access *DIBL* in 2D FETs. Therefore, we have removed the discussion on *DIBL* from our revised manuscript.

28. Why is eq (1) used to analyze the mobility vs. eq. (2)? Why should the square law model be favored over the Y-function model for this analysis. Are the two models in agreement? The U_y is given in the supplemental data. What is the reason for the gate length dependence? It is said to be due to “contact effects”, but what are these contact effects?

We are happy to provide clarification. We have extracted mobility using two methods. First, the field-effect mobility is extracted from the transfer characteristics using the peak transconductance using Eq. R10.

$$\mu_{g_m} = \frac{dI_{DS}}{dV_{GS}} \left(\frac{L_{CH}}{WC_{OX}V_{DS}} \right) \quad [R10]$$

Further, we employ the Y-function method to extract the mobility. Y-function is given by Eq. R11

$$Y = \frac{I_{DS}}{\sqrt{g_m}} = (\mu_Y WC_{OX} V_{DS})^{0.5} (V_{GS} - V_{t_Y}) \quad [R11]$$

Here, g_m is the transconductance, μ_Y is the Y-function mobility, and V_{t_Y} is the Y-function threshold. The Y-function mobility is extracted from the slope of Y-function *versus* $(V_{GS} - V_{t_Y})$. To compare the extraction methods, distributions are shown in Fig. R9a and R9b for both MoS₂ and WS₂. Here, the distribution of mobility is shown for different channel lengths, and close agreement is seen between both mobility extraction methods. Since both models are in agreement, we chose to show μ_{g_m} in the main manuscript and μ_Y in supporting information. The channel length dependence is seen as a result of high R_c . Note that the R_T of an FET device is given by, $R_T = 2R_c + R_{ch}$. While R_{ch} scales with L_{CH} , R_c remains constant, i.e. independent of L_{CH} , resulting in higher contribution of R_c in R_T for shorter devices. The relative contributions of R_{ch} and R_c to R_T are shown using stacked bar plots in Fig. R1c and R1d, for MoS₂ and WS₂, respectively. The increase in the contribution of R_c to R_T for shorter devices prevent devices from reaching its peak transconductance, leading to mobility underestimation [16]. This results in the extracted mobility being channel length dependent and being severely underestimated showing

Figure R9. Comparison of mobility extracted from peak transconductance and Y-function for a) MoS₂ and b) WS₂ FETs.

more than 80% degradation going from $L_{CH} = 5 \mu\text{m}$ to $L_{CH} = 100 \text{ nm}$ for both MoS_2 and WS_2 FETs.

We have made the clarification in the main manuscript.

29. The analysis of resistance ignores the interface trap density which appears to be large with respect to the channel sheet density. It seems the plots in Fig 5ij of RC vs. n_s have little meaning. The most important aspect is the contact resistance variation which as with all the parameters is large with respect to use in applications.

We have evaluated the effect of interface trap density on the carrier concentration extraction in response to reviewer's comment #11, and it is seen to be minimal. This is a result of the semiconductor capacitance being much larger than the capacitance due to interface traps. Additionally, at significantly high carrier concentrations, the contact resistance is not significantly affected by change in carrier concentration. Fig. R10a and R10b shows the dependence of contact resistance on carrier concentration for MoS_2 and WS_2 , respectively, for all TLM devices. The median and the data points representing the 25th and 75th percentile are also shown.

Figure R10. Dependence of contact resistance on carrier density for all a) MoS_2 and b) WS_2 TLM structures.

We agree with the reviewer that the contact resistance and its variation is high, when compared to $R_c = 0.1 \text{ k}\Omega - \mu\text{m}$, typically reported for state-of-the-art Si FETs. However, various methods have been developed to reduce the effect of SB limited carrier transport in 2D TMDs [17] such as work function engineering to reduce the SB height [18], introduction of interlayers such as graphene to decouple the metal/2D interface alleviating Fermi-level pinning [19, 20], and achieving higher

carrier concentration underneath or near the metal/2D contacts through substitutional or surface charge transfer doping to reduce the SB width [21, 22]. Nevertheless, our MOCVD grown monolayer MoS₂ FETs demonstrates R_c similar to values reported in the literature [1, 4, 23, 24]. Additionally, please note that our work marks the first study on extraction of contact resistance from multiple TLM structures for both MoS₂ and WS₂. Smithe *et al.* has demonstrated a pseudo-TLM analysis where independent devices with different channel lengths and widths were used to extract the distribution of R_T , TLM analysis is done on the devices between 10th and 90th percentile [1]. Our demonstration involves extraction of contact resistance from separate TLM structures and finding the variation across these TLM structures, and the analysis is not limited to a percentile limit.

We have revised the figures in the manuscript and added further discussion.

30. I find the remaining discussion of the data in Figs 6 and 7 and the tables to be simple and uncritical. Typically a few statements giving some values from the figures and then sweeping uncritical statements related to long and short channel effects which do not appear to be manifest from the measurements.

We have now added more critical discussion in the revised manuscript.

31. The final statement “Finally, low device-to-device variation, and high performance seen for scaled MoS₂ and WS₂ FETs exhibit high technology readiness level.” appears to be at odds with the measured data.

In response to reviewer’s comment #15, we have quantified the device-to-device variation and benchmarked the relevant device parameters with state-of-the-art 2D FETs and UTB Si MOSFETs. Similarly, in response to reviewer’s comment #17, we have provided additional details on film uniformity and crystallinity. Finally, we have removed our discussion on TRL.

Reviewer #3 (Remarks to the Author):

The paper is high quality, well written and in clear English. It provides a complete characterization of MoS₂ and WS₂ FETs. The considered metrics are well chosen (drive current, threshold voltage, field-effect carrier mobility, carrier saturation velocity, contact resistance, subthreshold slope, current ON/OFF ratio, and drain-induced barrier lowering). The MX₂ literature needs additional variability studies like this one. Here are my comments:

We are glad that the reviewer finds this work to be of high quality and well written. We also appreciate the fact that the reviewer acknowledges the need for similar work in the 2D community. We have provided point-by-point answers to the comments and concerns raised by the reviewer below.

1. (crucial) a record mobility for WS₂ of 33 cm²/Vs is quoted in abstract and text, but several higher values are found in literature. Can you specify if you mean only for monolayer and/or only synthetic material?

-this paper shows 83 <https://onlinelibrary.wiley.com/doi/abs/10.1002/adma.201502222>

-this paper shows 50 <https://onlinelibrary.wiley.com/doi/abs/10.1002/adfm.201703448>

-this paper shows 140 <https://pubs.acs.org/doi/10.1021/nn502362b>

The reviewer's observation is accurate. There are higher mobility values reported for monolayer WS₂ in the literature. For example, Cui, Y. *et al.*, [25] show room-temperature electron mobility of 83 cm²/Vs and Ovchinnikov, D. *et al.*, [26] show low temperature (83 K) electron mobility of 140 cm²/Vs in exfoliated monolayer WS₂. The room temperature electron mobility reported in [26] was found to be 50 ± 7 cm²/Vs. Aji, A. S. *et al.*, [27] reported an average electron mobility of 27 cm²/Vs with the highest value reaching 50 cm²/Vs in chemical vapor deposited (CVD) monolayer WS₂ with multilayer graphene (MLG) contacts. However, the electron mobility values were found to be in the range of 5 cm²/Vs when conventional Ti/Au metal contacts are used i.e. without contact engineering *via* MLG. In contrast, the electron mobility of 33 cm²/Vs in our WS₂ field-effect transistors (FETs) are extracted at room temperature and without contact engineering

for metal-organic chemical vapor deposition (MOCVD) grown synthetic monolayers. Hence, we feel that our claim of record high mobility in synthetic monolayer WS₂ is valid.

The following statement has been added in the main text.

While higher mobility values are reported for WS₂ in the literature, those are either for exfoliated materials at room temperature [25] and low temperatures [26], or for CVD grown materials with contact engineering *via* the use of multilayer graphene as interlayers [27].

2. (crucial) You mention “However, one must admit that a genuine assessment of the technology readiness level (TRL) for 2D materials should include a comprehensive benchmarking and scaling study of all relevant transport parameters using statistically significant number of devices.”

I think the state of the art and goal of the paper is not sufficiently well introduced. I recommend mentioning the following papers on variability

- [41] (your paper overlaps strongest with this one)
- [7] “H. Xu et al., *Small*, 14, 48, p.1803465 (2018)”
- [5] **Can you mention these papers in the introduction, what they lack, and what is innovative in your paper?**

We appreciate reviewer’s comment and concur with their suggestions. We have added excerpts from the following discussion in the revised manuscript.

To assess the technological maturity of FETs based on 2D materials, it is important to find the variation in key parameters that determine the ON-state and OFF-state performance across a large number of devices. Unfortunately, there are only few studies that report device-to-device variation in 2D FETs [1, 2, 4]. For example, Smithe *et al.*, measured multiple parameters across 200 MoS₂ FETs and demonstrated low threshold voltage variation and low contact resistance on the order of 1 k Ω – μ m [1]. Similarly, Xu *et al.*, analyzed 380 top-gated MoS₂ FETs and reported variation in threshold voltage and electron mobility [2]. However, both works concentrate on longer channel devices where the effects of contact resistance are not pronounced. In a separate study Smithe *et al.*, [23] measured scaled MoS₂ FETs based on synthetic monolayers, however they did not provide any statistics. Smets *et al.*, [4] demonstrated the most significant work on scaling study of CVD

grown monolayer MoS₂ where multiple devices with channel lengths ranging from 5 μm down to 29 nm were measured. However, their study was focused on the OFF-state performance. Finally, all of the aforementioned studies are based on MoS₂ FETs, and none exists for WS₂ FETs. Our work can, therefore, be considered more comprehensive since we report variation in key parameters related to both ON-state and OFF-state that include drive current, threshold voltage, field-effect carrier mobility, carrier saturation velocity, contact resistance, subthreshold slope, and ratio of maximum to minimum current for different channel lengths ranging from 5 μm down to 100 nm using 230 MoS₂ FETs and 160 WS₂ FETs. In addition, we offer extensive benchmarking of our devices with respect to the above-mentioned demonstrations as well as UTB Si MOSFETs with similar gate lengths to access the technological viability and maturity of 2D FETs.

We have included this discussion and the references in Introduction section of the revised manuscript.

3. I think “substrate agnostic growth” is incorrect. The substrate is crucial, as you later state, “Fig. 1e and 1f highlight the epitaxial relation between the sulfide monolayers and the underlying sapphire substrates”

We agree with the reviewer that substrate is critical to ensure high quality growth of 2D materials. We have modified statement in the revised manuscript:
 “In this context, chemical vapor deposition (CVD) and metal-organic CVD (MOCVD) are the most promising techniques, enabling growth of high quality 2D materials with different thermal budgets on various substrates.”

4. In our team, when we measure an uncapped TMD channel device twice, the second curve has a strong positive V_t shift (1V shift for 4V sweep) because traps have been pre-charged. In a TLM set, all devices are connected together and share the same back gate, so measuring the first device pre-charges the traps of all devices. This is why we avoid connected TLM sets and have only isolated devices. Did you also observe this effect? Because this affects your extracted DIBL (re-measuring the same device at different V_d) and V_t roll off (several

channel lengths in a TLM set). If the pre-charging effect is present, measuring a TLM set in both directions will yield different results.

The reviewer raises a valid point. We do see a similar behavior with our MoS₂ and WS₂ FETs. After the first measurement, we do see a positive shift in the threshold voltage. Due to this effect, we condition our FETs before the actual measurement is taken. The transfer characteristics i.e. the drain current (I_{DS}) versus back-gate voltage (V_{GS}) is measured three times to condition a FET and the fourth measurement is used for the analysis. The shift in threshold voltage for consecutive sweeps is shown for a MoS₂ FET in Fig R11a. The threshold voltage shifts by around 0.5 V between the first and fourth sweep. However, post-conditioned FETs are found to be very stable. Fig R11b shows 30 dual sweeps of the transfer characteristics of a post-conditioned MoS₂ FET. Since we follow this conditioning step for each and every FET in a transmission-line-measurement (TLM) structure our measurements were not impacted by direction. Fig R11c shows the transfer characteristics of MoS₂ FETs of different channel lengths corresponding to representative TLM structure measured in both directions. Starting from the channel length of 100 nm, all MoS₂ FETs up to the channel length of 5 μm are measured. Following this, the devices are remeasured in the reverse order. As expected, the characteristics are found to be very similar.

Figure R11 Measurement reliability. a) For an MoS₂ FET, transfer characteristics is measured four times to ensure that the FET is conditioned. By the fourth measurement, the threshold voltage stops shifting and is stabilized. b) Following the condition step, transfer characteristics is measured 30 times for an MoS₂ FET in both sweep directions using dual sweep measurements. Minimal variation across the measurements indicate robust conditioning. c) MoS₂ FETs with different channel lengths in a TLM structure measured in both directions.

We have included the measurement protocols in the *Method* section in the revised manuscript.

5. Impact of sapphire edge to center (bilayer island density) is shown by AFM. Did you make electrical devices from both regions? Or if only from one region, can you mention which in the article?

We would like to thank the reviewer for raising this question. Both MoS₂ and WS₂ were grown on 2-inch sapphire wafer using MOCVD technique. The 2-inch sapphire wafers were then cut into $1 \times 1 \text{ cm}^2$ pieces. For each material, two (2) $1 \times 1 \text{ cm}^2$ sapphire substrates were chosen, one corresponding to the center, and another one corresponding to the edge of the 2-inch wafer. Next the films were transferred to $1 \times 1 \text{ cm}^2$ device fabrication substrates i.e. 50 nm Al₂O₃ on Pt/TiN/p⁺⁺-Si following which MoS₂ and WS₂ FETs were fabricated across the substrates with a total count of 230 and 130 FETs, respectively.

We have mentioned this information in the revised manuscript.

6. In fig2a it might help to draw an additional layer of PMMA. The current image shows that after delamination, the TMD is free-floating in the liquid without any support layer.

We would like to thank the reviewer for this suggestion. We have added a PMMA layer in the schematic. The corrected schematic is shown in Fig R12.

Figure R12 a) Schematic representation of poly(methyl methacrylate) (PMMA)-assisted wet transfer

7. From fig2f, the SS of WS2 is much worse than MoS2. I think this is not discussed in the text. What could be the cause of the worse electrostatic control?

Reviewer's observation is correct. Unfortunately, we do not have a concrete explanation for the worse electrostatics in WS₂ compared to MoS₂. Here are some hypothesis, which require validation through further investigation. The XPS measurements show that S:Mo for MoS₂ is $\approx 2.07:1$, whereas S:W for WS₂ is $\approx 2.48:1$. The excess sulfur may lead to more interface trap states in WS₂ compared to MoS₂. While the contribution of metal (Mo or W) *d* orbitals is dominant in the formation of midgap states, the sulfur *p* and *d* orbitals can also contribute to the formation of localized states. For example, numerical simulation using density functional theory (DFT) suggests that excess sulfur atoms can introduce energy states near the valence band maximum in the band gap of MoS₂ [28]. We do not know where and why the excess S exists for WS₂. It is likely to be on the surface or at the interface between WS₂ and sapphire. Another difference between WS₂ and MoS₂ is in their grain boundary structure. MoS₂ has anti-phase boundaries which are known to be metallic in nature, whereas WS₂ boundaries are mostly translational whose electrical properties are unknown.

8. "...high performance with ON-state saturation current reaching 161 $\mu\text{A}/\mu\text{m}$ and 53 $\mu\text{A}/\mu\text{m}$, respectively, for $V_d = 5\text{ V}$ in "champion" devices". Could you additionally report champion Ion values at $V_d=1\text{V}$, which is more relevant?

We would like to thank the reviewer for this suggestion. The highest ON-state currents for $V_{DS} = 1\text{ V}$ was found to be $73\ \mu\text{A}/\mu\text{m}$ and $26\ \mu\text{A}/\mu\text{m}$ for MoS₂ and WS₂, respectively.

This is included in the revised manuscript.

9. (crucial) Also in other instances, instead of embracing the use of your large datasets and reporting median values, there is a tendency to focus on top performers (e.g. contact resistance of 2.6kOhm- μm , DIBL of 1mV/V, mobility of 33, benchmarking tables). I suggest mentioning top performers but to focus on median values since that is the strength of this paper.

We agree with the reviewer that the median values are the strength of this paper. We have focused on the median values wherever possible.

10. Fig4a-d with 3D histograms makes it difficult to read the axes and compare the values. Have you considered overlapping 2D histograms like in this example), or cumulative probability plots?

We thank the reviewer for the suggestion. We have replaced the 3D histograms with overlapping 2D histograms as suggested. However, the comparison of different extraction methods has been moved to *Supplementary Information*.

11. (crucial) After the results of fig.4a-d, you claim “These results establish low device-to-device variation across the substrate, which can be attributed to uniform and...”. Later you also claim “Furthermore, low device-to-device variation in these parameters imply that high quality and uniform growth of monolayer MoS₂ and WS₂ was achieved by our MOCVD technique”. Can you first compare your sigma_{V_t} to the four references on variability?

We are happy to make comparisons of threshold voltage with other references. We have included several benchmarking tables in the revised manuscript comparing our MOCVD grown monolayer MoS₂ and WS₂ FETs against other similar studies on 2D FETs found in the literature and also ultra-thin body (UTB) Si n-MOSFETs with similar gate lengths [1-6] as discussed below.

Variation in threshold voltage and intrinsic carrier concentration are routinely used for benchmarking emerging devices based on novel materials [1]. Note that the actual magnitude of the threshold voltage is dependent on the thickness of the gate oxide, work function of the gate metal, and unintentional/intrinsic doping of the 2D material. Therefore, the standard deviation of the threshold voltage (σ_{V_t}) is a better benchmarking metric, which also manifests in the variation in the intrinsic carrier concentration (σ_n) following Eq. R12.

$$\sigma_n = \frac{\sigma_{V_t} C_{OX}}{q} \quad [R12]$$

However, unlike σ_n , σ_{V_t} depends on the oxide thickness and hence for a fair comparison we use $S\sigma_{V_t}$, which is defined as the projected threshold voltage variation for a scaled effective oxide thickness (*EOT*) of 0.9 nm obtained using Eq. R13.

$$S\sigma_{V_t} = \sigma_{V_t} \frac{0.9}{EOT \text{ (in nm)}}; \quad EOT = \frac{t_{OX}\epsilon_{SiO_2}}{\epsilon_{OX}} \quad [R13]$$

Here ϵ_{SiO_2} is the dielectric constant of SiO₂. Our MoS₂ and WS₂ FETs with 50 nm Al₂O₃ gate dielectric show $\sigma_{V_t} = 0.8 V$ that corresponds to $\sigma_n = 8 \times 10^{11} cm^{-2}$ and $S\sigma_{V_t} = 33 mV$. These values are in agreement with the variability projected for CVD grown monolayer MoS₂ FETs reported by Smithe *et al.* [1]. We also employed this method to other reports on top-gated and wafer scale monolayer MoS₂ FETs and extracted $S\sigma_{V_t} = 45 mV$ for [2] and $S\sigma_{V_t} = 11 mV$ for [3], respectively. Recently, Smets *et al.*, [4] have demonstrated $\sigma_{V_t} = 44 mV$ for an *EOT* of 1.9 nm that would correspond to $S\sigma_{V_t} = 20 mV$ for monolayer MoS₂ FETs with channel lengths scaled down to 30 nm. These results are summarized in Table R6. Remarkably, these variations are much smaller than $\sigma_{V_t} = 78.5 mV$ for an *EOT* of 0.33 nm, i.e., $S\sigma_{V_t} = 214 mV$ projected for UTB Si MOSFETs with a thickness of 2 nm [5]. Hence, 2D materials offer an alternative for the realization of UTB MOSFETs. Moreover, while it is possible to reduce the threshold voltage variation for 2D materials through optimization of growth and improvement in fabrication process flow, it is fundamentally limited for UTB Si owing to the significant thickness variation and random dopant fluctuations. In addition, detrimental quantum confinement effects lead to increase in the bandgap of ultra-thin Si. Finally, there is no known manufacturable solution for Si beyond the 5 nm technology node opening up opportunities for 2D materials for advanced scaling nodes.

Table R6. Benchmarking Variation in Threshold voltage				
	$\sigma_{V_t} (V)$	$\sigma_n (cm^{-2})$	Gate Dielectric	$S\sigma_{V_t} (V)$ at EOT = 0.9 nm
[1]-MoS ₂	1.05	8×10^{11}	30 nm SiO ₂	33×10^{-3}
[2]-MoS ₂ 1 continuous layer	0.25	1.1×10^{12}	30 nm HfO ₂	45×10^{-3}
[2]-MoS ₂ 1 layer + ML	0.1	4.6×10^{11}	30 nm HfO ₂	19×10^{-3}
[3]-MoS ₂	0.17	2.8×10^{11}	30 nm Al ₂ O ₃	11×10^{-3}
[4]-MoS ₂	44×10^{-3}	5×10^{11}	4 nm HfO ₂	20×10^{-3}
Our Work- MoS₂, WS₂	0.8	8×10^{11}	50 nm Al ₂ O ₃	33×10^{-3}
[5] UTB SOI	78.5×10^{-3}	5.1×10^{12}	EOT = 0.33 nm	214×10^{-3}

The discussion has been added in the revised manuscript.

12. “SS was also found to be independent of the channel length for both MoS₂ and WS₂, suggesting that detrimental short-channel effects are absent from the subthreshold device characteristics”. SS degradation is expected at shorter channel lengths than your minimum 100nm. I propose you mention you’re outside the range and shorter channels are needed to check this.

We agree with the reviewer. The statement has been revised accordingly.

Subthreshold slope (*SS*) was also found to be independent of the channel length for both MoS₂ and WS₂, for channel lengths in the range of 5 μm to 100 nm . This is not surprising since *SS* degradation are not expected to show up even for 100 nm channel length in UTB FETs [4].

13. “The deviation of the *SS* from the ideal value of 60 mV/dec is attributed to the presence of traps states” Can you explicitly evaluate *D_{IT}* statistics?

We are happy to extract the density of interface states (*D_{IT}*), and we agree with the reviewer that it will be useful to include the statistics of *D_{IT}*. *D_{IT}* was evaluated using Eq. R14.

$$SS = \frac{k_B T \ln(10)}{q} \left(1 + \frac{C_{IT}}{C_{OX}} \right); C_{IT} = q D_{IT} \quad [R14]$$

Here, *SS* across four orders of magnitude is used, k_B is the Boltzmann’s constant, T is the temperature, q is charge of an electron, C_{IT} is the capacitance due to interface traps, and C_{OX} is the oxide capacitance. The distribution of *D_{IT}* is shown using histograms in Fig. R13a and R13b for

Figure R13 Variation in density of interface traps is found for a) MoS₂ and b) WS₂ FETs.

MoS₂ and WS₂, respectively. Median values of $6.2 \times 10^{12} \text{ eV}^{-1}\text{cm}^{-2}$ and $8 \times 10^{12} \text{ eV}^{-1}\text{cm}^{-2}$ were obtained for MoS₂ and WS₂, respectively. The variation is summarized in Table R7.

Table R7: Variation in Interface Traps D_{IT} ($10^{12} \text{ eV}^{-1}\text{cm}^{-2}$)			
MoS₂		WS₂	
Median, Mean \pm SD	Min, Max	Median, Mean \pm SD	Min, Max
6.2, 6.6 \pm 2.3	1.75, 15.7	8, 8.1 \pm 0.7	6.5, 11.2

The discussion of D_{IT} and figures are added in the revised manuscript.

14. You first make the statement “... neither MoS2 nor WS2 FETs show any notable increase in DIBL as the channel length is scaled from 5um down to 300nm. This highlights the superior electrostatic integrity of 2D FETs ...” but then you mention DIBL is present at 100nm and 200nm. I recommend removing the first part, since only the shortest channel values are relevant. If you want to claim competitive DIBL of MX2 to Silicon, and can quote the DIBL value for Silicon at Lch=100nm?

We feel that a more extensive study using aggressively scaled devices is required to access drain-induced-barrier-lowering (*DIBL*) in 2D FETs. Therefore, we have removed the discussion on *DIBL* from our revised manuscript.

15. “Higher DIBL values for WS2 FETs may indicate stronger capacitive coupling between the channel and the drain electrode, but requires more in-depth investigation, which is beyond the scope of this paper.” I think this can be explained with WS2 having worse SS, so more Dit, and hence less electrostatic control by the gate, so more DIBL. The results are therefore in agreement.

We have removed the discussion on *DIBL* from our revised manuscript.

16. (crucial) “Remarkably, for “champion” devices DIBL values as low as 1.1 mV/V and 2.6 mV/V were recorded for MoS2 and WS2, respectively”. Given the large standard deviation,

and the large amount of devices with negative DIBL in the histogram, this DIBL value close to zero doesn't make sense, so I recommend removing this sentence.

We thank the reviewer for their comment. This sentence has been removed from the manuscript.

17. When extracting I_{on}/I_{off} , can you mention how the value is obtained? If it's actually I_{max}/I_{min} , this is strongly dependent on the chosen measurement voltage range. Can you mention if I_{off} is taken at a fixed offset from V_t and therefore mainly determined by SS, or by the gate leakage floor or something else? Is it possible to identify this for each device and e.g. color code the plots? Since the definition of I_{off} is lacking, i don't understand the statement "... that I_{on}/I_{off} is expected to be independent of the channel length, when extracted at constant electric field". I think the entire paragraph is difficult to understand and should be re-arranged so the goal and key message is stated clearly upfront.

We thank the reviewer for their suggestion. Please note that we have extracted the ratio of maximum to minimum current (I_{max}/I_{min}) instead of the ON-OFF ratio (I_{ON}/I_{OFF}) in the revised manuscript. Here, I_{max} is the maximum current obtained from the transfer characteristics for $V_{DS} = 1 V$ and I_{min} corresponds to the OFF-current. I_{min} corresponds to the noise floor of the measurement tool. I_{min} is determined by the noise floor of the measurement tool or the gate leakage floor and is found by averaging the current in the OFF state of each device. Hence, subthreshold slope is not considered while determining the OFF state of the device. I_{max}/I_{min} is mostly found to be channel length independent as shown in Fig. R14a and R14b for MoS₂ and WS₂, respectively. However, I_{max} demonstrates channel length dependence for longer channel devices with $L_{CH} \geq 1 \mu m$ as shown in Fig. R14c and R14d. For shorter channel devices with $L_{CH} < 1 \mu m$, this trend is obscured by the contact resistance. This is discussed in detail in the discussion of Fig. 7. Additionally, some variation is seen in I_{max} and I_{min} as shown in Fig. R14c and R14d. Hence, the combined effect of deviation from the linear scaling law for I_{max} and variation in I_{max} and I_{min} results in I_{max}/I_{min} being mostly independent of the channel length. Additionally, we agree that I_{max}/I_{min} is strongly dependent on the voltage range, and we have

extracted it across the entire measurement range for MoS₂ (V_{GS} from 14 V to -3 V) and WS₂ (V_{GS} from 12 V to -5 V).

Figure R14. Distribution of I_{max}/I_{min} for different channel lengths for a) MoS₂ and b) WS₂ FETs. Distribution of I_{max} and I_{min} for different channel lengths for c) MoS₂ and d) WS₂ FETs.

The discussion of I_{max}/I_{min} and figures are added in the revised manuscript and Supplementary Information file.

18. “Ion/Ioff recorded for both TMDs are at par with the current CMOS technology”. In Si CMOS, Ion/Ioff is mainly limited by fixed Vdd=0.7V. If for your devices you consider the entire sweep range, you need Vdd=20V. So i think you need to soften the statement.

We agree with the reviewer. The median value for I_{max}/I_{min} is 3.5×10^7 and 3.9×10^7 for both MoS₂ and WS₂ FETs, respectively for channel length of 100 nm. These values are over an order of magnitude higher than the I_{max}/I_{min} of 1.3×10^6 for UTB Si MOSFETs at similar channel lengths [6]. The comparison of I_{max}/I_{min} is tabulated in Table R8. Please note that in all demonstrations in Table R8, I_{max}/I_{min} is extracted across the entire sweep range of V_{GS}

($V_{GS,Range}$). Since the sweep range is strongly dependent on the EOT , it is useful to project the voltage ranges ($SV_{GS,Range}$) to a similar EOT of 0.9 nm using Eq. R15.

$$SV_{GS,Range} = \frac{V_{GS,Range} \times 0.9}{EOT} \quad [R15]$$

We demonstrate a higher I_{max}/I_{min} compared to UTB Si MOSFETs, however the measurement range for our devices is higher. Simulation results, however, have predicted similar I_{max}/I_{min} for both Si and MoS₂ FETs for aggressively scaled devices [13].

Table R8. Benchmarking Median I_{max}/I_{min} for $L_{CH} = 100\text{ nm}$				
	I_{max}/I_{min}	$EOT\text{ (nm)}$	$V_{GS,Range}\text{ (V)}$	$SV_{GS,Range}\text{ (V)}$ at $EOT = 0.9\text{ nm}$
[4]-MoS ₂	$\approx 7 \times 10^6$	1.9	1.5 to -0.5 (2)	0.94
[4]-MoS ₂	$\approx 4 \times 10^6$	50	-	-
Our Work - MoS ₂	3.5×10^7	22	14 to -3 (17)	0.65
Our Work - WS ₂	3.9×10^7	22	12 to -5 (17)	0.65
[6] UTB SOI	1.3×10^6	4	1.8 to -0.2 (2)	0.45

The discussion of I_{max}/I_{min} and figures are added in the revised manuscript.

19. “In summary, none of the key performance indicators related to the OFF-state, i.e. threshold voltage, SS, DIBL, Ion/Ioff show any noticeable and detrimental short-channel effects due to aggressive length scaling”. Your minimum channel length is 100nm, at which you observe an increased DIBL. But Silicon also has negligible short channel effects at 100nm. So i suggest strongly softening the statement.

We agree with the reviewer suggestions. We believe that none of our devices including the ones with $L_{CH} = 100\text{ nm}$ would qualify as short channel devices given that we are using ultra-thin body channel, i.e. monolayer MoS₂ and WS₂, and relatively thin and high-k gate dielectric, i.e. 50 nm thick Al₂O₃ with an EOT of 22 nm. Therefore, following reviewer’s recommendation, we have removed the general assertion and revised the statements accordingly.

20. 3D bar charts in fig.5a, b, i, j are hard to read. I think the format of e.g (c) is much easier to read, and with additional quantile boxplots in the same plot, all required data would be shown.

We thank the reviewer for their suggestion. We have shown the distribution of parameters for each channel length in 2D plots instead of 3D bar charts. Additionally, we have denoted the median, 25th and 75th percentile.

21. In general, I find the captions too long and hard to read, and they overlap strongly with the main text repeating the arguments instead of focusing on the key message. E.g. the caption of figure 4 contains this very long explanation which is better suited for the main text “The median values for these extracted threshold voltages (Table 1) were found to be more positive for WS₂ FETs compared to MoS₂ FETs, which can be ascribed to higher intrinsic n-type doping of MoS₂ either due to specific nature of impurity present in the MoS₂ film grown using metal-organic chemical vapor deposition (MOCVD) or surface charge transfer induced doping due to the underlying Al₂O₃ grown using atomic layer deposition (ALD). This charge transfer is accredited to the higher conduction band offset between MoS₂ and Al₂O₃ compared to WS₂ and Al₂O₃”

We thank the reviewer for their comment. The figure captions have been shortened.

22. The discussion of contact-limited on-state current and 2-point probe field effect mobility, and the need for TLM structures, has been discussed in many other papers. I suggest moving this discussion and the formulas to the appendix.

We thank the reviewer for their suggestion.

23. Fig.5(g, h) only show top performers. Can you consider putting either every TLM line on the plot, and the median e.g. in red, or maybe a band to show the outer limits? Additionally it would be useful to calculate the error bar on expectation value of the contact resistance.

We thank the reviewer for their suggestion. We are happy to include the data from all TLM structures. We have changed the plot as shown in Fig. R15a and R15b, for MoS₂ and WS₂, respectively. The median and the data points representing the 25th and 75th percentile is also shown.

Figure R15. Dependence of contact resistance on carrier density for all a) MoS₂ and b) WS₂ TLM structures.

We have made these changes in the revised manuscript.

24. Fig.5(k,l) did you consider stacked bar charts (for each L_{ch}: one bar for R_{ch}, one for R_c on top of it) to show the relative contribution of both resistances? It might be easier to understand intuitively.

We thank the reviewer for their suggestion and agree that stacked bar charts will be more intuitive. The have replaced the plots in Fig. 5k and 5l (Fig. 5g and 5h in the revised manuscript) using

Figure R16. Fraction of contact resistance and channel resistance to the total resistance for different channel length for a representative a) MoS₂ and b) WS₂ TLM structures.

stacked bar plots. The replaced plots are shown here in Fig. R16a and R16b, for MoS₂ and WS₂, respectively.

25. The median long-channel extrinsic field effect mobility extracted from gm is 24cm²/Vs for MoS₂ and 33cm²/Vs for WS₂. Since these are valid for the longest channels where the impact of the contact is negligible, the same values should be obtained from the TLM mobility. You mention “From the slopes, following Eq 6, the effective mobility was extracted as 18 cm²/Vs and 14 cm²/Vs for the long-channel devices, 2.6cm²/Vs and 1 cm²/Vs, for the short-channel devices for MoS₂ and WS₂, respectively” but it is not clear if this is the TLM mobility, which should yield only a single value valid for all channel lengths, extracted from the slope of fig5(e) and (f).

We are sorry for the confusion. We agree with the reviewer that TLM mobility should yield only a single value valid for all channel lengths.

We have revised our discussion and added the TLM mobility in the revised manuscript.

26. This is more about personal preference; Many paragraphs are starting with e.g. “Fig. 5a-d and Table 2 show”. I recommend starting with the key message/finding.

We thank the reviewer for their suggestion. We have made sure that we start with the key message/finding wherever possible.

27. (crucial) The overall structure of the article should be improved and should be more concise. The last section “Benchmarking Monolayer MoS₂ and WS₂ FETs” contains 11 paragraphs but only one is about benchmarking, so i recommend splitting this across different sections. The paragraph starting with “Finally, high performance FETs are benchmarked...” spans several pages and is about a mix of several topics (1. the need for higher Ion, 2. Ion-Lch dependence (already treated in previous paragraph), 3. contact resistance limited (already treated in previous paragraph), 4. pinchoff and velocity

saturation, 5. Ion of champion devices). We recommend rearranging this part, deleting repetitions, or moving parts to the supplementary, to keep only the key novel findings.

We thank the reviewer for their recommendation. We have rearranged this part.

28. [47] is reliable extraction of saturation velocity for MoS₂. Can you comment how your extraction method differs from theirs, and why you obtain different vsat values?

We have used the same method as described in [47] to estimate the saturation velocity from the output characteristics using the linear dependence of the saturation current on the overdrive voltage ($V_{GS} - V_{t_{in}}$) following Eq. R16.

$$\frac{I_{ON}}{W} = \frac{I_{DS,SAT}}{W} = C_{OX}v_{SAT}(V_{GS} - V_{t_{in}}) = qn_s v_{SAT} \quad [R16]$$

Using this equation, v_{SAT} of 0.9×10^6 cm/sec was obtained for MoS₂ in [47], which is similar to v_{SAT} of 1.1×10^6 cm/sec extracted from our data. However, in [47] authors reestimated v_{SAT} to be $(3.4 \pm 0.4) \times 10^6$ cm/sec by correcting for underestimation due to contact resistance and self-heating effects.

We have stated this fact in the revised manuscript.

29. Table 4 contains values for Ion, but the Vd (5V?) is not mentioned. Including the data at Vd=1V would be more relevant.

We agree with the reviewer's suggestion. We have included Table R9 in the revised manuscript.

Table R9: Variation in Drive Current								
$I_{ON}(\mu A/\mu m)$ at $V_{DS} = 1 V$					$I_{ON}(\mu A/\mu m)$ at $V_{DS} = 5 V$			
	MoS_2		WS_2		MoS_2		WS_2	
$L_{CH}(\mu m)$	Median, Mean \pm SD	Min, Max	Median, Mean \pm SD	Min, Max	Median, Mean \pm SD	Min, Max	Median, Mean \pm SD	Min, Max
0.1	54, 52 \pm 13	14, 73	17, 18 \pm 5	10, 26	146, 141 \pm 32	42, 177	30, 34 \pm 10	25, 53

0.2	46, 41 ± 18	2, 68	11, 13 ± 8	3, 27	126, 109 ± 47	9, 180	20, 25 ± 14	7, 50
0.3	41, 38 ± 14	1, 57	14, 15 ± 6	6, 24	126, 116 ± 38	6, 144	28, 30 ± 13	9, 51
0.4	36, 31 ± 14	2, 50	15, 14 ± 5	2, 20	110, 104 ± 41	7, 155	30, 31 ± 9	10, 49
0.5	35, 32 ± 12	1, 48	12, 12 ± 4	3, 19	121, 110 ± 37	4, 146	30, 28 ± 10	12, 46
1	25, 24 ± 8	2, 35	8, 8 ± 3	2, 13	99, 92 ± 29	11, 125	26, 25 ± 8	7, 35
2	17, 16 ± 5	2, 21	5, 4 ± 2	1, 8	70, 64 ± 20	9, 82	20, 18 ± 7	3, 26
3	12, 11 ± 4	1, 14	4, 4 ± 1	1, 6	49, 45 ± 15	5, 60	15, 15 ± 4	7, 20
4	10, 9 ± 2	1, 11	3, 3 ± 1	2, 4	40, 37 ± 8	8, 46	12, 12 ± 3	6, 16
5	8, 7 ± 2	2, 10	3, 3 ± 1	1, 4	32, 31 ± 7	8, 39	10, 10 ± 3	5, 15

30. Tables 5.1 and 5.2 show champion values only. Can you please add the median values (e.g. in brackets) for your work and for the references if they are reported.

We agree with the reviewer's suggestion. The median values have been added in Tables 5.1 and 5.2, both tables are combined to form one table as shown in Table R10.

Table R10: Benchmarking ON-state at $V_{DS} = 1\text{ V}$- Best(Median/Mean)				
	μ ($\text{cm}^2/\text{V}\cdot\text{s}$)	R_c ($\text{k}\Omega - \mu\text{m}$)	I_{ON} ($\mu\text{A}/\mu\text{m}$)	n_s (cm^{-2})
[1] – MoS ₂	$\mu_{gm} = 42$ (34.2)	0.73 (1)	22, $L_{CH} = 5.4\ \mu\text{m}$	1.3×10^{13}
[23] – MoS ₂	$\mu_{eff} = 20$	6.5	270, $L_{CH} = 80\ \text{nm}$	1×10^{13}
[3] – MoS ₂	$\mu_{gm} = 80$ (≈ 40)	2.4	13, $L_{CH} = 4\ \mu\text{m}$	6.6×10^{12}
[29] – MoS ₂	$\mu_{eff} = 30$	1.7	260, $L_{CH} = 10\ \text{nm}$	4.7×10^{13}
[4] – MoS ₂	$\mu_{eff} = 15$	1	250, $L_{CH} = 29\ \text{nm}$	1.5×10^{13}
[2] – MoS ₂	$\mu_{eff, 4\text{-point}} \approx 75$ (70)	14	-	-
Our Work – MoS ₂	$\mu_{eff} = 47$ (27)	3(9.2)	73 (54), $L_{CH} = 100\ \text{nm}$	1×10^{13}
[30] – WS ₂	$\mu_{gm} = 11$	-	25, $L_{CH} = 4\ \mu\text{m}$	2.1×10^{13}

[31] – WS ₂	$\mu_{g_m} = 20.4$	-	0.6, $L_{CH} = 1 \mu m$	2.5×10^{12}
[27] – WS ₂	$\mu_{g_m} = 5$	-	≈ 0.05 , $L_{CH} = 10 \mu m$	$\approx 7.2 \times 10^{12}$
[27] – WS ₂ (Graphene contact)	$\mu_{g_m} = 50$ (27)	-	≈ 1.1 , $L_{CH} = 10 \mu m$	$\approx 7.2 \times 10^{12}$
Our Work – WS ₂	$\mu_{eff} = 33$ (16)	2.1 (29)	26 (17), $L_{CH} = 100 nm$	4.4×10^{12}
[32] – UTB SOI	$\mu_{eff, 4-point} = 6$	-	$\approx 35 \times 10^{-3}$, $L_{CH} = 100 \mu m$	$\approx 9 \times 10^{12}$

31. The caption of fig6. has partially cropped text (FET?) on the last line.

We would like to thank the reviewer for pointing this out. This has been fixed in the manuscript.

32. [48] has Quentin Smets

We are not sure what the reviewer meant. However, we have added the reference.

References

- [1] K. K. H. Smithe, S. V. Suryavanshi, M. Munoz Rojo, A. D. Tedjarati, and E. Pop, "Low Variability in Synthetic Monolayer MoS₂ Devices," *ACS Nano*, vol. 11, pp. 8456-8463, Aug 22 2017.
- [2] H. Xu, H. Zhang, Z. Guo, Y. Shan, S. Wu, J. Wang, *et al.*, "High-Performance Wafer-Scale MoS₂ Transistors toward Practical Application," *Small*, vol. 14, p. e1803465, Nov 2018.
- [3] L. Yu, D. El-Damak, U. Radhakrishna, X. Ling, A. Zubair, Y. Lin, *et al.*, "Design, Modeling, and Fabrication of Chemical Vapor Deposition Grown MoS₂ Circuits with E-Mode FETs for Large-Area Electronics," *Nano Lett*, vol. 16, pp. 6349-6356, Oct 12 2016.
- [4] Q. Smets, B. Groven, M. Caymax, I. Radu, G. Arutchelvan, J. Jussot, *et al.*, "Ultra-scaled MOCVD MoS₂ MOSFETs with 42nm contact pitch and 250μA/μm drain current," pp. 23.2.1-23.2.4, 2019.
- [5] K. Samsudin, F. Adamu-Lema, A. R. Brown, S. Roy, and A. Asenov, "Combined sources of intrinsic parameter fluctuations in sub-25nm generation UTB-SOI MOSFETs: A statistical simulation study," *Solid-State Electronics*, vol. 51, pp. 611-616, 2007.
- [6] C. Min, T. Kamins, P. V. Voorde, C. Diaz, and W. Greene, "0.18-μm fully-depleted silicon-on-insulator MOSFET's," *IEEE Electron Device Letters*, vol. 18, pp. 251-253, 1997.
- [7] C. Anghel, A. Ionescu, N. Hefyene, and R. Gillon, "Self-heating characterization and extraction method for thermal resistance and capacitance in high voltage MOSFETs," in *ESSDERC'03. 33rd Conference on European Solid-State Device Research, 2003.*, 2003, pp. 449-452.
- [8] S. Shin, M. Masduzzaman, J. Gu, M. Wahab, N. Conrad, M. Si, *et al.*, "Impact of nanowire variability on performance and reliability of gate-all-around III-V MOSFETs," in *2013 IEEE International Electron Devices Meeting, 2013*, pp. 7.5. 1-7.5. 4.
- [9] S. Islam, Z. Li, V. E. Dorgan, M.-H. Bae, and E. Pop, "Role of Joule heating on current saturation and transient behavior of graphene transistors," *IEEE electron device letters*, vol. 34, pp. 166-168, 2013.
- [10] X. Li, L. Yang, M. Si, S. Li, M. Huang, P. Ye, *et al.*, "Performance potential and limit of MoS₂ transistors," vol. 27, pp. 1547-52, Mar 4 2015.
- [11] E. Yalon, C. J. McClellan, K. K. Smithe, M. Muñoz Rojo, R. L. Xu, S. V. Suryavanshi, *et al.*, "Energy dissipation in monolayer MoS₂ electronics," *Nano Letters*, vol. 17, pp. 3429-3433, 2017.
- [12] T. M and D. J. Late, "Temperature dependent phonon shifts in single-layer WS(2)," *ACS Appl Mater Interfaces*, vol. 6, pp. 1158-63, Jan 22 2014.
- [13] K. Alam and R. K. Lake, "Monolayer MoS₂ Transistors Beyond the Technology Road Map," *IEEE Transactions on Electron Devices*, vol. 59, pp. 3250-3254, 2012.
- [14] J. Guo, B. Yang, Z. Zheng, and J. Jiang, "Observation of abnormal mobility enhancement in multilayer MoS₂ transistor by synergy of ultraviolet illumination and ozone plasma treatment," *Physica E: Low-dimensional Systems and Nanostructures*, vol. 87, pp. 150-154, 2017.
- [15] Z. Yu, Y. Pan, Y. Shen, Z. Wang, Z. Y. Ong, T. Xu, *et al.*, "Towards intrinsic charge transport in monolayer molybdenum disulfide by defect and interface engineering," *Nat Commun*, vol. 5, p. 5290, Oct 20 2014.
- [16] J. R. Nasr, D. S. Schulman, A. Sebastian, M. W. Horn, and S. Das, "Mobility Deception in Nanoscale Transistors: An Untold Contact Story," *Adv Mater*, vol. 31, p. e1806020, Jan 2019.
- [17] D. S. Schulman, A. J. Arnold, and S. Das, "Contact engineering for 2D materials and devices," *Chem Soc Rev*, vol. 47, pp. 3037-3058, May 8 2018.
- [18] S. Das, H. Y. Chen, A. V. Penumatcha, and J. Appenzeller, "High performance multilayer MoS₂ transistors with scandium contacts," *Nano Lett*, vol. 13, pp. 100-5, Jan 9 2013.
- [19] S. Lee, A. Tang, S. Aloni, and H. S. Wong, "Statistical Study on the Schottky Barrier Reduction of Tunneling Contacts to CVD Synthesized MoS₂," *Nano Lett*, vol. 16, pp. 276-81, Jan 13 2016.

- [20] Y. Liu, J. Guo, Y. Wu, E. Zhu, N. O. Weiss, Q. He, *et al.*, "Pushing the Performance Limit of Sub-100 nm Molybdenum Disulfide Transistors," *Nano Lett*, vol. 16, pp. 6337-6342, Oct 12 2016.
- [21] L. Yang, K. Majumdar, H. Liu, Y. Du, H. Wu, M. Hatzistergos, *et al.*, "Chloride molecular doping technique on 2D materials: WS₂ and MoS₂," *Nano Lett*, vol. 14, pp. 6275-80, Nov 12 2014.
- [22] P. Luo, F. Zhuge, Q. Zhang, Y. Chen, L. Lv, Y. Huang, *et al.*, "Doping engineering and functionalization of two-dimensional metal chalcogenides," *Nanoscale Horiz*, vol. 4, pp. 26-51, Jan 1 2019.
- [23] K. K. H. Smithe, C. D. English, S. V. Suryavanshi, and E. Pop, "Intrinsic electrical transport and performance projections of synthetic monolayer MoS₂ devices," *2D Materials*, vol. 4, p. 011009, 2016.
- [24] Y. Zhu, Y. Li, G. Arefe, R. A. Burke, C. Tan, Y. Hao, *et al.*, "Monolayer Molybdenum Disulfide Transistors with Single-Atom-Thick Gates," *Nano Lett*, vol. 18, pp. 3807-3813, Jun 13 2018.
- [25] Y. Cui, R. Xin, Z. Yu, Y. Pan, Z. Y. Ong, X. Wei, *et al.*, "High-Performance Monolayer WS₂ Field-Effect Transistors on High-kappa Dielectrics," *Adv Mater*, vol. 27, pp. 5230-4, Sep 16 2015.
- [26] D. Ovchinnikov, A. Allain, Y. S. Huang, D. Dumcenco, and A. Kis, "Electrical transport properties of single-layer WS₂," *ACS Nano*, vol. 8, pp. 8174-81, Aug 26 2014.
- [27] A. S. Aji, P. Solís-Fernández, H. G. Ji, K. Fukuda, and H. Ago, "High Mobility WS₂ Transistors Realized by Multilayer Graphene Electrodes and Application to High Responsivity Flexible Photodetectors," *Advanced Functional Materials*, vol. 27, p. 1703448, 2017.
- [28] T. Li and G. Galli, "Electronic Properties of MoS₂ Nanoparticles," *The Journal of Physical Chemistry C*, vol. 111, pp. 16192-16196, 2007/11/01 2007.
- [29] C. D. English, K. K. H. Smithe, R. L. Xu, and E. Pop, "Approaching ballistic transport in monolayer MoS₂ transistors with self-aligned 10 nm top gates," pp. 5.6.1-5.6.4, 2016.
- [30] Y. Gong, V. Carozo, H. Li, M. Terrones, and T. N. Jackson, "High flex cycle testing of CVD monolayer WS₂ TFTs on thin flexible polyimide," *2D Materials*, vol. 3, p. 021008, 2016.
- [31] S. J. Yun, S. H. Chae, H. Kim, J. C. Park, J. H. Park, G. H. Han, *et al.*, "Synthesis of centimeter-scale monolayer tungsten disulfide film on gold foils," *ACS Nano*, vol. 9, pp. 5510-9, May 26 2015.
- [32] M. Schmidt, M. C. Lemme, H. D. B. Gottlob, F. Driussi, L. Selmi, and H. Kurz, "Mobility extraction in SOI MOSFETs with sub 1nm body thickness," *Solid-State Electronics*, vol. 53, pp. 1246-1251, 2009.

REVIEWER COMMENTS

Reviewer #2 (Remarks to the Author):

The new data and revised discussion have significantly improved the manuscript.

I have only a few minor clarifications and questions for the authors to consider.

(1) Regarding the response to comment 4 and the new Fig. R1. What field-effect mobility is extracted from the dependence of I_{ON}/W on $1/LCH$? It seems this should go in the figures. Since $qns = COX(VGS-V_{TH})$, how was VGS tuned to set n_s precisely at $1e13 /cm^2$ or $4.4e13/cm^2$ for each transistor; this seems tricky since V_{TH} has variation. I see conditions for measuring qns are given later. In light of this, consider when it is best to explain how n_s is set. How does the mobility extracted in this method compare with your method of Eq. (R2)? How are these various values to be understood.

(2) New Fig. R1(c) and (d). How was contact resistance measured? If these contacts are barrier limited they are bias dependence; how was contact resistance determined, and VGS and VDS chosen and adjusted to produce figures (c) and (d).

(3) The statement "there is no known manufacturable solution for Si beyond the 5 *nmn* technology node." Please provide a reference to this claim. 3 nm and 2 nm nodes are under consideration and there is always no manufacturable solution until it is in production.

Reviewer #3 (Remarks to the Author):

The author has very well addressed the comments. Thank you for the detailed response letter. I have a few additional comments about σ_{Vt} .

1. Instead of σ_{Vt} , ideally $\sigma_{\Delta Vt}$ is extracted using mismatch pairs to eliminate extrinsic variability. Also, $\sigma_{\Delta Vt}$ should decrease as the device dimensions are increased (more averaging over a larger area) and go to zero in the limit of very large dimensions. Therefore it is ideally converted to a Pelgrom slope A_{Vt} which is a measure of intrinsic random variability. I think this goes beyond the scope of the paper, and in the main text and the tables, I suggest reporting the device dimensions for which σ_{Vt} is extracted.

2. Regarding the formula $S\sigma_{Vt}=SEOT/EOT$, this formula is expected to be valid when the dominant variability source scales with the electrostatic control and hence EOT. In this paper: "Variability sources in nanoscale bulk FinFETs and TiTaN- a promising low variability WFM for 7/5nm CMOS nodes" by M. Bhoir et al, IEDM 2019 (<https://ieeexplore.ieee.org/document/8993660>), σ_{Vt} is converted to Pelgrom slope A_{Vt} , and fig.10 shows A_{Vt} indeed scales linearly with EOT, but only up to the point that metal gate granularity becomes dominant. I suggest adding a warning to the text that the formula $S\sigma_{Vt}=SEOT/EOT$ is likely no longer valid for ultra-scaled EOT and add the reference.

3. Regarding table 1 reference [32] for UTB SOI, $\sigma_{Vt}=78mV$ seems quite high. I suggest putting the reference above instead which has $\sigma_{Vt}=10mV$ for the smallest reported device dimensions and physical oxide thickness=20Å, EOT=0.8nm. The value of 10mV is quite small, which shows 2D materials still need a lot of effort to achieve competitive variability.

Quentin Smets

Response to Reviewers' Comments

Reviewer's Comment

Our Response

Changes Made in the Manuscript

Reviewer #2 (Remarks to the Author):

The new data and revised discussion have significantly improved the manuscript. I have only a few minor clarifications and questions for the authors to consider.

We are glad to know that the reviewer finds that the manuscript has been significantly improved. Please see a point-by-point response to the new comments below.

1. Regarding the response to comment 4 and the new Fig. R1. What field-effect mobility is extracted from the dependence of I_{ON}/W on $1/L_{CH}$? It seems this should go in the figures. Since $qns = COX(VGS-V_{TH})$, how was VGS tuned to set n_s precisely at $1e13 /cm^2$ or $4.4e13/cm^2$ for each transistor; this seems tricky since V_{TH} has variation. I see conditions for measuring qns are given later. In light of this, consider when it is best to explain how n_s is set. How does the mobility extracted in this method compare with your method of Eq. (R2)? How are these various values to be understood?

We extracted $\mu = 18 \text{ cm}^2/Vs$ and $14 \text{ cm}^2/Vs$, for MoS₂ and WS₂, respectively, from the slopes of median I_{ON}/W versus $1/L_{CH}$, as shown in Fig. R1a and R1b for the longer channel devices ($L_{CH} \geq 1 \text{ }\mu m$) following Eq R1.

$$\frac{I_{ON}}{W} = qn_s\mu \frac{V_{DS}}{L_{CH}} \quad [R1]$$

Compared to this, median mobility obtained from peak transconductance (μ_{g_m}) using Eq. R2 across the all longer channel devices ($L_{CH} \geq 1 \text{ }\mu m$) was found to be $20 \text{ cm}^2/Vs$ and $15 \text{ cm}^2/Vs$, for MoS₂ and WS₂, respectively. Hence the mobility values are very similar in both the cases.

$$\mu_{g_m} = \frac{dI_{DS}}{dV_{GS}} \left(\frac{L_{CH}}{WC_{OX}V_{DS}} \right) \quad [R2]$$

This discussion was not included in the revised manuscript since we already have significant discussion on mobility extraction using different methods. However, we have included Fig. R1a and R1b in Fig. 7 in the revised manuscript.

Figure R1. Extracted median values for I_{ON} as a function of $1/L_{CH}$ for a) MoS_2 and b) WS_2 FETs at $V_{DS} = 1 V$.

In order to report I_{ON} at constant $n_S = 1 \times 10^{13} cm^{-2}$ and $n_S = 4.4 \times 10^{12} cm^{-2}$ for all MoS_2 and WS_2 FETs, we first calculated the required overdrive voltage ($V_{GS} - V_{t_{lin}}$) following Eq R3.

$$n_S = \frac{C_{OX}(V_{GS} - V_{t_{lin}})}{q} \quad [R3]$$

Here, C_{OX} is the oxide capacitance, V_{GS} is the gate voltage and $V_{t_{lin}}$ is the threshold voltage extracted using the linear extrapolation method. Since $V_{t_{lin}}$ was extracted for all devices, we can identify the corresponding V_{GS} values for reporting the I_{ON} for each of them. I_{ON} is extracted from the output characteristics with a V_{GS} step size of 2 V, and the median error in n_S is $0.11 \times 10^{12} cm^{-2}$ and $0.03 \times 10^{12} cm^{-2}$ for MoS_2 and WS_2 FETs, respectively. n_S is also found for the analysis of R_C . R_C is extracted from the transfer characteristics with a V_{GS} step size of 85 mV, and median error n_S is 0.003 and $0.004 \times 10^{12} cm^{-2}$ for MoS_2 and WS_2 FETs, respectively.

Fig. R1a and R1b in has been included Fig. 7 in the main manuscript.

We have added the following statement for n_S extraction in **Supplementary Note 4**: To obtain a constant n_S , a constant overdrive voltage ($V_{GS} - V_{t_{lin}}$) is ensured by extracting the $V_{t_{lin}}$ and then estimating the required V_{GS} for every device. I_{ON} is extracted from the output characteristics with a V_{GS} step size of 2 V, and the median error in n_S is $0.11 \times 10^{12} cm^{-2}$ and $0.03 \times 10^{12} cm^{-2}$ for MoS_2 and WS_2 FETs, respectively. n_S is also found for the analysis of R_C . R_C is extracted from the

transfer characteristics with a V_{GS} step size of 85 mV, and median error n_S is 0.003 and $0.004 \times 10^{12} \text{ cm}^{-2}$ for MoS₂ and WS₂ FETs, respectively.

2. New Fig. R1(c) and (d). How was contact resistance measured? If these contacts are barrier limited, they are bias dependence; how was contact resistance determined, and VGS and VDS chosen and adjusted to produce figures (c) and (d).

Reviewer's observation is correct. Contact resistance (R_C) is indeed dependent on the Schottky barrier width and hence V_{GS} . This leads to the dependence of R_C on the carrier concentration (n_S). To ensure consistency, R_C is extracted for a constant $n_S = 1 \times 10^{13} \text{ cm}^{-2}$ and $n_S = 4.4 \times 10^{13} \text{ cm}^{-2}$ for MoS₂ and WS₂ FETs, respectively, and for a fixed $V_{DS} = 1 \text{ V}$.

3. The statement “there is no known manufacturable solution for Si beyond the 5 nm technology node.” Please provide a reference to this claim. 3 nm and 2 nm nodes are under consideration and there is always no manufacturable solution until it is in production.

We agree with the reviewer's comment. We have revised the statement as follows.

It is encouraging that monolayer 2D FETs show $S\sigma_{V_t}$ comparable to the state-of-the-art Si FETs in spite of an order of magnitude smaller body thickness. Note that UTB Si FETs are expected to encounter challenges associated with the precise thickness control, random dopant fluctuations, and detrimental quantum confinement effects beyond 5 nm body thickness [1, 2], which are unlikely for 2D monolayers. At the same time further improvement in threshold voltage variation can be achieved for 2D FETs through optimization of the monolayer growth and improvement in the fabrication process flow (see *Supplementary Note 2* for further discussion).

Reviewer #3 (Remarks to the Author):

The author has very well addressed the comments. Thank you for the detailed response letter. I have a few additional comments about σ_{Vt} .

We are happy to know that the reviewer is satisfied with our response. We thank the reviewer for their new insightful comments on σ_{Vt} . Please see a point-by-point response to the new comments below.

1. Instead of σ_{Vt} , ideally $\sigma_{\Delta Vt}$ is extracted using mismatch pairs to eliminate extrinsic variability. Also, $\sigma_{\Delta Vt}$ should decrease as the device dimensions are increased (more averaging over a larger area) and go to zero in the limit of very large dimensions. Therefore it is ideally converted to a Pelgrom slope A_{Vt} which is a measure of intrinsic random variability. I think this goes beyond the scope of the paper, and in the main text and the tables, I suggest reporting the device dimensions for which σ_{Vt} is extracted.

We thank the reviewer for pointing this out. We have added the device dimensions in the text and table (Table R1). The following statement has also been added: Note that σ_{Vt} is found to be inversely proportional to the channel area in ultra-scaled devices as demonstrated using Pelgrom plots [3, 4]. We did not observe such a trend due to the relatively large channel area of our MoS₂ and WS₂ FETs.

2. Regarding the formula $\sigma_{Vt} = SEOT/EOT$, this formula is expected to be valid when the dominant variability source scales with the electrostatic control and hence EOT. In this paper: "Variability sources in nanoscale bulk FinFETs and TiTaN- a promising low variability WFM for 7/5nm CMOS nodes" by M. Bhoir et al, IEDM 2019 (<https://ieeexplore.ieee.org/document/8993660>), σ_{Vt} is converted to Pelgrom slope A_{Vt} , and fig.10 shows A_{Vt} indeed scales linearly with EOT, but only up to the point that metal gate granularity becomes dominant. I suggest adding a warning to the text that the formula $\sigma_{Vt} = SEOT/EOT$ is likely no longer valid for ultra-scaled EOT and add the reference.

We agree with the reviewer's suggestion. We have added the following statement: This equation assumes linear scaling of variation in threshold voltage with respect to the EOT . However, for ultra-scaled devices, deviation from the linear scaling can be expected due to increased effect of metal-gate granularity [3].

3. Regarding table 1 reference [32] for UTB SOI, $\sigma_{V_t}=78\text{mV}$ seems quite high. I suggest putting the reference above instead which has $\sigma_{V_t}=10\text{mV}$ for the smallest reported device dimensions and physical oxide thickness= 20\AA , $EOT=0.8\text{nm}$. The value of 10mV is quite small, which shows 2D materials still need a lot of effort to achieve competitive variability.

We agree with the reviewer that the variation of 78.5 mV is quite high. We have replaced the reference with a new reference with an UTB fully depleted silicon-on-insulator structure showing a much lower variation for $t_{body} = 7\text{ nm}$ [4]. However, when scaled beyond thickness of 5 nm , an increase in variation is expected owing to quantum confinement effects and thickness variations [1]. Additionally, the reference suggested by the reviewer, demonstrating variation in FinFETs has been added [5].

Table R1. Benchmarking device-to-device variation in threshold voltage				
	σ_{V_t} (V)	Gate Dielectric	$S\sigma_{V_t}$ (V) at $SEOT = 0.9\text{ nm}$	Channel dimensions (μm)
[6]-MoS ₂	1.05	30 nm SiO ₂	33×10^{-3}	$W = 11.6, L_{CH} = 4- 8.6$
[7]-MoS ₂ 1 continuous layer	0.25	30 nm HfO ₂	45×10^{-3}	$W = -, L_{CH} = 30$
[7]-MoS ₂ 1 layer + ML	0.1	30 nm HfO ₂	19×10^{-3}	$W = -, L_{CH} = 30$
[8]-MoS ₂	0.17	30 nm Al ₂ O ₃	11×10^{-3}	$W =30, L_{CH} = 4$
[9]-MoS ₂	44×10^{-3}	4 nm HfO ₂	20×10^{-3}	$W = 1, L_{CH} = 0.1$
Our Work- MoS₂, WS₂	0.8	50 nm Al ₂ O ₃	33×10^{-3}	$W = 5, L_{CH} = 0.1, 0.2, 0.3, 0.4, 0.5, 1, 2, 3, 4, 5$
[4] UTB SOI	24.5×10^{-3}	$EOT = 1.65\text{ nm}$	13×10^{-3}	$W = 0.060, L_{CH} = 0.025$
[3] FinFET	10×10^{-3}	$EOT = 0.8\text{ nm}$	11×10^{-3}	$W = 0.0075, L_{CH} = 0.034$

References

- [1] G. Tsutsui, M. Saitoh, T. Nagumo, and T. Hiramoto, "Impact of SOI Thickness Fluctuation on Threshold Voltage Variation in Ultra-Thin Body SOI MOSFETs," *IEEE Transactions On Nanotechnology*, vol. 4, pp. 369-373, 2005.
- [2] K. Samsudin, F. Adamu-Lema, A. R. Brown, S. Roy, and A. Asenov, "Combined sources of intrinsic parameter fluctuations in sub-25nm generation UTB-SOI MOSFETs: A statistical simulation study," *Solid-State Electronics*, vol. 51, pp. 611-616, 2007.
- [3] M. S. Bhoir, T. Chiarella, L. Å. Ragnarsson, J. Mitard, N. Horiguchi, and N. R. Mohapatra, "Variability sources in nanoscale bulk FinFETs and TiTaN- a promising low variability WFM for 7/5nm CMOS nodes," in *2019 IEEE International Electron Devices Meeting (IEDM)*, 2019, pp. 36.2.1-36.2.4.
- [4] O. Weber, O. Faynot, F. Andrieu, C. Buj-Dufournet, F. Allain, P. Scheiblin, *et al.*, "High immunity to threshold voltage variability in undoped ultra-thin FDSOI MOSFETs and its physical understanding," pp. 1-4, 2008.
- [5] M. S. Bhoir, T. Chiarella, L. A. Ragnarsson, J. Mitard, N. Horiguchi, and N. R. Mohapatra, "Variability sources in nanoscale bulk FinFETs and TiTaN- a promising low variability WFM for 7/5nm CMOS nodes," pp. 36.2.1-36.2.4, 2019.
- [6] K. K. H. Smithe, S. V. Suryavanshi, M. Munoz Rojo, A. D. Tedjarati, and E. Pop, "Low Variability in Synthetic Monolayer MoS₂ Devices," *ACS Nano*, vol. 11, pp. 8456-8463, Aug 22 2017.
- [7] H. Xu, H. Zhang, Z. Guo, Y. Shan, S. Wu, J. Wang, *et al.*, "High-Performance Wafer-Scale MoS₂ Transistors toward Practical Application," *Small*, vol. 14, p. e1803465, Nov 2018.
- [8] L. Yu, D. El-Damak, U. Radhakrishna, X. Ling, A. Zubair, Y. Lin, *et al.*, "Design, Modeling, and Fabrication of Chemical Vapor Deposition Grown MoS₂ Circuits with E-Mode FETs for Large-Area Electronics," *Nano Lett*, vol. 16, pp. 6349-6356, Oct 12 2016.
- [9] Q. Smets, B. Groven, M. Caymax, I. Radu, G. Arutchelvan, J. Jussot, *et al.*, "Ultra-scaled MOCVD MoS₂ MOSFETs with 42nm contact pitch and 250 μ A/ μ m drain current," pp. 23.2.1-23.2.4, 2019.